# Differentially Private Source-Target Clustering

**Shachar Schnapp**                                    *schnapp@post.bgu.ac.il*
*Department of Computer Science, Ben-Gurion University of the Negev*

**Sivan Sabato**                                       *sabatos@mcmaster.ca*
*Department of Computing and Software, McMaster University*
*Canada CIFAR AI Chair, Vector Institute*
*Department of Computer Science, Ben-Gurion University of the Negev*

**Reviewed on OpenReview:** *https: // openreview. net/ forum? id= ojeCoOKwWp*

## Abstract

We consider a new private variant of the Source-Target Clustering (STC) setting, which was introduced by de Mathelin et al. (2022). In STC, there is a target dataset that needs to be clustered by selecting centers, in addition to centers that are already provided in a separate source dataset. The goal is to select centers from the target, such that the target clustering cost given the additional source centers is minimized. We consider *private STC*, in which the source dataset is private and should only be used under the constraint of differential privacy. This is motivated by scenarios in which the existing centers are private, for instance because they represent individuals in a social network. We derive lower bounds for the private STC objective, illustrating the theoretical limitations on worst-case guarantees for this setting. We then present a differentially private algorithm with asymptotically advantageous results under a *data-dependent* analysis, in which the guarantee depends on properties of the dataset, as well as more practical variants. We demonstrate in experiments the reduction in clustering cost that is obtained by our practical algorithms compared to baseline approaches. Code is publicly available on https://github.com/ShacharSchnapp/STC.

## 1    Introduction

*Differential Privacy* (DP) (Dwork et al., 2006c) is the gold standard for privacy-preservation in data-intensive tasks, defined so as to prevent information about specific individuals from leaking from a dataset. Differential privacy is usually implemented by adding randomness to a computation such that it becomes impossible to determine the specifics of any individual's data, while still allowing the data to be used for statistical analysis and machine learning tasks. It has been a popular topic of research in the field of machine learning, where researchers have applied its principles to a variety of tasks, including quantile calculation (Kaplan et al., 2022), logistic regression Chaudhuri et al. (2011), principal component analysis (Hardt & Roth, 2013), boosting (Dwork et al., 2010b), support vector machines (Senekane, 2019), and deep learning (Abadi et al., 2016). In this work, we study the application of DP to the problem of *Source-Target Clustering (STC)*.

The STC problem, first introduced by De Mathelin et al. (2022), involves a source dataset $S$ and a target dataset $T$. The goal is to cluster the target dataset $T$ under the $k$-medoids cost function. In other words, the goal is to find up to $k$ centers from $T$, such that the total distance of any point in $T$ from its closest center is the smallest. However, unlike standard $k$-medoids clustering, in STC all of the data points in $S$ are already selected as centers for "free", that is, they do not count towards the budget of $k$ centers. Thus, the objective is to select $k$ points from $T$, such that that the medoids cost over $T$, when using these points as centers in addition to the points in $S$, is as small as possible.

In this work, we study a private version of this problem, in which the source dataset $S$ is considered private and the target dataset $T$ is considered public. We would like to achieve the same goal as above, while

guaranteeing differential privacy for the source dataset $S$. In other words, the selection of the $k$ additional centroids from $T$ must be done while preserving the privacy of $S$.

**Applications.** Private STC is relevant in any scenario in which an existing set of centers can be used when selecting centers for a target data set, but the selected centers must not violate the privacy of the existing centers. We provide here several concrete examples.

**Maximizing influence in a social network.** The goal is to find a set of individuals in the network that should be approached to promote a certain message or product. The target dataset $T$ is the set of individuals that we wish to reach during this promotion, and so we wish to minimize the total distance (in the social network) of each individual from a promoting individual. We have a budget of $k$ individuals that we can approach. In addition, there are individuals in the social network who are already promoting this message or product independently. The set $S$ consists of these individuals. Each of them can be considered a center for the purpose of calculating the cost on $T$, but their identities should not be leaked, as their participation in the campaign is confidential at this stage. Thus, the choice of users to approach must be done such that the privacy of $S$ is preserved.

**Selecting first-aid service locations.** The goal is to decide on the location of first-aid service points within a residential area. The target dataset $T$ is the set of geographical locations that need to be covered by first-aid services, and there is a budget of $k$ service locations. However, there are already residents who are on-call paramedics and can be dispatched by authorities to provide first-aid services. The dataset $S$ that lists these residents is not open to the public, and should not be divulged by the choice of additional service locations.

**Domain Adaptation**. This was the original motivation of de Mathelin et al. (2022) to introduce the original (non-private) STC problem. Domain Adaptation (see, e.g., Nigam et al., 2000; Long et al., 2015) is a setting in which a dataset from a source distribution is available, and we wish to use it to learn a classifier for a target distribution, with the help of a relatively small number of labeled data from the target. De Mathelin et al. (2022) showed that the task of domain adaptation (under certain conditions) can be solved via a reduction to the STC problem. In this case, $S$ is a dataset of points from the source distribution, and $T$ is a dataset of points from the target distribution. $k$ is the budget for labeling examples from $T$, while it is assumed that all of $S$ is already labeled. De Mathelin et al. (2022) show that by selecting to label the $k$ examples from $T$ that minimize the medoids cost based on taking these $k$ examples and all the examples in $S$ as centers, a target hypothesis can be successfully learned by applying empirical risk minimization to the resulting combined set of labeled examples. Note that STC does not require the labels of $S$, although in this case they are available.

In this case, private STC is concerned with domain adaptation scenarios in which the dataset $S$ is private. For instance, suppose that a national research institute (NRI) holds a private country-wide labeled medical dataset of patients, while a local pharmaceutical company (PC) wants to train classifiers to predict treatment outcomes for patients in a specific region. PC has an unlabeled medical dataset $T$ that represents the patients in the region, which it is allowed to share with NRI, and the budget to label $k$ of its entries. NRI would thus like to provide PC with a list of examples to label out of $T$, without breaching the privacy of its own dataset. NRI can use private STC, by setting $S$ to be the list of its private examples, and providing PC with $k$ examples to label from $T$. The reduction of de Mathelin et al. (2022) implies that *private* domain adaptation can be done using a solution to the private STC problem that we present here, together with a private ERM for minimizing the source risk, of which there are many off-the-shelf solutions.

**Summary of contributions.** We first formally define the new private formulation of the STC problem (Section 3). We then derive (Section 4) strong lower bounds for private STC (Observation 4.2), showing theoretical barriers on the achievable worst-case guarantees. Having shown the limitations of worst-case guarantees for this setting, we provide (Section 5) algorithms with *data-dependent* guarantees—guarantees that depend on properties of the input data (Theorems 5.5, 5.10, 5.12). We also provide a lower bound that shows that in the worst case, the error could strongly depend on the number of clusters, where fewer clusters lead to a larger worst-case error (Observation 5.8). We further present a variant of the algorithm that works better empirically, and demonstrate in experiments (Section 6) its ability to reduce the clustering cost compared to several baselines.

## 2 Related work

Differentially private clustering has been gaining attention in recent years and various settings have been studied. Feldman et al. (2009) introduced the concept of private coresets, which can be used to reduce the amount of data that needs to be released in order to perform certain types of data analysis. These could potentially be used to privately perform clustering on a subset of the data. Gupta et al. (2010) developed foundational techniques for incorporating differential privacy into clustering algorithms, addressing challenges in privacy-preserving data partitioning and analysis. Wang et al. (2015) proposed a differentially private algorithm for subspace clustering. Nock et al. (2016) presented a differentially private variant of $k$-means++, for a setting in which the entire dataset to cluster is private. Su et al. (2016) proposed a differentially private algorithm for $k$-means clustering. Feldman et al. (2017) studied the use of coresets for differentially private $k$-means clustering and its applications in mobile sensor networks. Balcan et al. (2017a) proposed a differentially private algorithm for clustering in high-dimensional Euclidean spaces. Nissim & Stemmer (2018) compared the centralized and local differential privacy models in the context of clustering algorithms. Huang & Liu (2018) presented differentially private algorithms for $k$-means clustering that have optimal sample complexity and computational complexity. Stemmer & Kaplan (2018) proposed a differentially private algorithm for $k$-means clustering with a constant multiplicative error. Ghazi et al. (2020) achieved tight approximation ratios for differentially private clustering. We are not aware of any works that considered the joint private-non-private clustering setting that we study here.

The STC setting was introduced by de Mathelin et al. (2022), motivated by a Domain Adaptation scenario. The setting of Domain Adaptation was first suggested in Nigam et al. (2000). Since then, it has been studied under various scenarios (Sugiyama et al., 2008; Pan et al., 2008; Silva et al., 2012; Shen et al., 2013; Ganin & Lempitsky, 2015) and applied to a variety of applications, such as natural language processing (Blitzer et al., 2007; Ben-David et al., 2006; Jiang & Zhai, 2007; Ramponi & Plank, 2020), speech processing (Leggetter & Woodland, 1995; Gauvain & Lee, 1994; Bacchiani & Roark, 2003; Sun et al., 2017) and computer vision (Martinez, 2002; Xu et al., 2019). In domain adaptation, a classifier is trained on a dataset from a source distribution and then used for prediction on a target distribution. To adapt the model to the target domain when target labels are not available, if a machine learning model is trained on labeled data from one source domain, it may not be able to generalize to new target domains (Saenko et al., 2010). To remedy this problem, unsupervised domain adaptation techniques can be employed (Ganin et al., 2016). It has been shown (see, .e.g, Motiian et al., 2017) that incorporating a small amount of labeled data from the target distribution can significantly improve the performance of the classifier.

Several previous works study differentially private domain adaptation. Wang et al. (2020) propose a method for privately adapting deep learning models to new domains using DP. Peterson et al. (2019) study private domain adaptation in the setting of federated learning in which individual private data comes from diverse domains and the goal is to learn a single private shared model. Bassily et al. (2022) studied domain adaptation with a private unlabeled target dataset and a non-private labeled source dataset. This is the mirror image of our setting, in which the source is private and the target is labeled. In addition, there is no attempt to select centers in that work. We are not aware of previous work in which the source dataset is private and needs to be used to select target examples to label.

## 3 Setting and notation

For an integer m, we define $[m] = \{1, \ldots, m\}$. Assume an example domain $\mathcal{X}$, equipped with a distance function on $\mathcal{X}$ denoted $\Delta : \mathcal{X} \times \mathcal{X} \to \mathbb{R}_+$. For an example $x \in \mathcal{X}$ and a (finite) set $S \subseteq \mathcal{X}$, define $\Delta(x, S) := \min_{s \in S} \Delta(x, s)$. Let $\mathcal{S} \in \mathcal{X}^m$ be the private source dataset, and let $\mathcal{T} \in \mathcal{X}^n$ be the target (non-private) dataset. Let $T_k$ denote the set of $k$ selected centers from $\mathcal{T}$. The cost of the solution $T_k$ is the sum of distances between each point in $\mathcal{T}$ and its closest center in $\mathcal{S} \cup T_k$. Formally, we define

$$\mathbf{Cost}(\mathcal{T}, \mathcal{S}, T_k) := \sum_{x \in \mathcal{T}} \Delta(x, \mathcal{S} \cup T_k)/n.$$

The goal in private STC is to select a set of $k$ points $T_k \in \mathcal{T}^k$ that minimizes this objective function, while preserving the privacy of $\mathcal{S}$. In this work, we study this problem assuming $\mathcal{X} \subseteq \mathbb{R}^d$ for some integer $d$ and taking $\Delta$ to be the Euclidean distance.

We define privacy preservation following the Differential Privacy (DP) framework of Dwork et al. (2006c), which we now recall. Given two databases $X, X'$ of size $n$ from the example domain $\mathcal{X}$, they are considered *neighbors* under DP if one of them can be obtained from the other by adding or removing a single element. DP is then defined as follows:

**Definition 3.1** (Dwork et al. (2006c))**.** *Let $\epsilon, \delta \geq 0$. Let $\mathcal{Y}$ be an output domain. A randomized algorithm $\mathcal{A} \colon \mathcal{X}^* \to \mathcal{Y}$ is $(\varepsilon, \delta)$-differentially private $((\epsilon, \delta)$-DP) if for every pair of neighboring databases $X, X'$ and every output subset $Y \subseteq \mathcal{Y}$,*

$$\Pr[\mathcal{A}(X) \in Y] \leq e^\varepsilon \cdot \Pr[\mathcal{A}(X') \in Y] + \delta,$$

*where the probability is over the randomization of $\mathcal{A}$. If $\delta > 0$, we say that $\mathcal{A}$ satisfies* approximate differential privacy*. If $\delta = 0$, we say that $\mathcal{A}$ satisfies* pure differential privacy*, and that it has $\varepsilon$-differential privacy ($\varepsilon$-DP).*

We further study an important recently proposed variant of Differential Privacy, *Zero Concentrated Differential Privacy* (zCDP) (Bun & Steinke, 2016). This formulation offers smoother composition properties than standard $(\varepsilon, \delta)$-DP. The general idea is to compare the Rényi divergence of the privacy losses random variables for neighboring databases.

**Definition 3.2** (Zero-Concentrated Differential Privacy (zCDP) Bun & Steinke (2016))**.** *An algorithm $\mathcal{A} \colon \mathcal{X}^* \to \mathbb{R}$ is $\rho$-zCDP if for all neighbouring datasets $X, X'$ and $\alpha \in (1, \infty)$ $\mathrm{RD}_\alpha(\mathcal{A}(Z), \mathcal{A}(Z')) \leq \rho\alpha$, where $\mathrm{RD}_\alpha := \frac{1}{1-\alpha} \log(\int_\omega P(x)^\alpha Q(x)^{1-\alpha}\, dx)$ is the $\alpha$-Rényi divergence between random variables $A$ and $B$.*

In the non-private setting, selecting a set of $k$ points $T_k \in \mathcal{T}^k$ that minimizes the cost function defined above is equivalent to solving a $k$-medoids problem with a suitable distance function, a problem which is well-studied (see, e.g., Kaufman & Rousseeuw, 2009; Ng & Han, 2002; Park & Jun, 2009). However, this reduction does not preserve the privacy of $\mathcal{S}$. In the next section, we discuss the hardness of private STC.

## 4   Lower bounds

In this section, we show that private STC can have a high sensitivity compared to standard DP private clustering, a possible obstacle to DP algorithms. We further provide a lower bound on the **Cost** obtainable by any $(\varepsilon, \delta)$-DP algorithm in this setting. From these hardness results, we conclude that any success of a useful algorithm for this setting must be data-dependent.

The property of $L_1$-*Sensitivity* (Dwork et al., 2006c) is crucial for providing guarantees for DP algorithms. A function $f$ mapping databases to $\mathbb{R}^w$ ($w \in \mathbb{N}$) has an $L_1$-sensitivity of $\lambda$ if $\|f(X) - f(X')\|_1 \leq \lambda$ for all pairs $X \in \mathcal{X}^m, X' \in \mathcal{X}^{m-1}$ of neighboring datasets. The $L_1$-sensitivity determines the amount of noise that needs to be added to the function to ensure differential privacy (Dwork et al., 2006c; Wang & Chang, 2018). A large sensitivity thus implies that the accuracy of the output could deteriorate significantly in a private setting, unless additional measures are taken. In standard DP clustering (Blum et al., 2005), the entire dataset is private and the goal is to privately select $k$ centers from the domain. The $L_1$-sensitivity of the cost function in this case is $2/n$ (assuming the domain is a subset of the unit sphere), where $n$ is the size of the dataset. In contrast, the following observation shows that the sensitivity of the cost function in private STC is constant, and does not become smaller for large datasets.

**Observation 4.1** ($L_1$-sensitivity of **Cost**)**.** *Let $\mathcal{X}$ be the $d$-dimensional unit sphere: $\mathcal{X} = \{x \in \mathbb{R}^d : \|x\|_2 \leq 1\}$. For any $k \leq \min\{(2\sqrt{d})^d - 1, n/2 - 1\}$, there exists a target dataset $\mathcal{T} \in \mathcal{X}^n$ and a centroid selection $T_k \in \mathcal{X}^k$ such that the $L_1$-sensitivity of the function $X \mapsto \mathbf{Cost}(\mathcal{T}, X, T_k)$ is at least $1/4$.*

*Proof of Observation 4.1.* Since $k + 1 \leq (2\sqrt{d})^d$, there exists a set of $k + 1$ points such that each pair of them is at least a distance of $1/2$ apart (see, e.g., Conway & Sloane, 2013). Let $x_1, \ldots, x_{k+1} \in \mathcal{X}$ be such that $\Delta(x_i, x_j) \geq 1/2$ for all $i \neq j$. Let $\mathcal{T} \in \mathcal{X}^n$ be a dataset that contains $x_2, \ldots, x_{k+1}$ and $n - k$ copies of $x_1$. Let

$\mathcal{S}_1 \in \mathcal{X}^m$ be a dataset that contains $x_1$ and $m-1$ copies of $x_2$. Let $\mathcal{S}_2 \in \mathcal{X}^{m-1}$ be a dataset of size $m-1$, where all points are copies of $x_2$. Note that $\mathcal{S}_1$ and $\mathcal{S}_2$ are neighbors. For $T_k = \{x_2, \dots x_{k+1}\}$ we have:

$$|\mathbf{Cost}(\mathcal{T}, \mathcal{S}_1, T_k) - \mathbf{Cost}(\mathcal{T}, \mathcal{S}_2, T_k)| = |0 - \frac{1}{n} \sum_{x \in T} \Delta(x, \mathcal{S}_2 \cup \mathcal{T})| \geq \frac{1}{2} \cdot \frac{n-k}{n} \geq \frac{1}{4}.$$

This proves the claim. $\square$

Next, we observe that any $(\varepsilon, \delta)$-DP algorithm for private STC must incur, with a non-negligible probability, an additive error of $\Omega(1/k)$ relative to the optimal achievable cost using $k$ centers from $\mathcal{T}$.

**Observation 4.2.** *Let $\mathcal{X}$ be the $d$-dimensional unit sphere $\mathcal{X} = \{x \in \mathbb{R}^d : \|x\|_d \leq 1\}$. Let $k \leq (2\sqrt{d})^d - 2$. For any algorithm $\mathcal{A} : \mathcal{X}^* \times \mathcal{X}^* \rightarrow \mathcal{T}^k$ which is $(\varepsilon, \delta)$-DP with respect to its second argument, there exist a target dataset $\mathcal{T} \in \mathcal{X}^n$ and a source dataset $\mathcal{S} \in \mathcal{X}^m$ such that there is a probability of at least $\frac{1-2\delta}{2e^\varepsilon}$ that the output $T_k = A(\mathcal{T}, \mathcal{S})$ satisfies*

$$\mathbf{Cost}(\mathcal{T}, \mathcal{S}, T_k) \geq \inf_{\tilde{T}_k \in \mathcal{T}^k} \mathbf{Cost}(\mathcal{T}, \mathcal{S}, \tilde{T}_k) + 1/(2k+4). \tag{1}$$

*where the probability is over the randomness of $\mathcal{A}$.*

The proof is provided in Appendix A. To understand the implications of this observation, note that in DP the values of $\delta, \epsilon$ may be very small. If, for instance, $\delta, \epsilon \leq 0.1$, Eq. (1) would hold with probability larger than $1/3$. This means that no algorithm for private STC can guarantee an additive error of less than $1/(2k+4)$ over all datasets. Nonetheless, as we show in Theorem 5.5 and Theorem 5.10 below, by avoiding worst-case analysis and instead deriving a *data-dependent* guarantee, which factors in the properties of the dataset, it is possible to obtain an additive error that does not depend on $k$, and can be significantly smaller.

## 5 Algorithms

In this section, we present two algorithms for private STC. Section 5.1 presents an approximate DP algorithm with theoretical guarantees, including a data-dependent cost lower and upper bound which holds with high probability. Section 5.2 presents a practical pure differential privacy algorithm, which is suitable for privacy parameter values commonly used in practice, requiring considerably less added noise. In Section 5.3, we analyse these algorithms also under the zCDP formulation, showing that under this definition the added noise can be reduced even more.

All of the algorithms below use $\mathcal{T}$ and $\mathcal{S}$ to calculate a new privacy-preserving set $\mathcal{S}'$, which does not violate the privacy of $\mathcal{S}$. $\mathcal{S}'$ can then be used as input to a (non private) STC algorithm (e.g., de Mathelin et al., 2022) in which $\mathcal{T}$ is the target and $\mathcal{S}'$ is the source dataset. To evaluate the quality of the algorithms' output, it suffices to upper bound the difference in cost between solving (non-private) STC with respect to $\mathcal{S}'$ and solving it with respect to $\mathcal{S}$. Denote

$$\mathbf{Diff}(\mathcal{S}, \mathcal{S}') := \max_{T_k \in \mathcal{T}^k} |\mathbf{Cost}(\mathcal{T}, \mathcal{S}, T_k) - \mathbf{Cost}(\mathcal{T}, \mathcal{S}', T_k)|.$$

If $\mathbf{Diff}(\mathcal{S}, \mathcal{S}')$ is small, then solving (non-private) STC with respect to (non-private) $\mathcal{S}'$ is almost equivalent to solving it with respect to (private) $\mathcal{S}$, up to an additive error of $\mathbf{Diff}(\mathcal{S}, \mathcal{S}')$. Thus, we provide guarantees for the algorithms below by upper bounding this quantity.

### 5.1 An Approximate Differentially Private Algorithm

We propose an $(\varepsilon, \delta)$-DP algorithm for private STC, named Noisy Average Set (NAS). We assume for simplicity that for any $x, x' \in \mathcal{X}$, $\Delta(x, x') \equiv \|x - x'\|_2 \leq 1$. We start by considering a naive approach for using the private dataset $\mathcal{S}$. For $x \in \mathcal{X}$, denote the nearest neighbor of $x$ in $\mathcal{S}$ by $s_x := s_x^1 := \mathrm{argmin}_{s \in \mathcal{S}} \Delta(x, s)$. Denote the set of the nearest neighbors in $\mathcal{S}$ of the points in $\mathcal{T}$ by $\mathcal{S}^* := \{s_x \mid x \in \mathcal{T}\}$. It suffices to consider the points in $\mathcal{S}^*$ when selecting the best set of centroids in $\mathcal{T}$. This is made formal in the following observation.

---

**Algorithm 1** Noisy Average Set (NAS)

---

**input** Private source dataset $\mathcal{S} \in \mathcal{X}^m$, target dataset $\mathcal{T} \in \mathcal{X}^n$, privacy parameters $\varepsilon$ and $\delta$,
  configuration parameter $t \in [m]$, a map $\mathcal{M} : \mathbb{R}^d \to \mathbb{R}^d$.
1: **for** $x \in \mathcal{T}$ **do** $c_x \leftarrow \frac{1}{t} \sum_{i=1}^{t} s_x^i$ and $s_x' \leftarrow \mathcal{M}(c_x)$.
2: $\mathcal{S}' \leftarrow \{s_x' \mid x \in \mathcal{T}\}$
3: **return** $\mathcal{S}'$

---

**Observation 5.1.** *For any $T_k \in \mathcal{T}^k$, $\mathbf{Cost}(\mathcal{T}, \mathcal{S}, T_k) = \mathbf{Cost}(\mathcal{T}, \mathcal{S}^*, T_k)$.*

The proof is provided in Appendix A. Motivated by this simple observation, we aim to design a private algorithm that computes a "sanitized" version of $\mathcal{S}$, called $\mathcal{S}'$, hopefully containing points that are close to the points in $\mathcal{S}^*$. Then, we could solve the problem non-privately using $\mathcal{S}'$ instead of $\mathcal{S}$. Clearly, due to the DP requirement, we cannot simply copy points from $\mathcal{S}^*$ to $\mathcal{S}'$. To overcome this, we consider the set of $t$ points closest to $x$ in $\mathcal{S}$, where $t \in [m]$ is a parameter, and average them in a privacy-preserving manner. NAS, listed in Alg. 1, accepts as input the private source dataset $\mathcal{S}$, the target dataset $\mathcal{T}$, the DP parameters $\epsilon, \delta$, and a configuration parameter $t \in [m]$. In addition, it uses a provided map $\mathcal{M} : \mathbb{R}^d \to \mathbb{R}^d$, which needs to have appropriate DP properties, as we discuss below. For every $x \in \mathcal{T}$, Alg. 1 calculates the average of the $t$ points in $\mathcal{S}$ closest to $x$, where the $i$'th nearest neighbor of $x$ in $\mathcal{S}$ is denoted $s_x^i := \text{argmin}_{s \in \mathcal{S} \setminus \{s_x^1, \dots, s_x^{i-1}\}} \Delta(x, s)$. It then transforms the obtained average $c_x$ using the map $\mathcal{M}$. Lastly, it returns a privacy-preserving output dataset $\mathcal{S}' \in \mathcal{X}^n$, which contains all transformed averages.

To obtain a DP guarantee for NAS, we set $\mathcal{M}$ to follow the well-known differentially private Gaussian mechanism (Dwork et al., 2006b), defined as follows. For a given function $f : \mathcal{X}^t \to \mathbb{R}^d$ with $L_2$-sensitivity $\lambda$, the Gaussian mechanism gets a tuple of points $X$ and outputs $g(X) := f(X) + Z$, where $Z \sim \mathcal{N}(0, \sigma^2 I)$ is a $d$-dimensional vector of independent $\mathcal{N}(0, \sigma^2)$ random variables.

The following guarantee is known for the Gaussian mechanism:

**Lemma 5.2** (Gaussian Mechanism, Dwork et al., 2006b). *Let $\varepsilon, \delta \in (0, 1)$, and assume $f : \mathcal{X}^t \to \mathbb{R}^d$ has $L_2$-sensitivity $\lambda$. Let $\sigma \geq \frac{\lambda}{\varepsilon} \sqrt{2 \ln(1.25/\delta)}$. The mechanism that on input $X \in \mathcal{X}^t$ outputs $g(X) = f(X) + \mathcal{N}(0, \sigma^2 I)$ is $(\varepsilon, \delta)$-DP.*

In our case, $f$ is the average function, and we set $\mathcal{M} : \mathbb{R}^d \to \mathbb{R}^d$ to add the necessary Gaussian noise. Define

$$\sigma_{\text{NAS}} := \frac{1}{t\varepsilon} \sqrt{18n \log(1/\delta) \cdot \log\left(1.25(n+1)/\delta\right)}.$$

We provide the following privacy guarantee for NAS.

**Theorem 5.3.** *If Alg. 1 runs with $\mathcal{M}$ set as the Gaussian mechanism with $\sigma = \sigma_{\text{NAS}}$ above and $\epsilon \leq 1/\sqrt{n}$, then Alg. 1 is $(\varepsilon, \delta)$-DP.*

To prove Theorem 5.3, we use the advanced composition property (Dwork et al., 2010a) to aggregate the overall privacy of $n$ application of the subroutine $\mathcal{M}$. This property states that for all $\varepsilon, \delta \geq 0$ and $\delta' > 0$, the adaptive composition of $L$ algorithms, each of which is $(\varepsilon, \delta)$-DP, is $(\tilde{\varepsilon}, \tilde{\delta})$-DP, where:

$$\tilde{\varepsilon} = \varepsilon \sqrt{2L \ln\left(1/\delta'\right)} + L\varepsilon \frac{e^\varepsilon - 1}{e^\varepsilon + 1} \quad \text{and} \quad \tilde{\delta} = L\delta + \delta'.$$

Taking $\delta' := \delta$ and $\epsilon \leq 1/\sqrt{L}$, we get that $\tilde{\varepsilon} \leq 3\varepsilon \sqrt{L \ln\left(1/\delta\right)}$ and $\tilde{\delta} = (L+1)\delta$.

*Proof of Theorem 5.3.* From our assumption that $\Delta(x, x') \leq 1$, the $L_2$-Sensitivity of the average function is at most $\frac{1}{t}$. By Lemma 5.2, each activation of $\mathcal{M}$ is $(\varepsilon/\sqrt{9n \ln(1/\delta)}, \delta/(n+1))$-DP. Alg. 1 applies the map $\mathcal{M}$ for each $x \in \mathcal{T}$, giving $n$ applications. By the advanced composition property, Alg. 1 is $(\varepsilon, \delta)$-DP. $\square$

Having established the DP property of Alg. 1, we now provide a data-dependent clustering cost upper bound for its outcome. The bound depends on the data-dependent quantity $\Delta^t(\mathcal{S})$, which we define below.

**Definition 5.4.** *Denote the average distance between $x \in \mathcal{T}$ and $s_x^1, \ldots, s_x^t$ by $\Delta^t(x, \mathcal{S}) = \sum_{i=1}^t \Delta(x, s_x^i)/t$, and let $\Delta^t(\mathcal{S}) = \sum_{x \in \mathcal{T}} \Delta^t(x, \mathcal{S})/n$.*

$\Delta^t(\mathcal{S})$ can be privately calculated using the Gaussian mechanism with $\sigma = 1/(t\varepsilon)$. We now use $\Delta^t(\mathcal{S})$ to upper bound $\mathbf{Diff}(\mathcal{S}, \mathcal{S}')$, thus bounding the additive error that results from using $\mathcal{S}'$ instead of $\mathcal{S}$.

**Theorem 5.5.** *Suppose that Alg. 1 is run with $\mathcal{M}$ set as above and $\sigma = \sigma_{\mathrm{NAS}}$. Define*

$$\Lambda(d, n, \delta, \gamma) := \frac{6}{t\varepsilon} \sqrt{dn \log\left(\frac{1}{\delta}\right) \log\left(\frac{1.25(n+1)}{\delta}\right) \log\left(\frac{2dn}{\gamma}\right)}.$$

*Then for any $\gamma \in (0, 1)$ Alg. 1 outputs $\mathcal{S}'$ such that with probability $1 - \gamma$,*

$$\mathbf{Diff}(\mathcal{S}, \mathcal{S}') \leq \Delta^t(\mathcal{S}) + \Lambda(d, n, \delta, \gamma).$$

To prove this theorem, we first provide the following lemma, which upper bounds with high probability the absolute difference between $\Delta(s_x', x)$ and $\Delta(s_x, x)$.

**Lemma 5.6.** *Suppose that Alg. 1 is run with $\mathcal{M}$ as defined above and $\sigma = \sigma_{\mathrm{NAS}}$. Then Alg. 1 outputs $\mathcal{S}'$ such that, with probability at least $1 - \gamma$, for each $x \in \mathcal{T}$,*

$$|\Delta(x, s_x') - \Delta(x, s_x)| \leq \Delta^t(x, \mathcal{S}) + \Lambda(d, n, \delta, \gamma).$$

*Proof.* We consider two cases. In the first case, $\Delta(x, s_x') \leq \Delta(x, s_x)$. Then

$$\Delta(x, s_x) - \Delta(x, s_x') \leq \Delta(x, s_x) \leq \frac{1}{t} \sum_{i=1}^t \Delta(x, s_x) \leq \frac{1}{t} \sum_{i=1}^t \Delta(x, s_x^i) = \Delta^t(x, \mathcal{S}).$$

Thus, in this case the bound holds.

In the second case, $\Delta(x, s_x') \geq \Delta(x, s_x)$. Then, letting $Z \sim \mathcal{N}(0, \sigma_{\mathrm{NAS}}^2 I)$, we have

$$\begin{aligned}
\Delta(x, s_x') - \Delta(x, s_x) &\leq \Delta(x, s_x') \\
&= \Delta(x, c_x + Z) \\
&\leq \Delta(x, c_x) + \Delta(c_x, c_x + Z) \\
&\leq \Delta(x, c_x) + \|Z\|_2 \\
&\leq \frac{1}{t} \sum_{i=1}^t \Delta(x, s_x^i) + \|Z\|_2 \\
&= \Delta^t(x, \mathcal{S}) + \|Z\|_2 \leq \Delta^t(x, \mathcal{S}) + \sqrt{d}\|Z\|_\infty.
\end{aligned}$$

Since $Z \sim \mathcal{N}(0, \sigma_{\mathrm{NAS}}^2 I)$, for any $i \leq d$ and $t > 0$, $\Pr(|Z_i| > t) \leq 2\exp\left(-\frac{t^2}{2\sigma_{\mathrm{NAS}}^2}\right)$. By the union bound, $\Pr(\max_{i>d} |Z_i| > t) \leq 2d\exp\left(-\frac{t^2}{2\sigma_{\mathrm{NAS}}^2}\right)$. Setting $t = \sigma_{\mathrm{NAS}}\sqrt{2\log\left(\frac{2d}{\gamma}\right)}$, we get

$$\Pr\left(\|Z\|_\infty > \sigma_{\mathrm{NAS}}\sqrt{2\log(2d/\gamma)}\right) \leq 2d\exp\left(-\log(2d/\gamma)\right) = \gamma.$$

Therefore, by the definition of $\sigma_{\mathrm{NAS}}$, and taking a union bound over the $n$ vectors $x \in \mathcal{T}$, with probability $1 - \gamma$,

$$\|Z\|_\infty \leq \frac{6}{t\varepsilon}\sqrt{n\log(1/\delta)\log(1.25(n+1)/\delta)\log(2dn/\gamma)} = \Lambda(d, n, \delta, \gamma).$$

Substituting this in the upper bound above gives the statement of the lemma. $\qquad\square$

Next, define $\alpha_x := \operatorname{argmin}_{x' \in \mathcal{S} \cup T_k} \Delta(x, x')$ and $\alpha_x' := \operatorname{argmin}_{x' \in \mathcal{S}' \cup T_k} \Delta(x, x')$. The following lemma uses the lemma above to upper bound the difference between distances to these points.

**Lemma 5.7.** *Suppose that Alg. 1 is run with $\mathcal{M}$ set as above and $\sigma = \sigma_{\mathrm{NAS}}$. Then Alg. 1 outputs $\mathcal{S}'$ such that with probability $1 - \gamma$, for any $T_k \in \mathcal{T}^k$,*

$$|\Delta(x, \alpha'_x) - \Delta(x, \alpha_x)| \leq \Delta^t(x, \mathcal{S}) + \Lambda(d, n, \delta, \gamma).$$

*Proof.* Suppose that the bound of Lemma 5.6 holds, and fix some $T_k \in \mathcal{T}^k$. We consider four cases and show that the bound holds in each.

1. If $\alpha'_x, \alpha_x \in T_k$ then $\alpha_x = \alpha'_x$, hence $|\Delta(x, \alpha'_x) - \Delta(x, \alpha_x)| = 0$.

2. If $\alpha'_x \in \mathcal{S}', \alpha_x \in \mathcal{S}$, then by Lemma 5.6,

$$|\Delta(x, \alpha'_x) - \Delta(x, \alpha_x)| = |\Delta(x, s'_x) - \Delta(x, s_x)| \leq \Lambda(d, n, \delta, \gamma).$$

3. If $\alpha'_x \in \mathcal{S}', \alpha_x \in T_k$, then $\Delta(x, \alpha'_x) \leq \Delta(x, \alpha_x) \leq \Delta(x, s_x)$. Therefore,

$$|\Delta(x, \alpha'_x) - \Delta(x, \alpha_x)| \leq \Delta(x, \alpha_x) \leq \Delta(x, s_x) \leq \Delta^t(x, \mathcal{S}).$$

4. $\alpha'_x \in T_k, \alpha_x \in \mathcal{S}$:

$$\Delta(x, \alpha_x) = \Delta(x, s_x) \leq \Delta(x, \alpha'_x) \leq \Delta(x, s'_x).$$

Combine with Lemma 5.6 and we get:

$$|\Delta(x, \alpha'_x) - \Delta(x, \alpha_x)| = \Delta(x, \alpha'_x) - \Delta(x, \alpha_x) \leq \Delta(x, s'_x) - \Delta(x, s_x) \leq \Lambda(d, n, \delta, \gamma).$$

This completes the proof. $\square$

To prove Theorem 5.5, observe that by the triangle inequality, $|\mathbf{Cost}(\mathcal{T}, \mathcal{S}, T_k) - \mathbf{Cost}(\mathcal{T}, \mathcal{S}', T_k)| \leq \frac{1}{n} \sum_{x \in \mathcal{T}} |\Delta(x, \alpha'_x) - \Delta(x, \alpha_x)|$. Thus, by Lemma 5.7:

$$|\mathbf{Cost}(\mathcal{T}, \mathcal{S}, T_k) - \mathbf{Cost}(\mathcal{T}, \mathcal{S}', T_k)| \leq \frac{1}{n} \sum_{x \in \mathcal{T}} \Delta^t(x, \mathcal{S}) + \Lambda(d, n, \delta, \gamma) = \Delta^t(\mathcal{S}) + \Lambda(d, n, \delta, \gamma).$$

This completes the proof of the theorem. To show that a dependence on $\Delta^t(\mathcal{S})/k$ is necessary, we provide the following observation, which is a data-dependent variant of Observation 4.2.

**Observation 5.8.** *Under the same assumptions of Observation 4.2, except that $k \leq (\sqrt{d}/\Delta^t(\mathcal{S}))^d - 2$, it can be shown that with a probability at least $\frac{1-2\delta}{2e^\varepsilon}$,*

$$\mathbf{Cost}(\mathcal{T}, \mathcal{S}, T_k) \geq \inf_{\tilde{T}_k \in \mathcal{T}^k} \mathbf{Cost}(\mathcal{T}, \mathcal{S}, \tilde{T}_k) + \frac{\Delta^t(\mathcal{S})}{k+2}$$

The proof is identical to that of Observation 4.2, except that the example consists of $k + 2$ points that are at least $\Delta^t(\mathcal{S})$ apart, instead of $1/2$ as in the original observation.

## 5.2 Pure Differentially Private Algorithms

Alg. 1 with the Laplace mechanism parameters defined above satisfies approximate DP. However, its privacy guarantees depend on the value of $\delta$. In practice, it is recommended to use small values of $\delta$ to ensure a good level of privacy. This would add an additional constant to the amount of noise injected by the algorithm, which makes it less practical. Therefore, we next introduce two variants of this algorithm which satisfy pure DP. We show in our experiments that these variants work better in practice.

The Gaussian mechanism by definition cannot obtain pure DP. Therefore, we implement the following variants using the Laplace mechanism (Dwork et al., 2006c), defined as follows. We say that a random variable is distributed as $\mathrm{Lap}(b)$ if its probability density function is $h(y) = \exp(-|y|/b)/2b$. For a given function $f : \mathcal{X}^t \to \mathbb{R}^d$ with $L_1$-sensitivity $\lambda$ and for a parameter $b > 0$, the Laplace mechanism gets a tuple of points $X$ and outputs $g(X) := f(X) + \mathrm{Lap}^d(b)$, where $\mathrm{Lap}^d(b)$ is a $d$-dimensional vector of independent $\mathrm{Lap}(b)$ random variables. The following useful properties of the Laplace mechanism have been shown:

---

**Algorithm 2** Neighbor Noisy Averages (NNA)

---

**input** Source data $\mathcal{S} \in \mathcal{X}^m$, target data $\mathcal{T} \in \mathcal{X}^n, \gamma \in (0,1)$, privacy parameter $\varepsilon$,
    a map $\mathcal{M} : \mathbb{R} \times \mathbb{R}^d \to \mathbb{R} \times \mathbb{R}^d$
1:  $\mathcal{S}' \leftarrow \{\}$
2: **for** $x \in \mathcal{T}$ **do**
3:     $h_x \leftarrow \{s \in \mathcal{S} \mid x = \operatorname{argmin}_{x' \in \mathcal{T}} \Delta(x', s)\}$
4:     $n_x, r_x \leftarrow |h_x|, \sum_{s \in h_x} s$
5:     $\hat{n}_x, \hat{r}_x \leftarrow \mathcal{M}(n_x, r_x)$
6:     **if** $\hat{n}_x \geq 1 + \log((\sqrt{d}+1)/\gamma)/\varepsilon$ **then**
7:         $\mathcal{S}' \leftarrow \mathcal{S}' \cup \{\hat{r}_x/\hat{n}_x\}$
8:     **end if**
9: **end for**
10: **return** $\mathcal{S}'$

---

**Lemma 5.9** (Dwork et al. 2006c). *The Laplace mechanism is $\lambda/b$-DP. In addition, for all $X \in \mathcal{X}^d$, $\gamma \in (0,1)$, $\Pr[\|g(X) - f(X)\|_\infty > b \cdot \log(d/\gamma)] \leq \gamma$.*

The first variant is an implementation of Alg. 1 with the Laplace mechanism. This is provided in the following result, adapted from Theorem 5.3 and Theorem 5.5.

**Theorem 5.10** (Alg. 1 Utility with $\varepsilon$-DP). *Suppose we implement $\mathcal{M}$ using the Laplace mechanism with $b = \frac{n\sqrt{d}}{t\varepsilon}$. Then, Alg. 1 is $\varepsilon$-DP and with probability $1-\gamma$ it outputs $\mathcal{S}'$ such that:* $\mathbf{Diff}(\mathcal{S}, \mathcal{S}') \leq \Delta^t(\mathcal{S}) + \frac{nd}{t\varepsilon} \log(\frac{nd}{\gamma})$.

To prove this theorem, we use the basic composition property (Dwork et al., 2006a;c), which states that if $\mathcal{A}_1, \ldots, \mathcal{A}_k$ are algorithms that satisfy $(\varepsilon_1, \delta_1)$-,..., $(\varepsilon_k, \delta_k)$-differential privacy, respectively, then running $\mathcal{A}_1, \ldots, \mathcal{A}_k$ satisfies $\left(\sum_{i=1}^k \varepsilon_i, \sum_{i=1}^k \delta_i\right)$-differential privacy.

*Proof of Theorem 5.10.* Recall that we assume that $\Delta(x, x') \equiv \|x - x'\|_2 \leq 1$ for any two data set points. The $L_1$ change in the average when changing a single point is at most $\|x - x'\|_1/t \leq \sqrt{d}\|x - x'\|_2/t \leq \sqrt{d}/t$. Therefore, by Lemma 5.9, each application of $\mathcal{A}$ is $\varepsilon/n$-DP. Since we have $n$ applications of $\mathcal{M}$, by the basic composition property, Alg. 1 is $\varepsilon$-DP. The cost bound follows analogously to the proof of Theorem 5.5. $\quad\square$

This obtains pure DP. However, the bound is significantly larger than that of the previous variant, due to the dependence on $O(n)$, instead of the $O(\sqrt{n})$ in Theorem 5.5. This comes hand in hand with added noise of size $O(n)$, which can be prohibitively large in practice. We therefore present an additional pure DP algorithm, called Neighbor Noisy Averages (NNA), which builds on the ideas of NAS but is more practical, and adds noise that does not depend on $n$ at all. In addition, it is parameter-free.

NNA accepts the same inputs as NAS except that it also accepts a confidence parameter $\gamma$, and that it requires a map with two arguments. The first argument is used for generating a private count and the second has the same role as the mechanism of NAS. NNA first creates for each $x \in \mathcal{T}$ the set $h_x$ that contains all the points from $\mathcal{S}$ such that $x$ is their nearest point in $\mathcal{T}$. Next, NNA computes the size of $h_x$ and the sum of its points and uses the mechanism $\mathcal{M}$ to generate a private version of those. Then, it adds the private average of the points to the output $\mathcal{S}'$ only if their private number is larger than a threshold. Note that NNA avoids inserting to $\mathcal{S}'$ averages that include no points, that is cases where $|h_x| = 0$. It can be seen from Lemma 5.9 that for the Laplace mechanism used above, with probability $1 - \gamma$, $|\hat{n}_x - n_x| \geq \log((\sqrt{d}+1)/\gamma)/\varepsilon$ for each $x \in \mathcal{T}$. Therefore, if $n_x \geq 1 + \log((\sqrt{d}+1)/\gamma)/\varepsilon$, then with probability $1 - \gamma$, $h_x$ contains at least one point.

Figure 1 illustrates the run of NNA with $d = 2$. The key idea behind NNA is that each data point participates in the calculation of a single center, so that each pair of neighboring datasets has the same center distribution, except for one center. As a result, we can avoid using composition. The following theorem shows that NNA is pure DP.

**Theorem 5.11.** *If Alg. 2 runs with the Laplace mechanism with $b = (\sqrt{d}+1)/\varepsilon$ then it is $\varepsilon$-DP.*

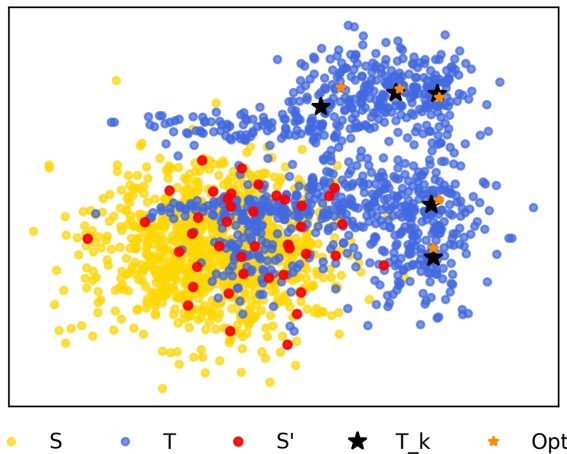

Figure 1: An example of a run of NNA. Colors represent points in the sets as given in the legend.

*Proof.* Let $\mathcal{S} = \hat{\mathcal{S}} \cup \{\hat{s}\}$ and $\hat{\mathcal{S}}$ be neighboring databases. Let $\hat{x} = \text{argmin}_{x' \in \mathcal{T}} \Delta(x', \hat{s})$, and note that $\hat{s}$ belongs to $h_{\hat{x}}$ and to no other $h_x$. Therefore, for each $x \neq \hat{x}$ the distribution of the possible outputs $\hat{r}_x / \hat{n}_x$ is the same under $\mathcal{S}$ and $\hat{\mathcal{S}}$. Since $\forall x, x' \in \mathcal{X}, \Delta(x, x') \leq 1$, the sum function used to calculate $r_x$ has an $L_1$-sensitivity of $\sqrt{d}$, and the counting function has an $L_1$-sensitivity of 1. Therefore, $\mathcal{M}$ has an $L_1$-sensitivity of $\sqrt{d} + 1$. Since $\hat{n}_x, \hat{r}_x$ are obtained from the Laplace mechanism with $b = (\sqrt{d} + 1)/\varepsilon$, by Lemma 5.9 the algorithm satisfies $\varepsilon$-DP. $\square$

Unlike for NAS, we do not have a cost upper bound for NNA. Nonetheless, we show below that in practice, the latter is more successful than the former, perhaps due to the reduction in added noise.

### 5.3 Concentrated Differential Privacy

We now study NAS and NNA under the zCDP formulation of Differential Privacy. It is known that DP implies zCDP (Bun & Steinke, 2016). Nonetheless, by tailoring the noise mechanism parameters, we obtain improved zCDP guarantees compared to a naive transformation, which require significantly less noise than the DP formulation. It is known (Bun & Steinke, 2016) that the Gaussian mechanism is $\frac{\lambda^2}{2\sigma^2}$-zCDP. We first obtain a guarantee for NAS.

**Theorem 5.12** (NAS with zCDP)**.** *Let $\rho > 0$. Suppose we implement $\mathcal{M}$ using the Gaussian mechanism with $\sigma = \frac{1}{t} \sqrt{\frac{n}{2\rho}}$. Then, Alg. 1 is $\rho$-zCDP. Moreover, with probability $1 - \gamma$, for the output $\mathcal{S}'$ of Alg. 1,*

$$\mathbf{Diff}(\mathcal{S}, \mathcal{S}') \leq \Delta^t(\mathcal{S}) + \frac{1}{t} \sqrt{\frac{nd}{\rho} \log\left(\frac{2nd}{\gamma}\right)}.$$

To prove this theorem, we use the zCDP composition theorem (Bun & Steinke, 2016), which states that for $\mathcal{M} : \mathcal{X}^n \to \mathcal{Y}$ and $\mathcal{M}' : \mathcal{X}^n \to \mathcal{Z}$, if $\mathcal{M}$ satisfies $\rho$-zCDP and $\mathcal{M}'$ satisfies $\rho'$-zCDP, then the mechanism $\mathcal{M}'' : \mathcal{X}^n \to \mathcal{Z}$, defined by $\mathcal{M}''(X) = \mathcal{M}'(X, \mathcal{M}(X))$, is $(\rho + \rho')$-zCDP.

*Proof of Theorem 5.12.* By our assumption that $\Delta(x, x') \leq 1$, the $L_2$-Sensitivity of the Gaussian mechanism we use is $1/t$. Therefore, by the properties of the Gaussian Mechanism, each application of $\mathcal{M}$ is $\rho/n$-DP. Since we have $n$ applications of $\mathcal{M}$, by the composition theorem, Alg. 1 is $\rho$-zCDP. The cost bound is proved analogously to the proof of Theorem 5.5. $\square$

For NNA, the noise can be a constant that does not depend on $d$, unlike the standard DP version.

**Theorem 5.13** (NNA with zCDP)**.** *Alg. 2 with the Gaussian mechanism with $\sigma = \sqrt{2/\rho}$ is $\rho$-zCDP.*

*Proof.* By our assumption that $\Delta(x, x') \leq 1$, the $L_2$-Sensitivity of the Gaussian mechanism we use is 2. As in the proof of Theorem 5.11, we can avoid the composition in the applications of $\mathcal{M}$, each of which is $\rho$-zCDP. Therefore, Alg. 1 is $\rho$-zCDP. $\qquad\qquad\square$

Thus, with the zCDP formulation the algorithms require less added noise.

## 6   Experiments

We implemented Alg. 1 in the $\varepsilon$-DP version, as well as Alg. 2. The python code is publicly available on https://github.com/ShacharSchnapp/STC. For Alg. 1, we tested several values of $t$ and got similar results, thus we provide below results for $t = 150$, and report the others in Appendix C.1. As the non-private STC algorithm that is applied to the output $\mathcal{S}'$, we used the state-of-the-art algorithm Accelerated K-medoids (AkM) (de Mathelin et al., 2022).

Since the noise added by Alg. 1 grows with $n$ (the size of $\mathcal{T}$) we used a *coreset* to reduce the amount of noise. A coreset (Har-Peled & Mazumdar, 2004) is a subset of a dataset that approximately preserves the properties of the original dataset. By constructing a coreset $\mathcal{T}'$ such that each point in $\mathcal{T}$ is $C$-close to a point in $\mathcal{T}'$, we can get the same guarantee as in Theorem 5.10 up to an additive term of $C$, but with a smaller dependence on the dataset size (see Appendix B for a formal result). We constructed the coresets using Bentley & Friedman (1979) with a termination condition of $C = 0.05$ (see Appendix B for more on the choice of $C$).

We ran our algorithms on synthetic and real-world datasets, and compared them to the following $\varepsilon$-DP baselines: (1) `ClusterT`: Apply AkM on $\mathcal{T}$ without considering $\mathcal{S}$; (2) `DP-GAN` (Xie et al., 2018): This algorithm privately learns a distribution from source data. We trained `DP-GAN` on the input $\mathcal{S}$, then drew 100 i.i.d. samples from the output distribution, and used them as the source for AkM. We used the implementation and best-practice parameters of Falk & Meneses (2019). (3) `PrivateKmeans` (Balcan et al., 2017b): We privately clustered $\mathcal{S}$ to 100 centers, then used them as input to the AkM algorithm. For the zCDP version of `PrivateKmeans` and `DP-GAN`, we set $\varepsilon = \sqrt{\rho/2}$ to obtain $\rho$-zCDP (see Bun & Steinke, 2016). In all of the experiments, the points were normalized to have a maximal norm of $1/2$. We fixed $\varepsilon = 3$ for DP and $\rho = 3$ for zCDP. For each dataset, algorithm and $k$, we averaged $\mathbf{Cost}(\mathcal{T}, \mathcal{S}, T_k)$ over 30 runs.

We first report results on three simple 2-dimensional synthetic datasets, plotted in Figure 2. These datasets were constructed to highlight the drawbacks of the baseline algorithms, all of which ignore the target data $\mathcal{T}$. In each of these datasets, a smart choice of centers should avoid selecting target centers in an overlapping region. The datasets are generated as follows:

**Synthetic 1** Two 2-dimensional Guassians with different means. The source data is $\mathcal{S} \sim \mathbb{N}((0.15, 0.15), 0.4)$ and the target data is $\mathcal{T} \sim \mathbb{N}((0.95, 0.95), 0.4)$.

**Synthetic 2** $\mathcal{S}$ has 18 clusters on the top and 9 clusters on the bottom right. $\mathcal{T}$ has 9 clusters on the bottom left and 9 clusters on the bottom right. Each cluster is a Gaussian with standard deviation of 0.01.

**Synthetic 3** $\mathcal{S}$ is composed of 6 clusters, with means drawn uniformly at random out of $[0,1] \times [0,1]$. Each cluster is a Gaussian with standard deviation 0.03. $\mathcal{T}$ has 12 clusters which are also Guassians with means drawn uniformly at random. Each cluster is a Gaussian with standard deviation 0.01.

The results for the synthetic datasets for standard DP are provided in Figure 3 (top). The results for zCDP are provided in Appendix C.2. It can be seen `PrivateKmeans` and `DP-GAN` perform comparably or worse than the vanilla `ClusterT` that does not use $\mathcal{S}$ at all. This may be because they cluster the entire source data instead of focusing on the parts relevant to $\mathcal{T}$ like our algorithms. In contrast, our algorithms improve over the vanilla approach and their cost is close to that of the non-private STC algorithm. In Appendix C.3, we present ablation studies on coreset usage, demonstrating cost improvements in many cases and no harmful effects. As expected, the effect on NAS is much more significant than the effect on NNA, since the added

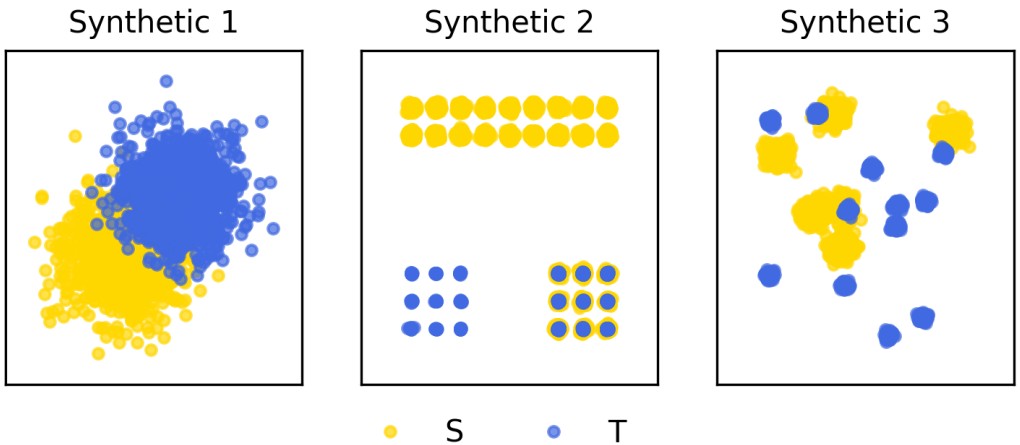

Figure 2: The synthetic datasets

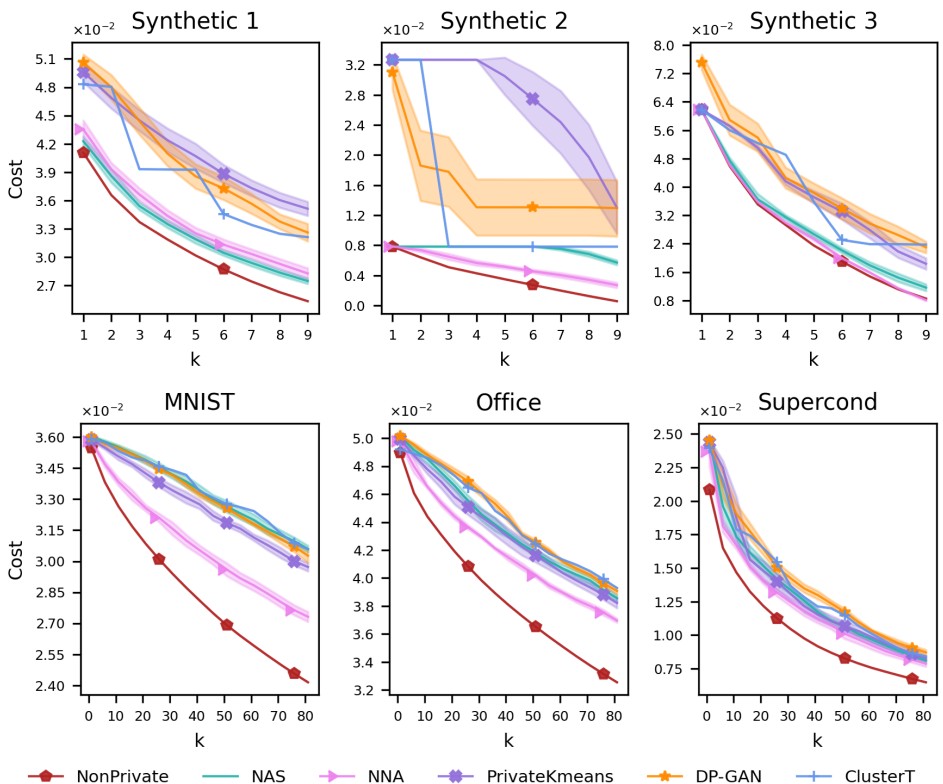

Figure 3: Top: Synthetic DP experiments. Bottom: Real-world DP experiments; MNIST: $6 \mapsto 9$, Office: amazon $\mapsto$ webcam, Superconductivity: high $\mapsto$ middle-high

noise in NAS depends on $n$, unlike in NNA. We also demonstrate, in Appendix **??**, that the results of the baseline algorithms are not improved by using coresets.

Next, we report experiments on real-world datasets. `MNIST` (Deng, 2012) contains 70,000 grayscale images of handwritten digits. We tested three (source,target) pairs of digits: (1,7), (5,2) and (9, 6). `Office` (Saenko et al., 2010) contains images of office items from different sources: "amazon" (2817 samples), "webcam" (795 samples)

and "dslr" (digital single-lens reflex) (480 samples). We tested all (source,target) pairs. `Superconductivity` (Hamidieh, 2018) is an 82-dimensional dataset of 16,000 superconducting materials. Following the domain adaptation setup of Pardoe & Stone (2010), We split the dataset into four subsets termed low (l), middle-low (ml), middle-high (mh) and high (h) of around 4000 instances each. We tested all possible (source, target) pairs.

We applied random dimensionality reduction (Johnson, 1984) with $d = 8$ to all datasets, a privacy-preserving transformation. Fig. 3 (bottom) provides the results of the experiments for DP, for a single (source,target) pair for each dataset. The rest of the pairs, as well as experiments for zCDP, are reported in Sec. C.4 and **??**, respectively. For both privacy formulations, the baselines `PrivateKmeans` and `DP-GAN` perform like the vanilla `ClusterT` that does not use $\mathcal{S}$ at all. In contrast, our algorithms show a significant cost improvement, with the exception of NAS in the standard DP setting. This is expected, as discussed in Sec. 5.2, due to the significant added noise required in this case. We further compare, in Sec. **??**, the run-time of all algorithms that maintain source privacy, showing that all algorithms require similar running times.

## 7 Conclusions

We proposed a new setting of private Source-Target Clustering. We derived lower bounds and data-dependent upper bounds for this objective, proposed two private-STC algorithms, and showed in experiments that they perform significantly better than the baselines. In future work, we hope to study also cases where the target dataset has privacy restrictions.

### Acknowledgements

We acknowledge the support of the Natural Sciences and Engineering Research Council of Canada (NSERC), [funding reference number RGPIN-2024-05907]. Resources used in preparing this research were provided, in part, by the Province of Ontario, the Government of Canada through CIFAR, and companies sponsoring the Vector Institute; see `https://vectorinstitute.ai/partnerships/current-partners/`.

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

## A Deferred proofs

*Proof of Observation 4.2.* Since $k + 2 \leq (2\sqrt{d})^d$, there exists a set of $k + 2$ points such that each pair of them is at least a distance of $1/2$ apart (see Conway & Sloane (2013)). Let $x_1, \ldots, x_{k+2} \in \mathcal{X}$ be such that $\Delta(x_i, x_j) \geq 1/2$ for $i \neq j$. Assume for simplicity that $N = n/(k + 2)$ is an integer. Let $\mathcal{T}$ be a dataset that contains $2N$ copies of $x_1$ and $N$ copies of each of $x_2, \ldots, x_{k+1}$. Let $\mathcal{S}_1 \in \mathcal{X}^m$ be a dataset that contains $x_1$ and $m - 1$ copies of $x_{k+2}$. Let $\mathcal{S}_2 \in \mathcal{X}^{m-1}$ be a dataset that contains only $m - 1$ copies of $x_{k+2}$. Note that $\mathcal{S}_1$ and $\mathcal{S}_2$ are neighbors.

Since $\mathcal{S}_1$ contains $x_1$, the (unique) optimal solution is $T'_k = \{x_2, \ldots, x_{k+1}\}$, which satisfies $\textbf{Cost}(\mathcal{T}, \mathcal{S}_1, T_k) = 0$. Any other solution $T_k \neq T'_k$ satisfies $\textbf{Cost}(\mathcal{T}, \mathcal{S}_1, T'_k) \geq 1/(2k+4)$. Therefore, if $\Pr[A(\mathcal{T}, \mathcal{S}_1) = T'_k] \leq 1 - \frac{1-2\delta}{2e^\varepsilon}$, then $A$ incurs an additive error of at least $1/(2k + 4)$, as claimed. We thus henceforth assume that $\Pr[A(\mathcal{T}, \mathcal{S}_1) = T'_k] > 1 - \frac{1-2\delta}{2e^\varepsilon}$.

Consider now the solution for the private dataset $\mathcal{S}_2$ with the same target dataset $\mathcal{T}$. Since $T'_k \in \mathcal{T}^k$ does not contain $x_1$, we have:

$$\textbf{Cost}(\mathcal{T}, \mathcal{S}_2, T'_k) = \frac{1}{n} \sum_{x \in \mathcal{T}} \Delta(x, \mathcal{S}_2 \cup \mathcal{T})) \geq \frac{2N}{n} \cdot \Delta(x_1, \mathcal{S}_2 \cup \mathcal{T}) = 2/(2k+4).$$

On the other hand, letting $T^*_k = \{x_1, \ldots, x_k\}$, we have:

$$\textbf{Cost}(\mathcal{T}, \mathcal{S}_2, T^*_k) = \frac{1}{n} \sum_{x \in \mathcal{T}} \Delta(x, \mathcal{S}_2 \cup \mathcal{T})) = \frac{N}{n} \cdot \Delta(x_{k+1}, \mathcal{S}_2 \cup \mathcal{T}) = 1/(2k+4).$$

Therefore, whenever $A(\mathcal{T}, \mathcal{S}_2)$ outputs $T'_k$, an additive error of at least $1/(2k + 4)$ is incurred. Since $\mathcal{S}_1, \mathcal{S}_2$ are neighbors, then due to the $(\epsilon, \delta)$-DP of $A$,

$$e^{-\varepsilon}(\Pr[A(\mathcal{T}, \mathcal{S}_1) = T'_k] - \delta) \geq e^{-\varepsilon}(1 - \frac{1-2\delta}{2e^\varepsilon} - \delta) \geq e^{-\varepsilon}(1 - \frac{1}{2e^\varepsilon}) - \frac{2\delta}{2e^\varepsilon} \geq \frac{1-2\delta}{2e^\varepsilon}.$$

This lower bounds the probability of an additive error of $1/(2k + 4)$, as claimed. □

*Proof of Observation 5.1.* For $x \in \mathcal{T}$, let $\alpha_x := \operatorname{argmin}_{x' \in \mathcal{S} \cup T_k} \Delta(x, x')$. We have:

$$\textbf{Cost}(\mathcal{T}, \mathcal{S}, T_k) = \frac{1}{n} \sum_{x \in \mathcal{T}} \Delta(x, \mathcal{S} \cup T_k) = \frac{1}{n} \sum_{x \in \mathcal{T}} \Delta(x, \alpha_x) = \frac{1}{n} \sum_{\substack{x \in \mathcal{T} \\ \alpha_x \in \mathcal{S}}} \Delta(x, \alpha_x) + \frac{1}{n} \sum_{\substack{x \in \mathcal{T} \\ \alpha_x \in T_k}} \Delta(x, \alpha_x).$$

For any $x \in \mathcal{T}$ such that $\alpha_x \in \mathcal{S}$, $\alpha_x = s_x \in \mathcal{S}^*$. Therefore,

$$\textbf{Cost}(\mathcal{T}, \mathcal{S}, T_k) = \frac{1}{n} \sum_{\substack{x \in \mathcal{T} \\ \alpha_x \in \mathcal{S}^*}} \Delta(x, \alpha_x) + \frac{1}{n} \sum_{\substack{x \in \mathcal{T} \\ \alpha_x \in T_k}} \Delta(x, \alpha_x) = \textbf{Cost}(\mathcal{T}, \mathcal{S}^*, T_k),$$

as claimed. □

## B A Guarantee for Coresets

The following is an immediate Corollary of Theorem 5.5.

**Corollary B.1.** *Given a corset $T' \subseteq \mathcal{T}$ for size $n'$ which satisfies $d(x', x) \leq C$ for each $x \in \mathcal{T}, x' \in T'$, every solution $T_k \in \mathcal{T}^k$ satisfies, with probability $1 - \gamma$:*

$$\|\textbf{Cost}(S, T_k) - \textbf{Cost}(S', T_k)\| \leq \Delta^t(\mathcal{S}) + \frac{6}{t\varepsilon} \sqrt{dn' \log\left(\frac{1}{\delta}\right) \log\left(\frac{1.25(n'+1)}{\delta}\right) \log\left(\frac{2dn'}{\gamma}\right)} + C.$$

Note that this corollary shows the natural trade-off between $n'$ and $C$, as a lower value of $C$ would result in a higher value of $n'$ and vice versa. Note that avoiding coresets altogether is equivalent to taking $C = 0, n' = n$. It is possible to optimize this trade-off using private access to the source dataset, but this would require sacrificing some of the privacy budget. In preliminary experiments, we found that it does not improve the results.

## C   Full experiment results

### C.1   Experiments with other values of $t$

Figure 4 and Figure 5 report experiments comparing the performance of NAS over different values of $t$. It can be seen that the results are not very sensitive to the value of $t$.

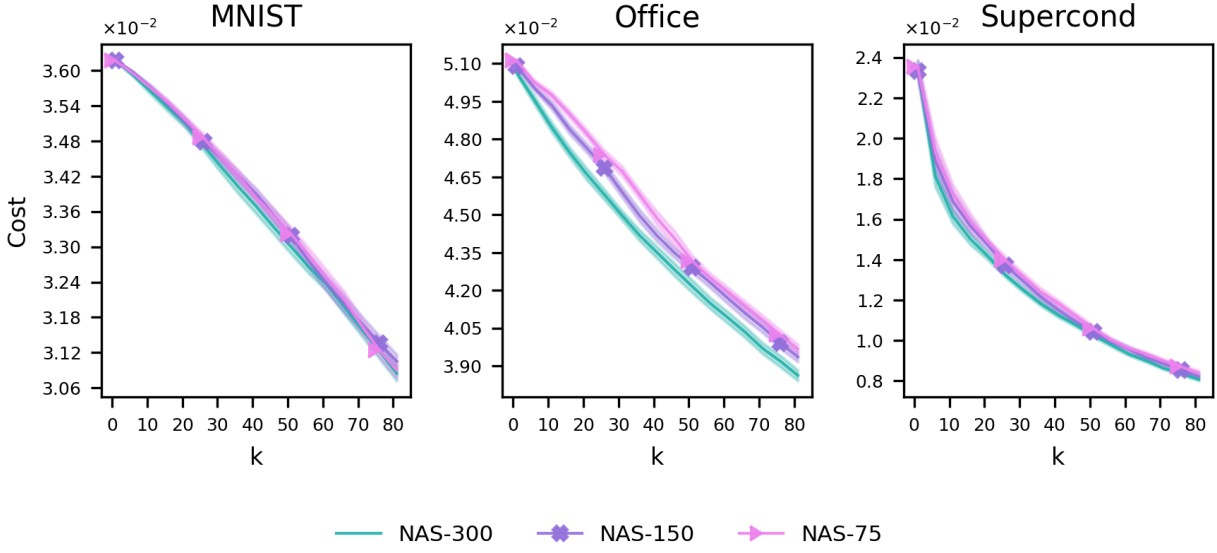

Figure 4: NAS DP real-word cost with $t \in [75, 150, 300]$. MNIST: $6 \mapsto 9$, Office: Amazon $\mapsto$ webcam and Superconductivity: h $\mapsto$ mh

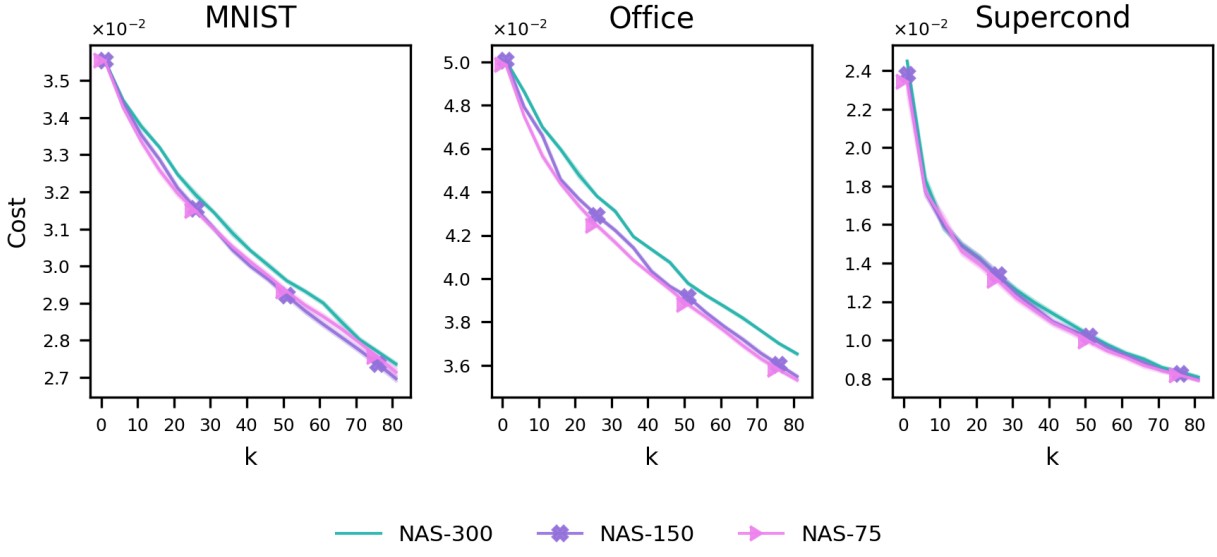

Figure 5: NAS zCDP real-word cost with $t \in [75, 150, 300]$. MNIST: $6 \mapsto 9$, Office: Amazon $\mapsto$ webcam and Superconductivity: h $\mapsto$ mh

## C.2 Experiments for zCDP with synthetic datasets

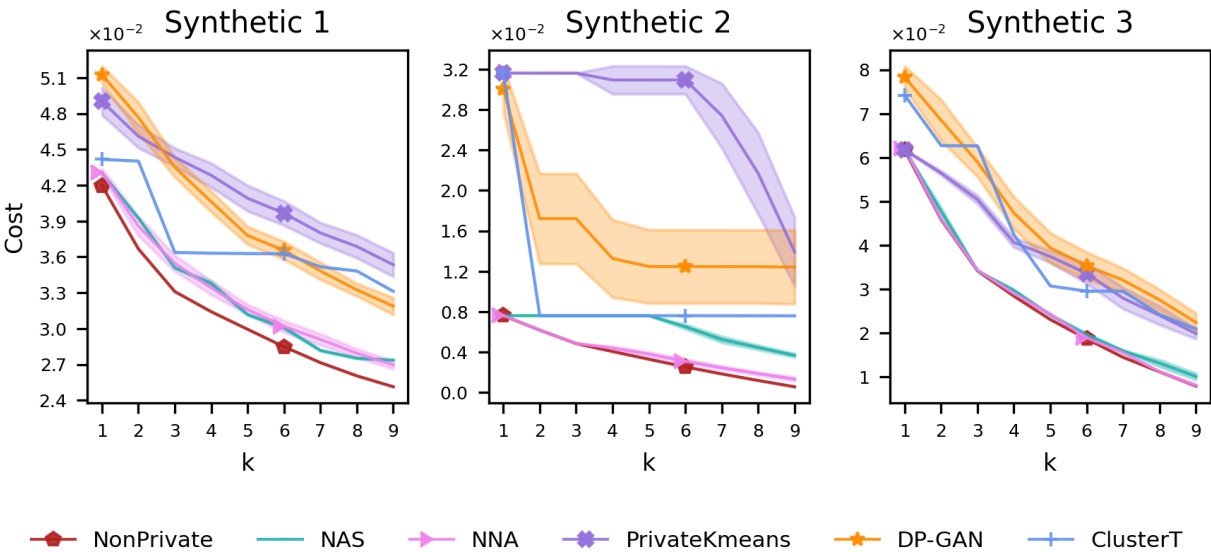

Figure 6: Results of zCDP experiments with synthetic datasets

## C.3 Coreset ablation experiments

In the following experiments, we report a comparison between using coresets and not using them, for the two algorithms NAS and NNA, and for all the synthetic and real-world datasets that we tested. In all the experiments $C$ was set to 0.05. The synthetic experiments (Figure 7) show that coresets can provide significant improvements. In real-world datasets, coresets were helpful in some of the cases (see Figures 23, 24, 25, 26, 27, 30, 34, 35, 36, 47, 50), while in others there was no observable difference. There was no case in which coresets were harmful.

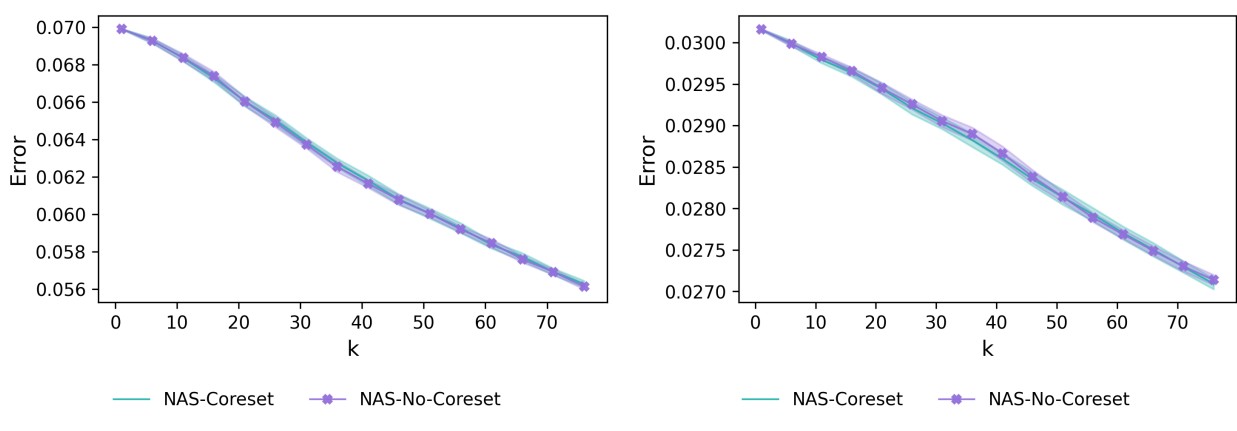

Figure 9: NAS, MNIST: $7 \rightarrow 1$    Figure 10: NAS, MNIST: $7 \rightarrow 1$

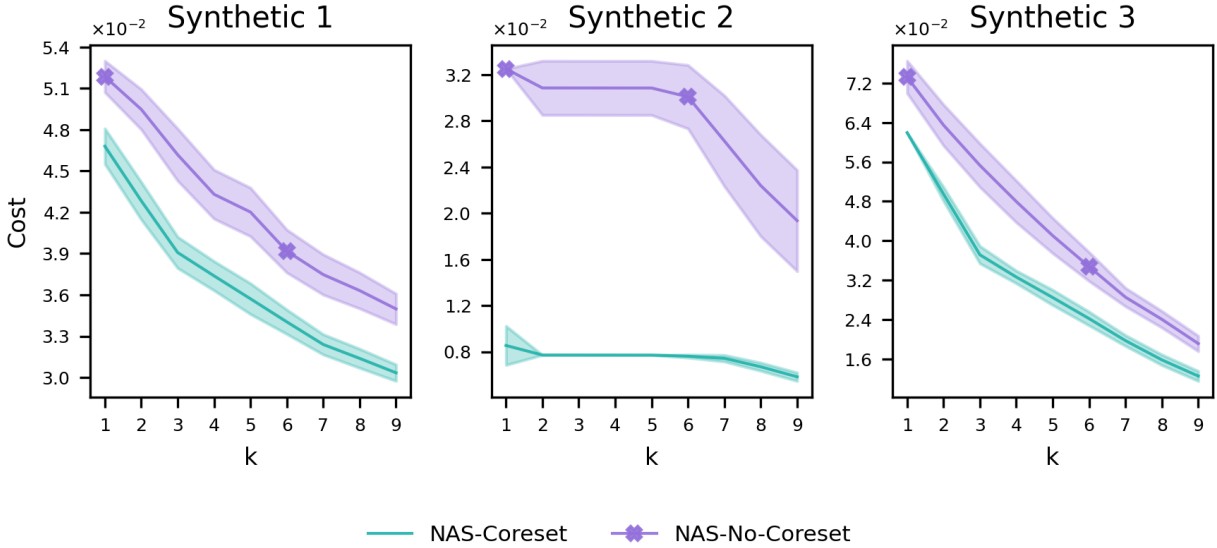

Figure 7: NAS results of ablation studies for using coresets, synthetic datasets

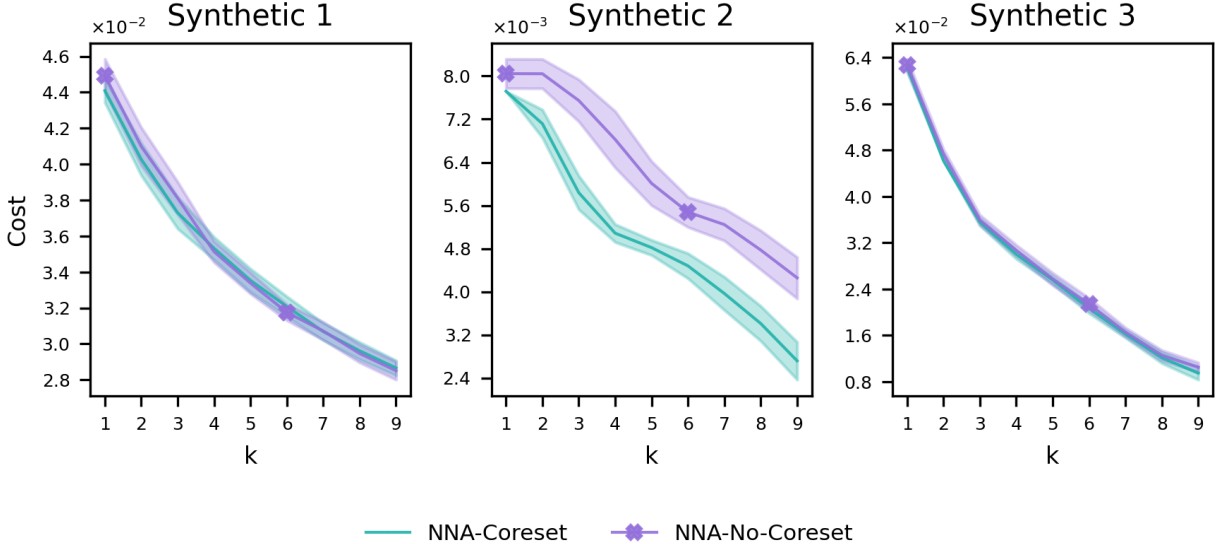

Figure 8: NNA results of ablation studies for using coresets, synthetic datasets

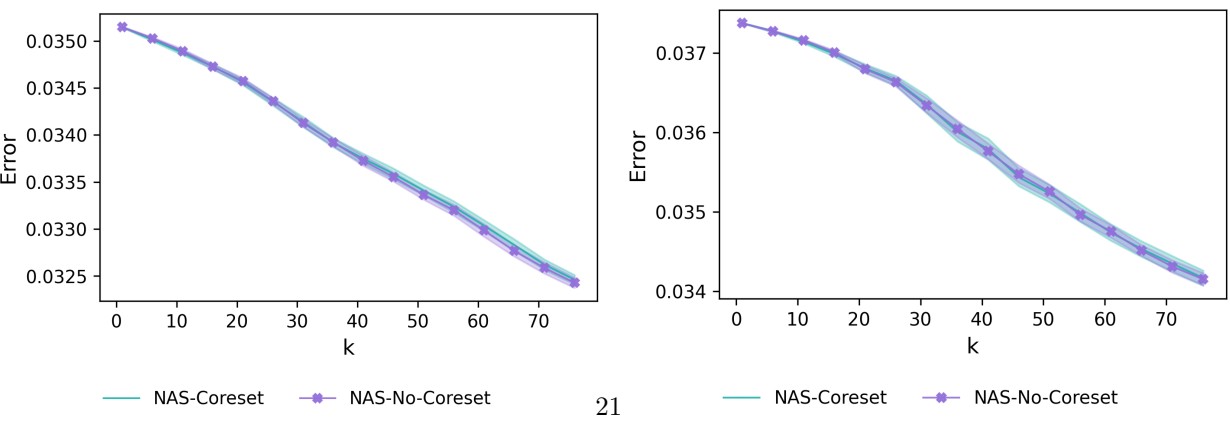

21

Figure 11: NAS, MNIST: $6 \rightarrow 7$                    Figure 12: NAS, MNIST: $9 \rightarrow 6$

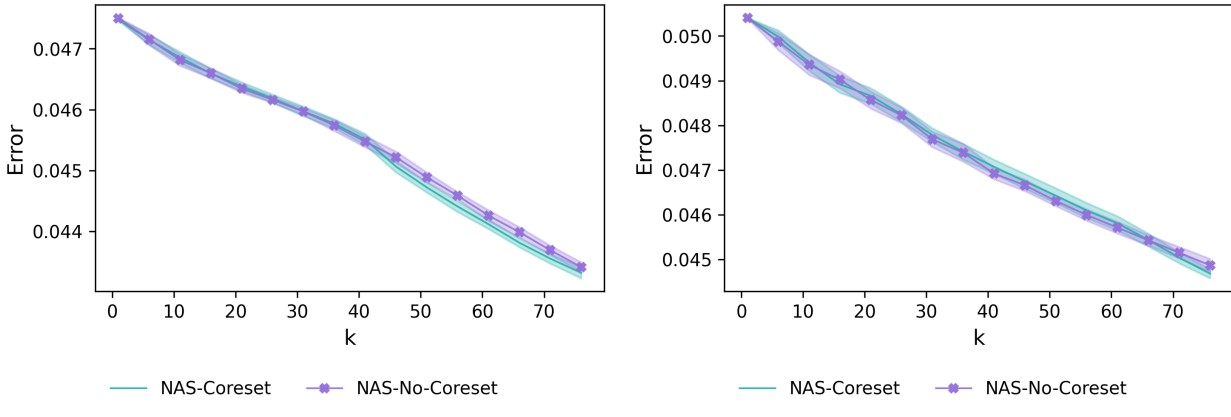

Figure 13: NAS, MNIST: $2 \to 5$

Figure 14: NAS, MNIST: $5 \to 2$

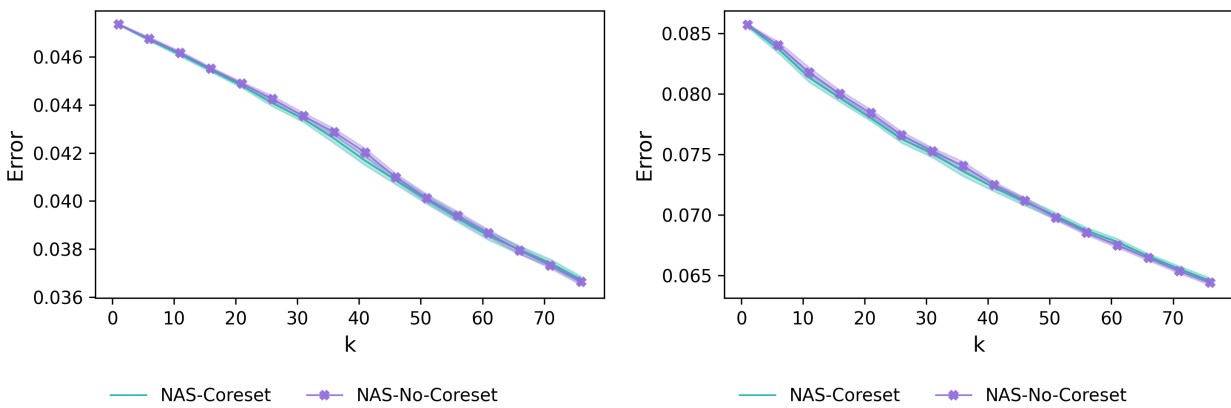

Figure 15: NAS, Office: amazon $\to$ dslr

Figure 16: NAS, Office: dslr $\to$ amazon

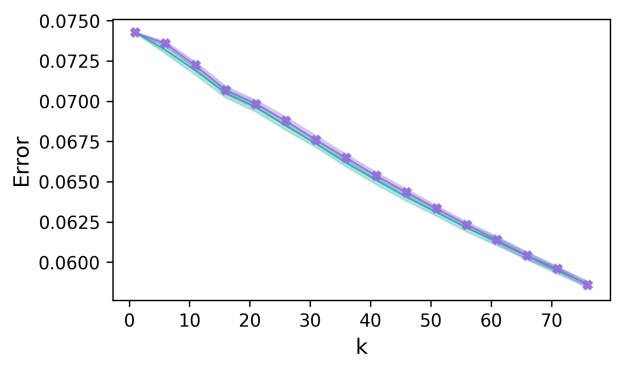
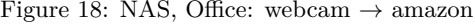

Figure 17: NAS, Office: amazon $\to$ webcam

Figure 18: NAS, Office: webcam $\to$ amazon

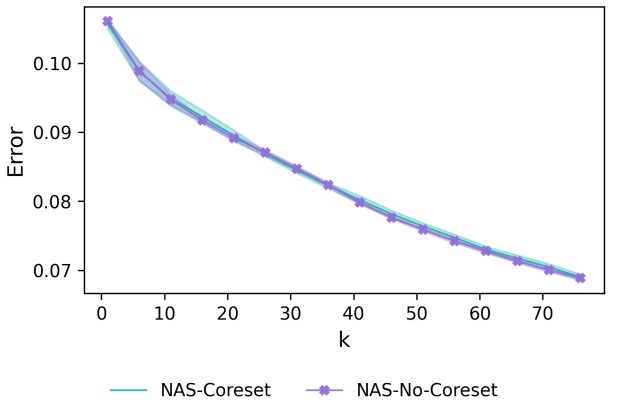

Figure 19: NAS, Office: dslr → webcam

Figure 20: NAS, Office: webcam → dslr

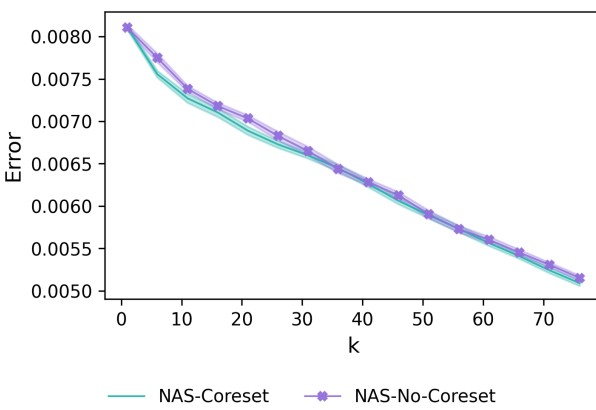

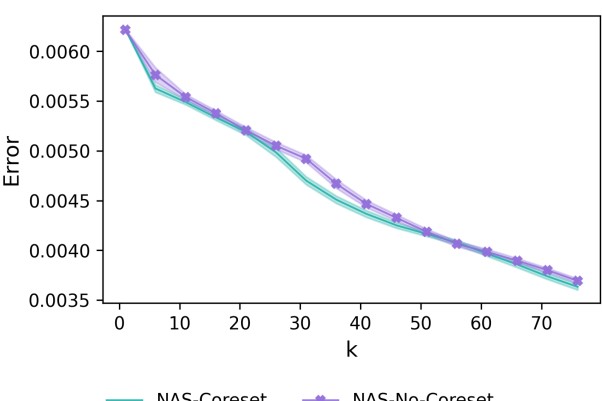

Figure 21: NAS, Superconductivity: l → ml

Figure 22: NAS, Superconductivity: ml → l

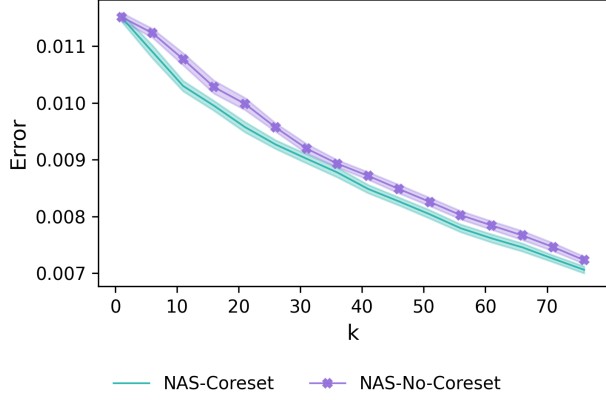

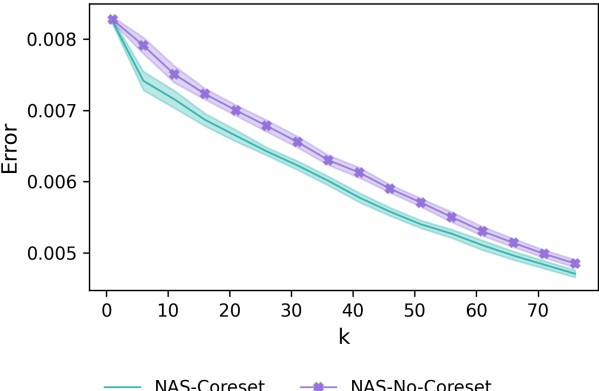

Figure 23: NAS, Superconductivity: ml → mh

Figure 24: NAS, Superconductivity: mh → ml

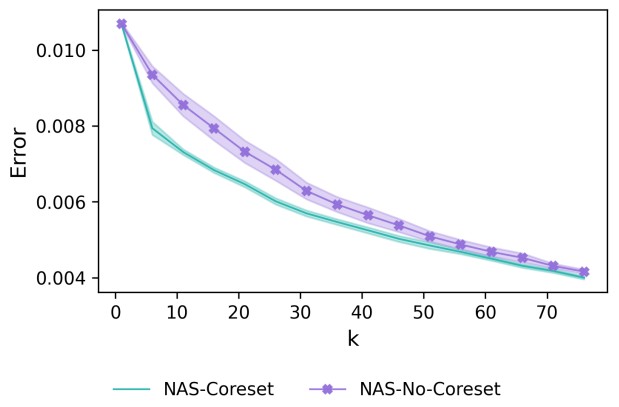

Figure 25: NAS, Superconductivity: mh → l

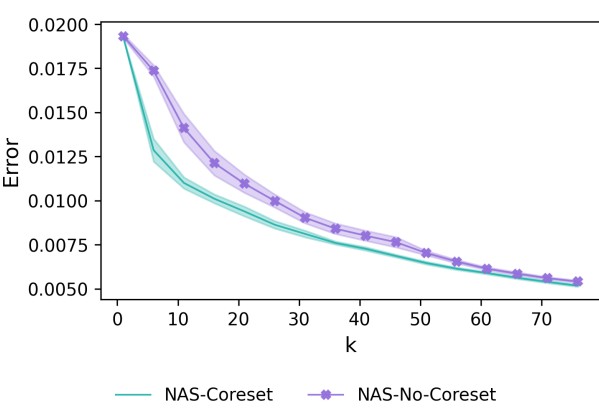

Figure 27: NAS, Superconductivity: h → l

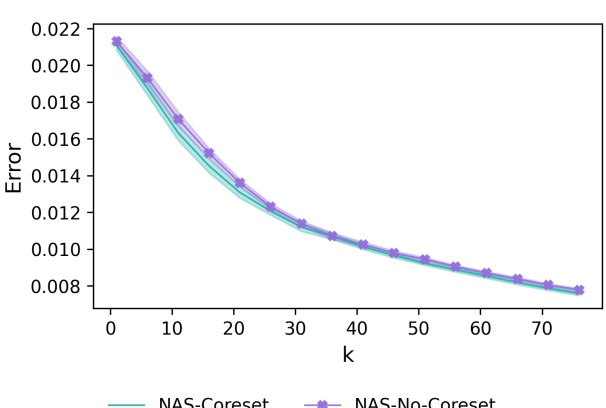

Figure 26: NAS, Superconductivity: l → mh

Figure 28: NAS, Superconductivity: l → h

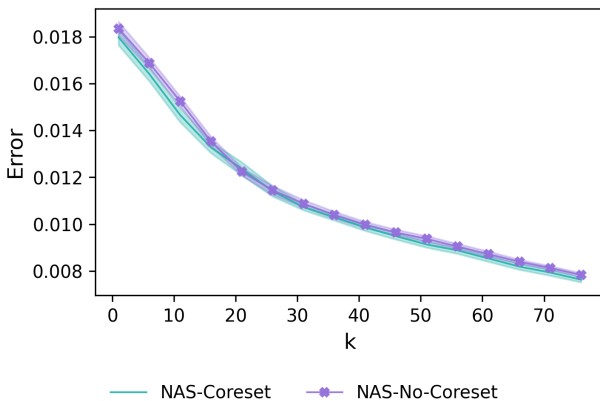

Figure 29: NAS, Superconductivity: ml → h

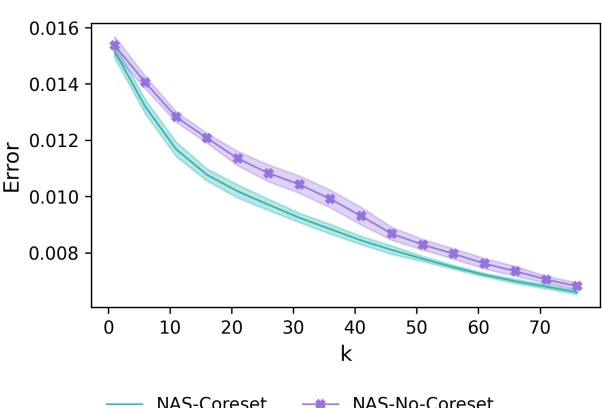

Figure 30: NAS, Superconductivity: h → ml

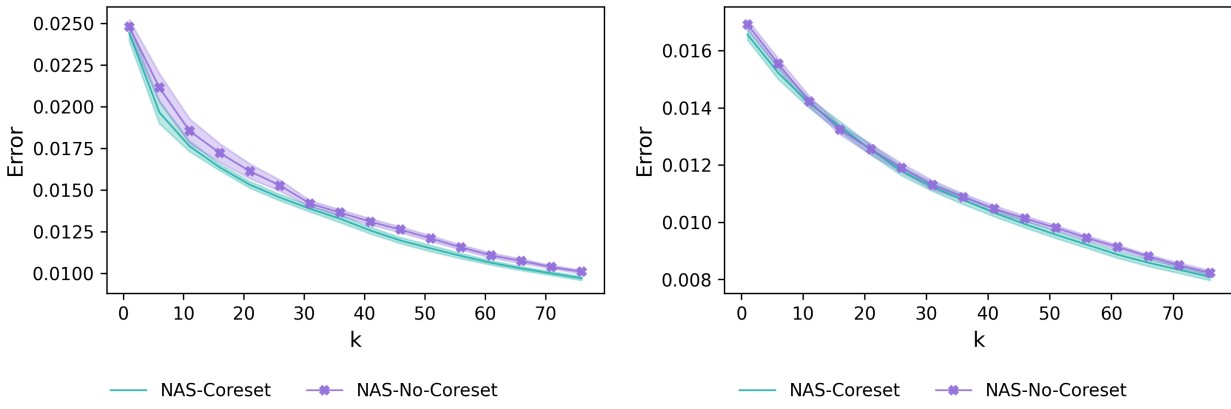

Figure 31: NAS, Superconductivity: h → mh     Figure 32: NAS, Superconductivity: mh → h

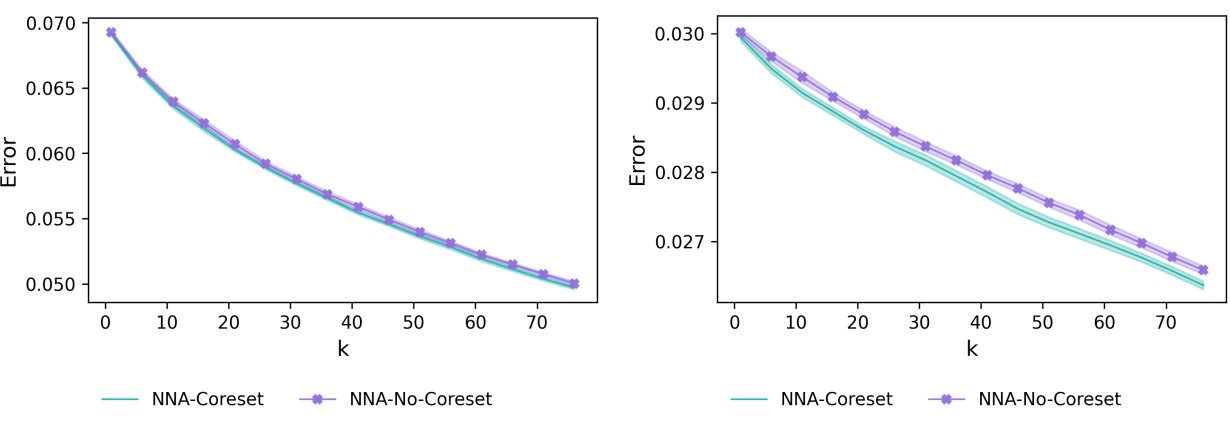

Figure 33: NNA, MNIST: 7 → 1     Figure 34: NNA, MNIST: 7 → 1

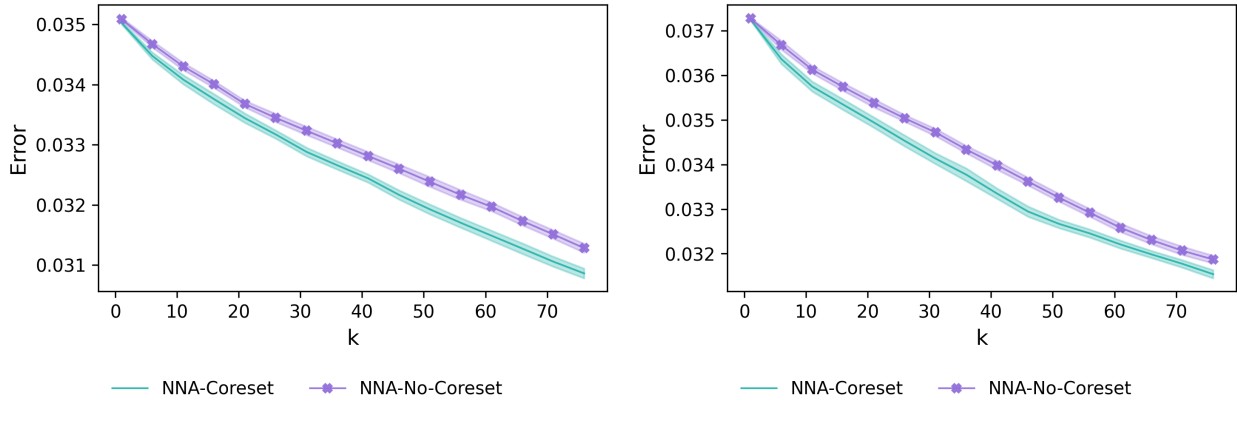

Figure 35: NNA, MNIST: 6 → 7     Figure 36: NNA, MNIST: 9 → 6

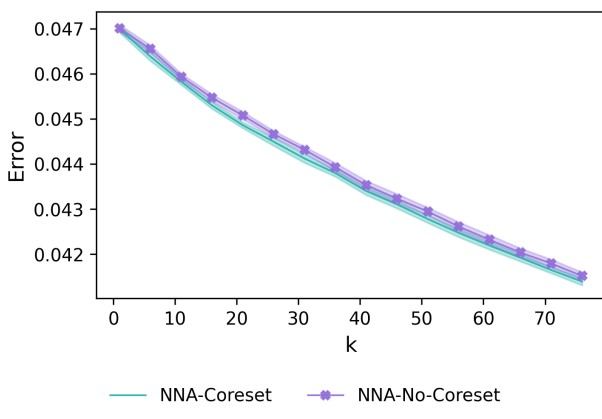

Figure 37: NNA, MNIST: 2 → 5

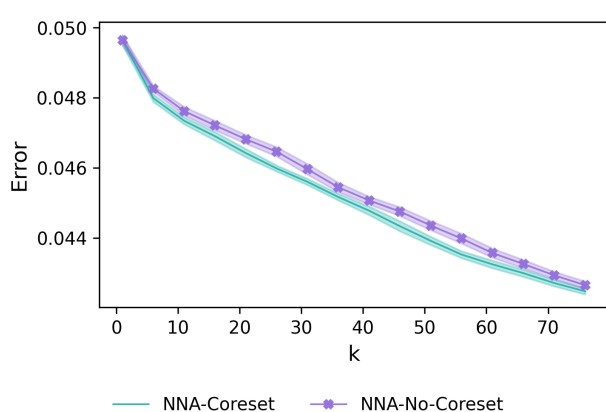

Figure 38: NNA, MNIST: 5 → 2

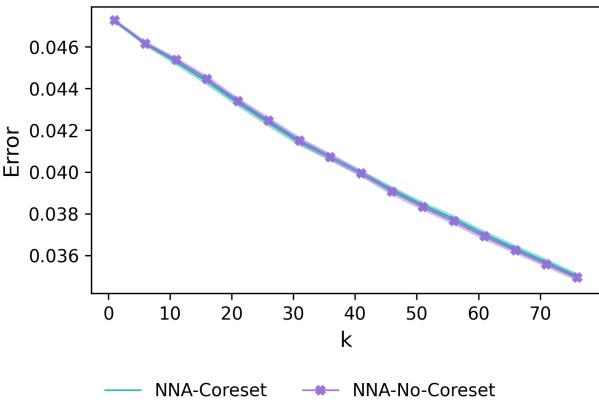

Figure 39: NNA, Office: amazon → dslr

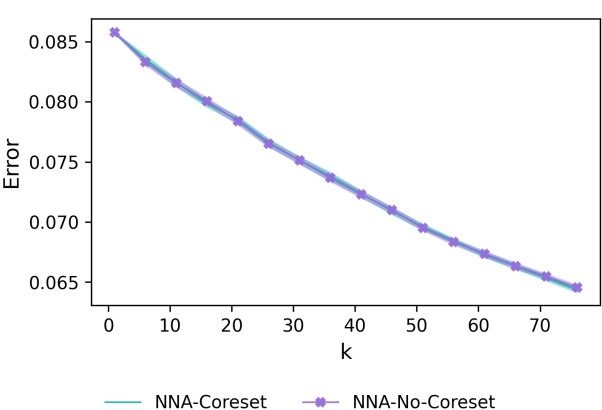

Figure 40: NNA, Office: dslr → amazon

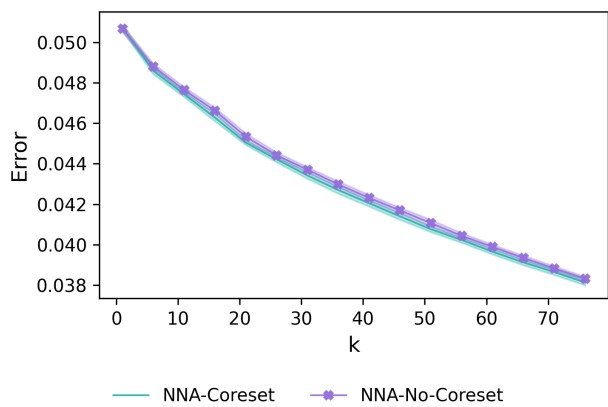

Figure 41: NNA, Office: amazon → webcam

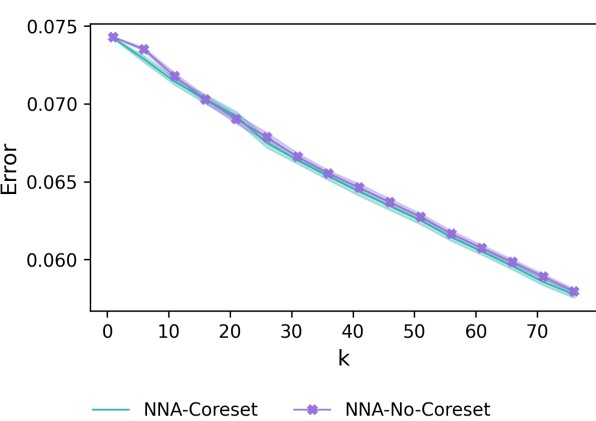

Figure 42: NNA, Office: webcam → amazon

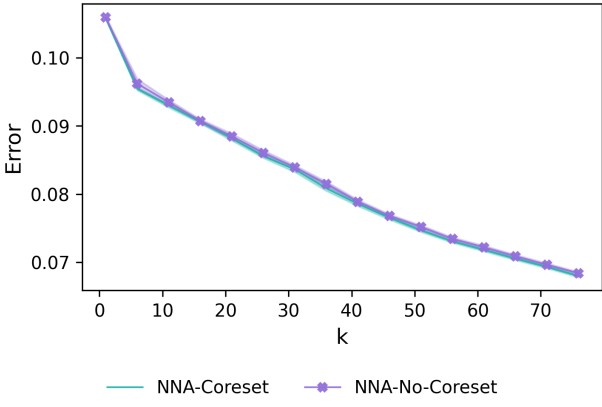

Figure 43: NNA, Office: dslr → webcam

Figure 44: NNA, Office: webcam → dslr

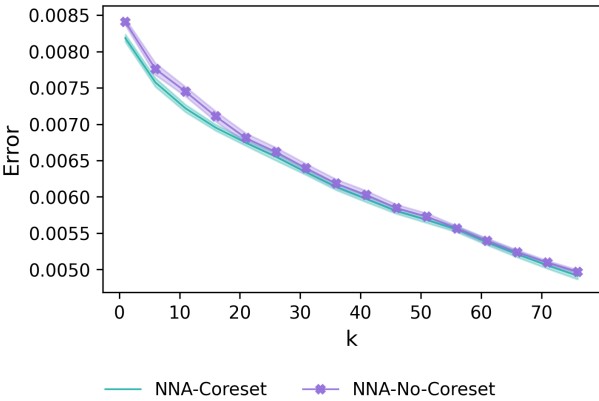

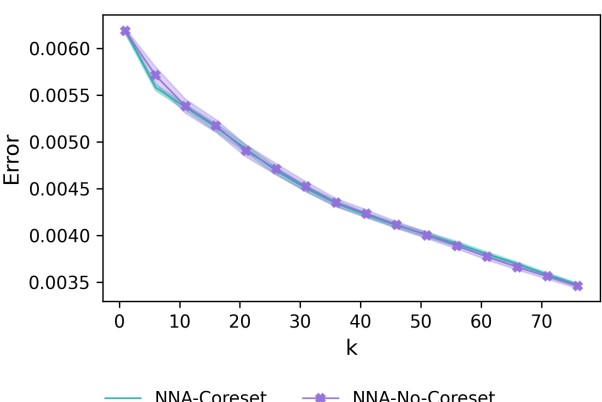

Figure 45: NNA, Superconductivity: l → ml

Figure 46: NNA, Superconductivity: ml → l

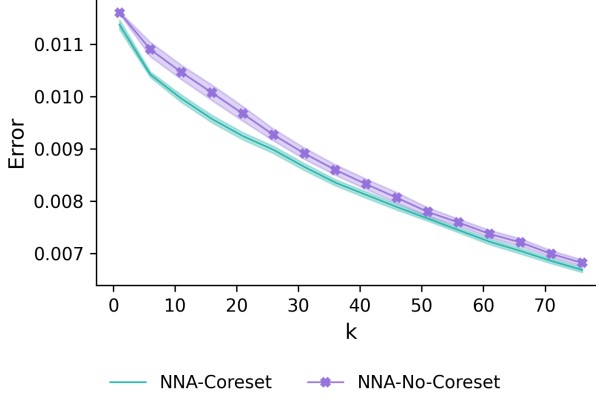

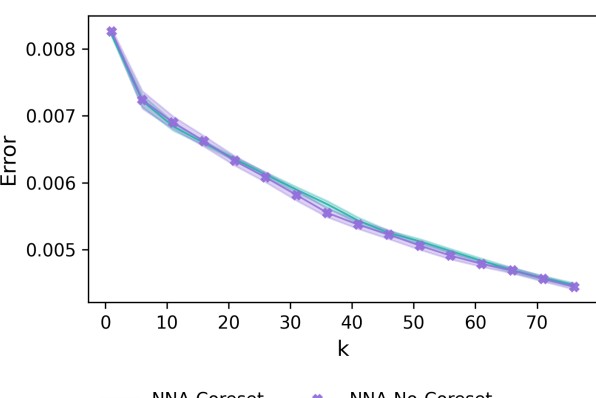

Figure 47: NNA, Superconductivity: ml → mh

Figure 48: NNA, Superconductivity: mh → ml

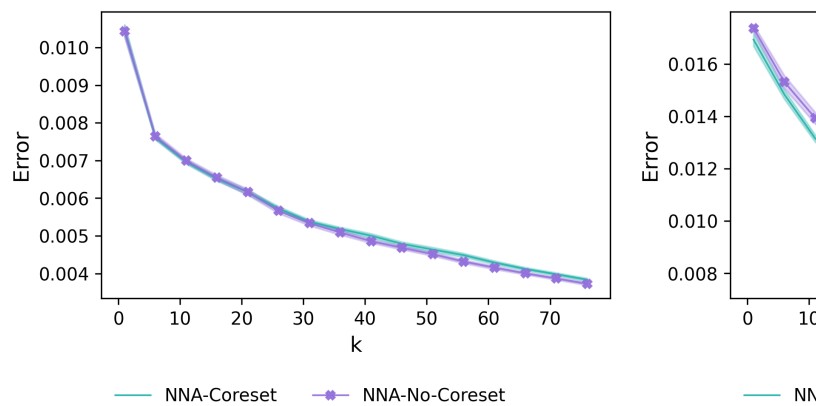

Figure 49: NNA, Superconductivity: mh → l

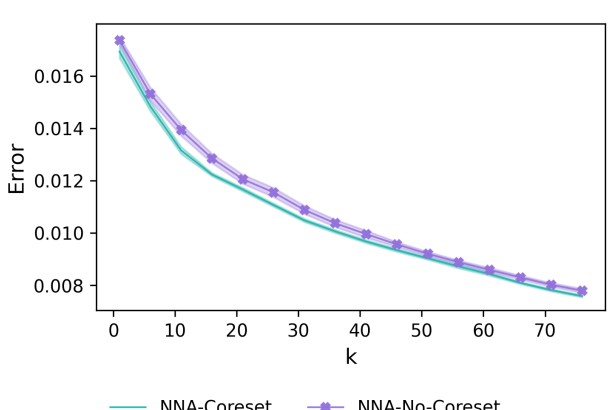

Figure 50: NNA, Superconductivity: l → mh

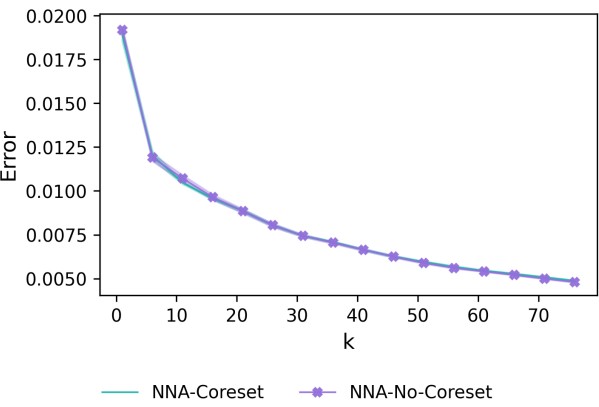

Figure 51: NNA, Superconductivity: h → l

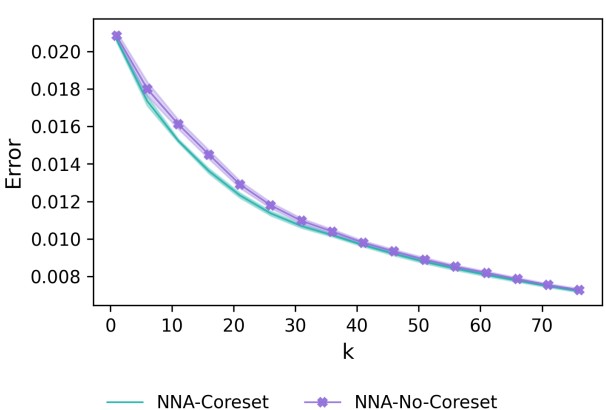

Figure 52: NNA, Superconductivity: l → h

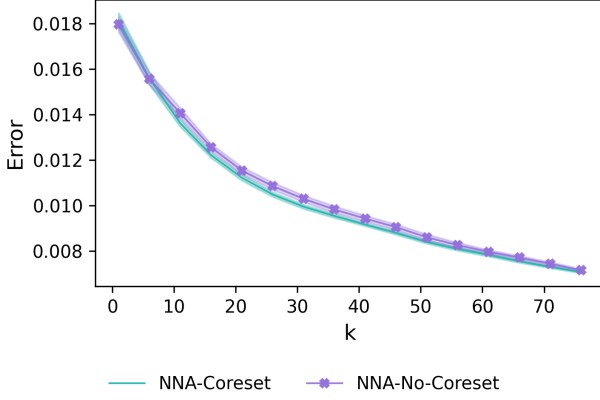

Figure 53: NNA, Superconductivity: ml → h

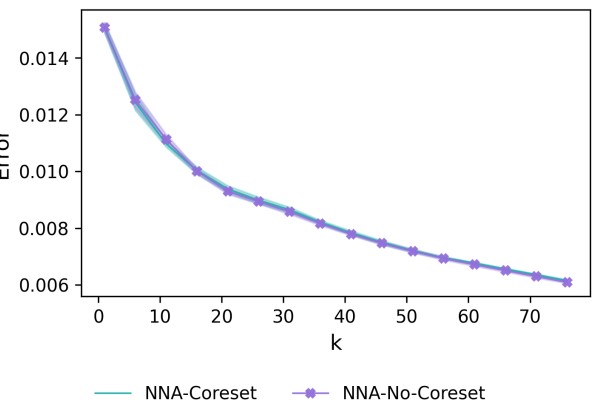

Figure 54: NNA, Superconductivity: h → ml

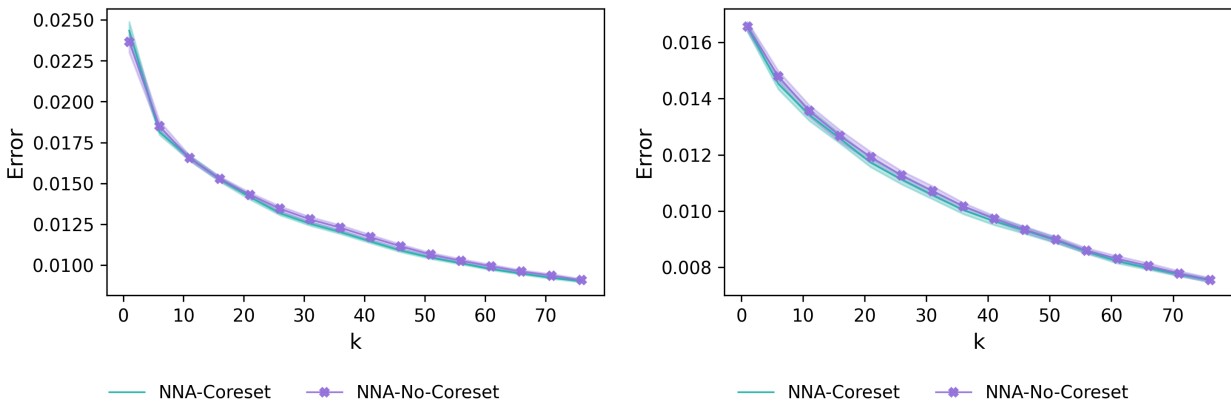

Figure 55: NNA, Superconductivity: h → mh                    Figure 56: NNA, Superconductivity: mh → h

## C.4 Experiments for DP with real-world datasets

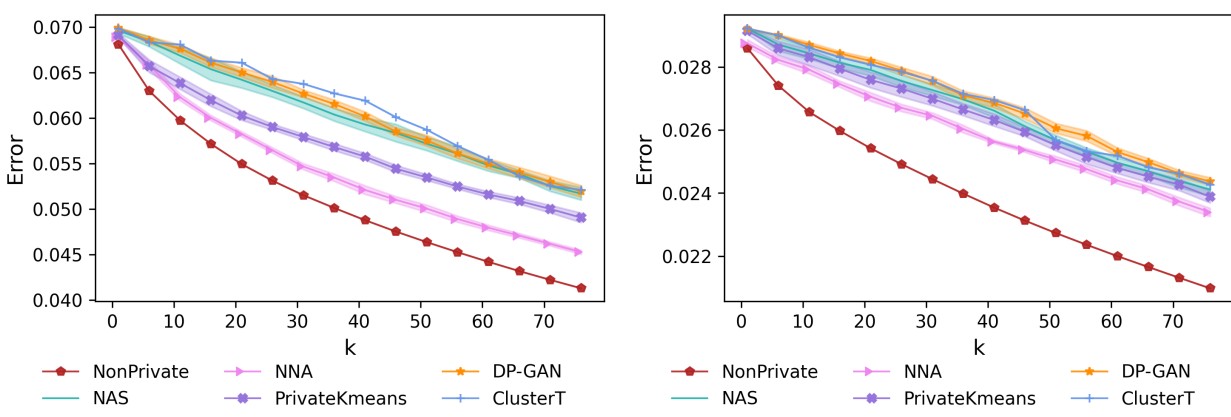

Figure 57: MNIST: 7 → 1                    Figure 58: MNIST: 7 → 1

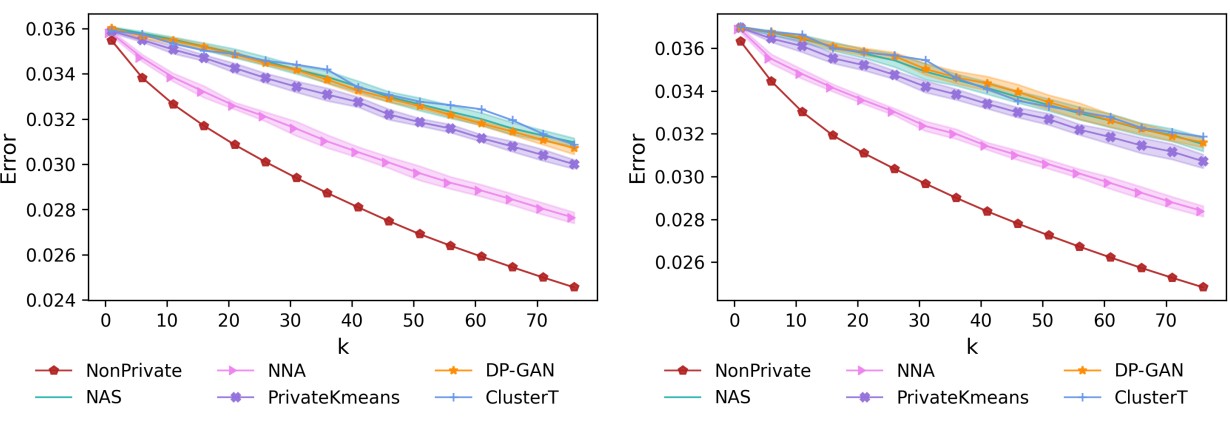

Figure 59: MNIST: 6 → 7                    Figure 60: MNIST: 9 → 6

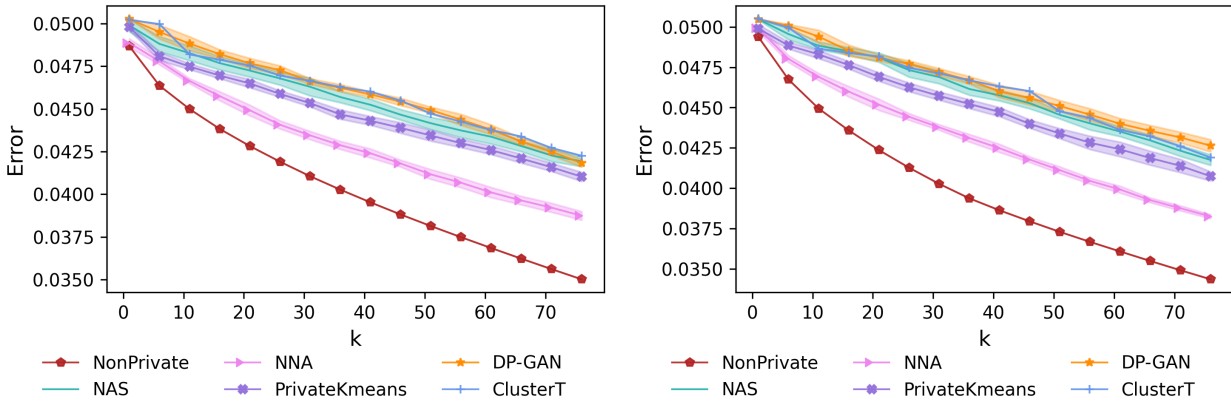

Figure 61: MNIST: 2 → 5

Figure 62: MNIST: 5 → 2

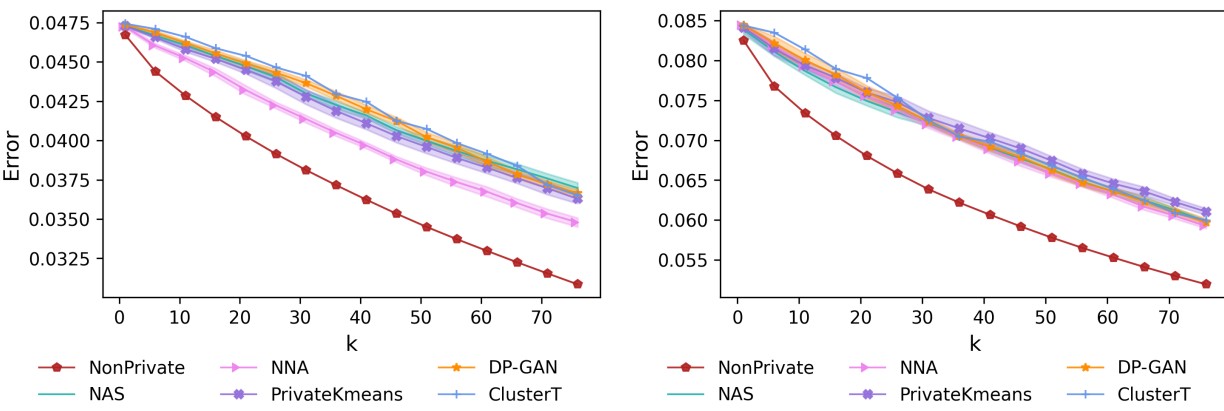

Figure 63: Office: amazon → dslr

Figure 64: Office: dslr → amazon

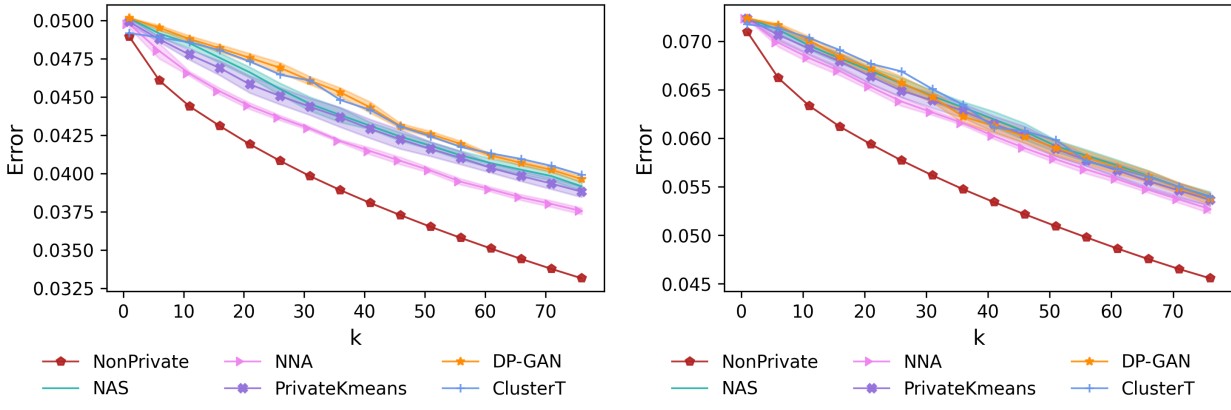

Figure 65: Office: amazon → webcam

Figure 66: Office: webcam → amazon

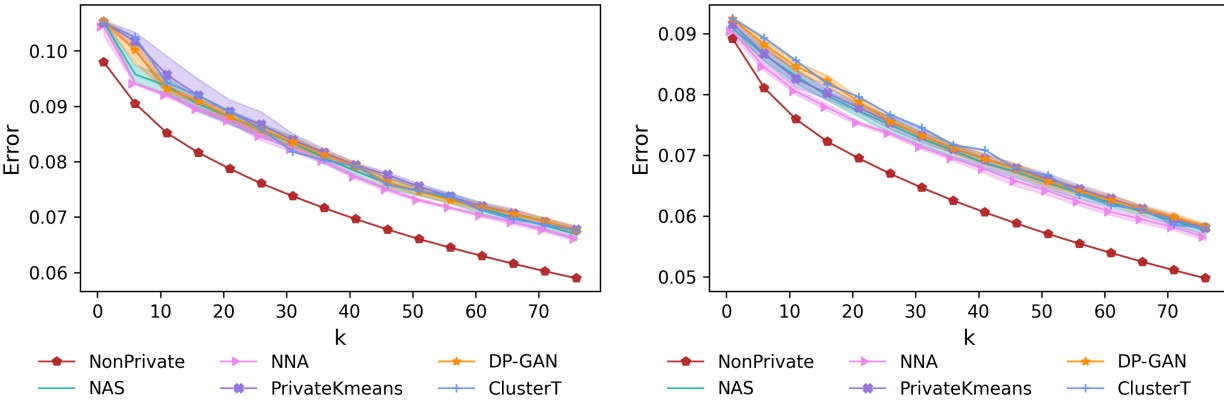

Figure 67: Office: dslr → webcam

Figure 68: Office: webcam → dslr

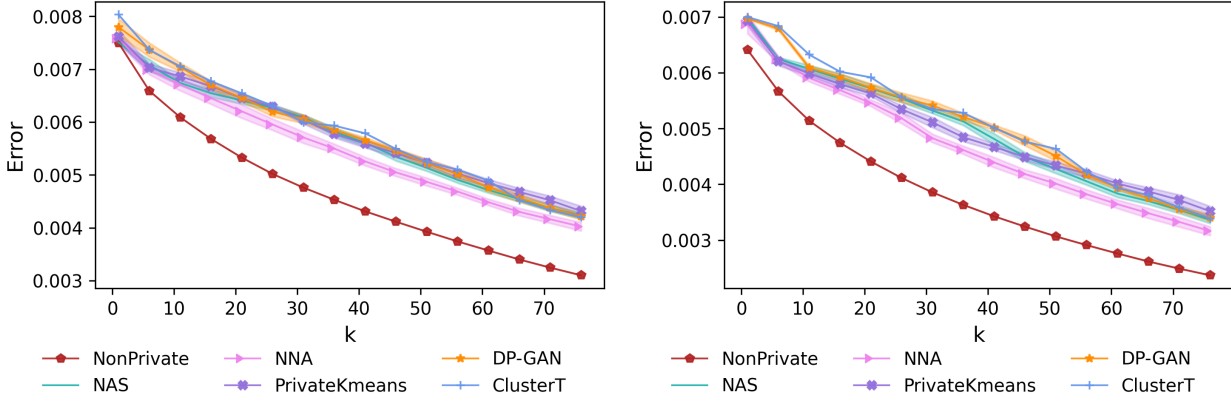

Figure 69: Superconductivity: l → ml

Figure 70: Superconductivity: ml → l

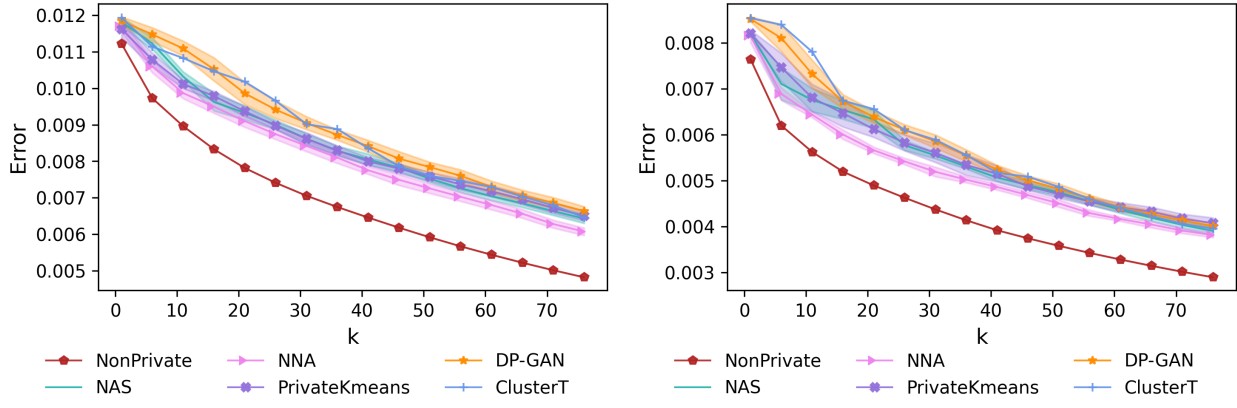

Figure 71: Superconductivity: ml → mh

Figure 72: Superconductivity: mh → ml

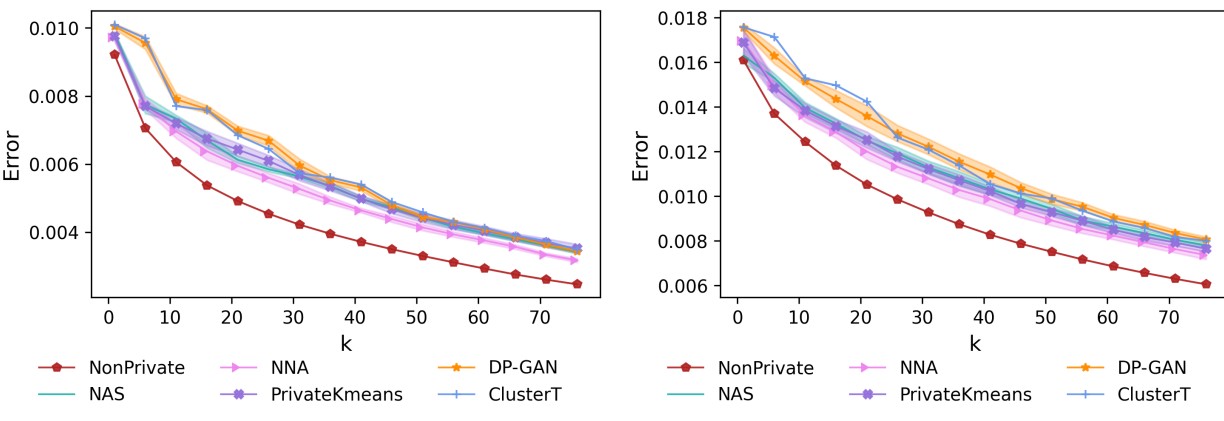

Figure 73: Superconductivity: mh → l

Figure 74: Superconductivity: l → mh

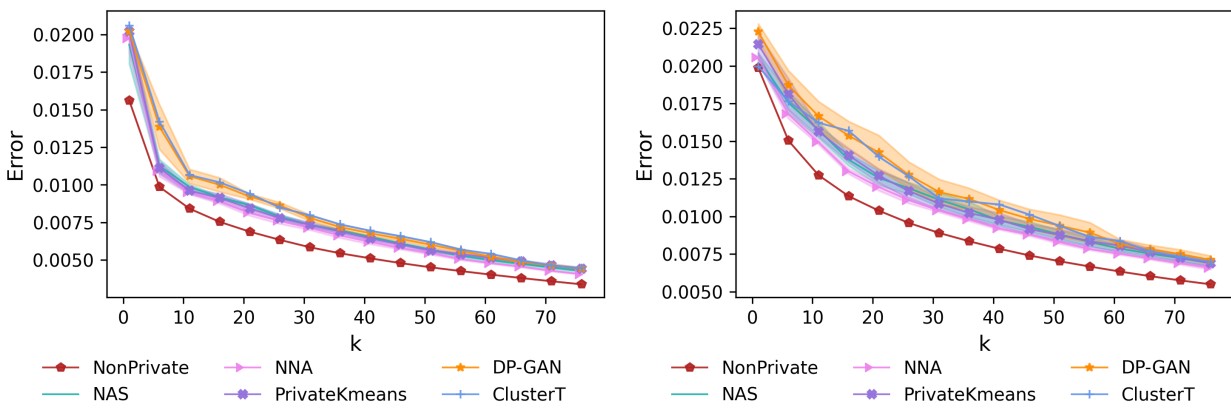

Figure 75: Superconductivity: h → l

Figure 76: Superconductivity: l → h

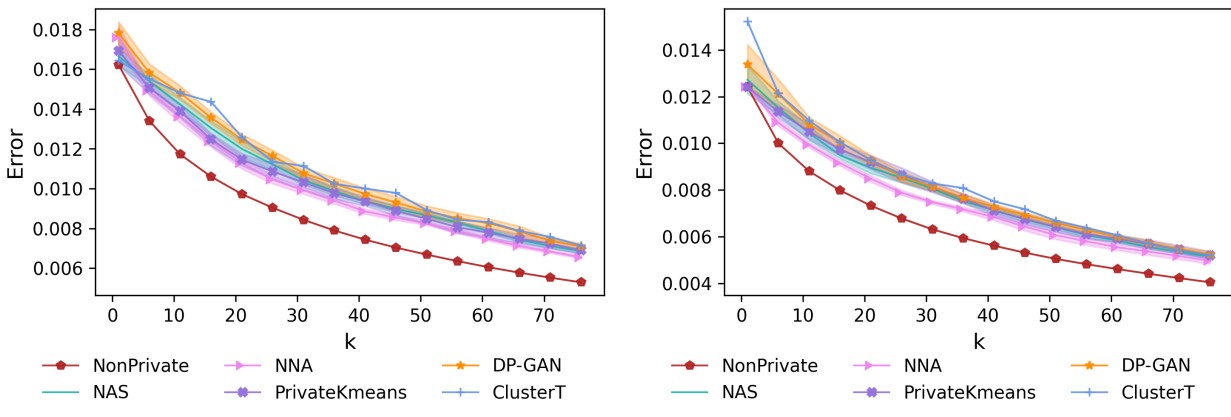

Figure 77: Superconductivity: ml → h

Figure 78: Superconductivity: h → ml

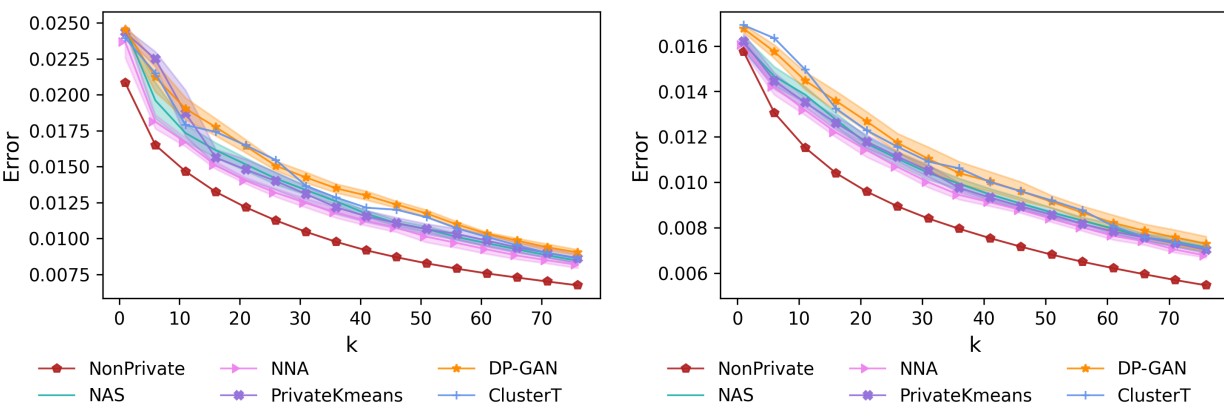

Figure 79: Superconductivity: h → mh

Figure 80: Superconductivity: mh → h

## C.5    Experiments for zCDP with real-world datasets

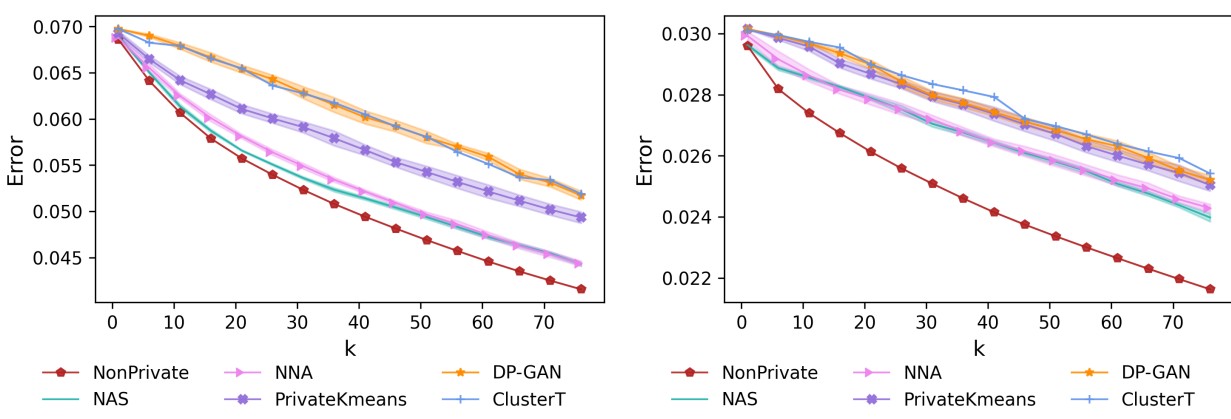

Figure 81: MNIST: 1 → 7

Figure 82: MNIST: 7 → 1

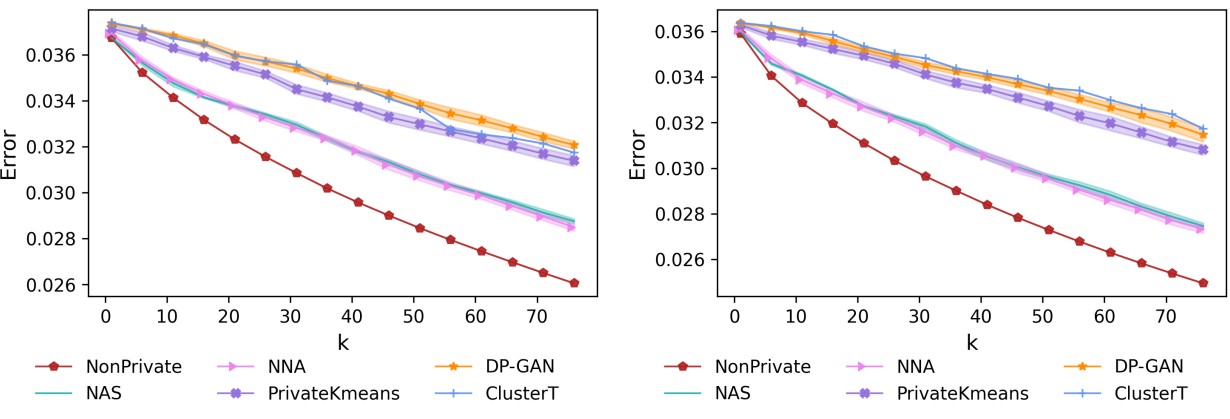

Figure 83: MNIST: 6 → 7

Figure 84: MNIST: 9 → 6

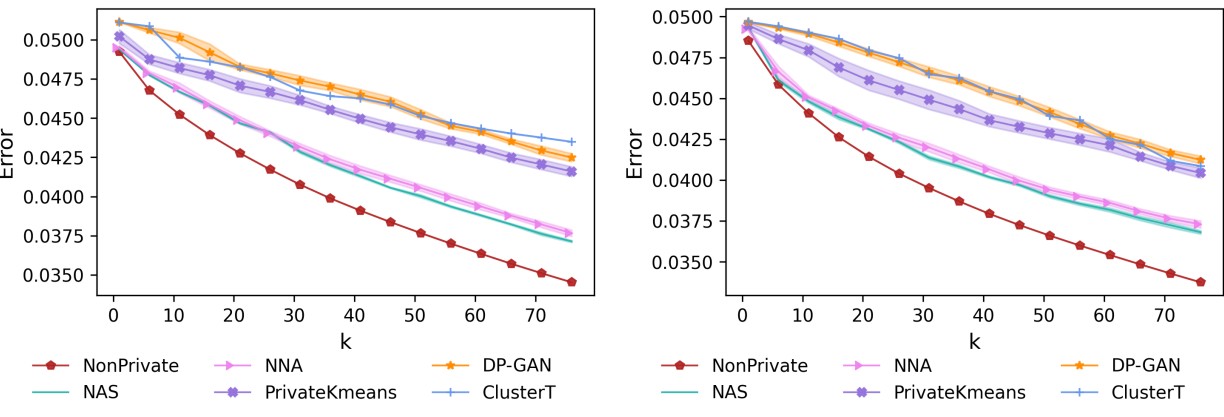

Figure 85: MNIST: 2 → 5

Figure 86: MNIST: 5 → 2

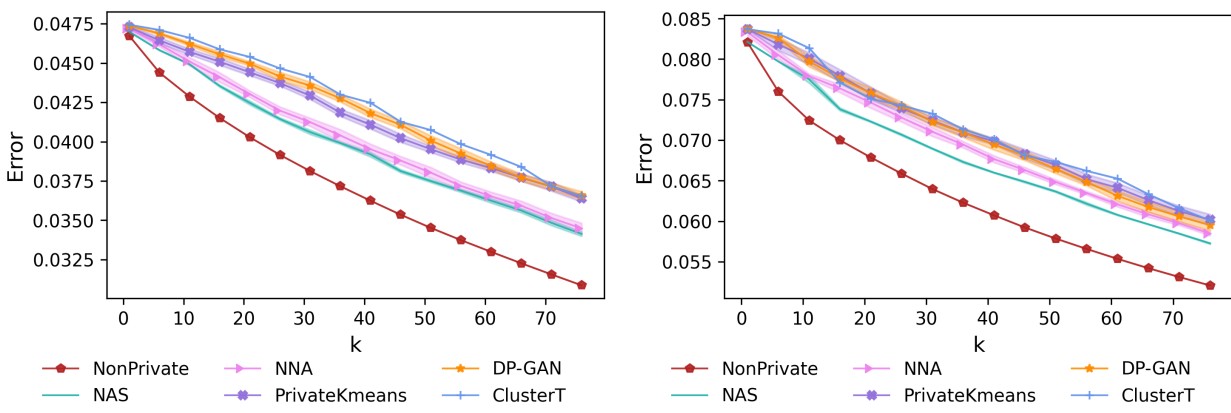

Figure 87: Office: amazon → dslr

Figure 88: Office: dslr → amazon

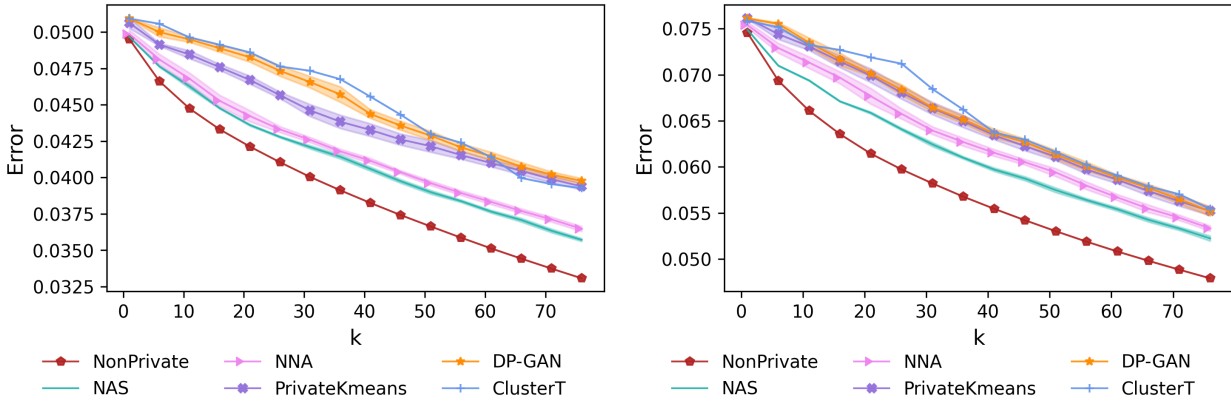

Figure 89: Office: amazon → webcam

Figure 90: Office: webcam → amazon

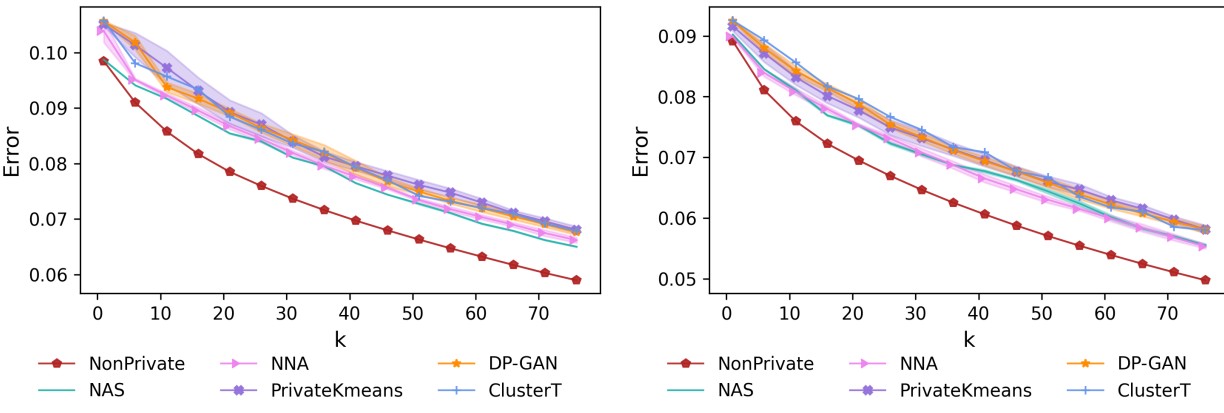

Figure 91: Office: dslr → webcam

Figure 92: Office: webcam → dslr

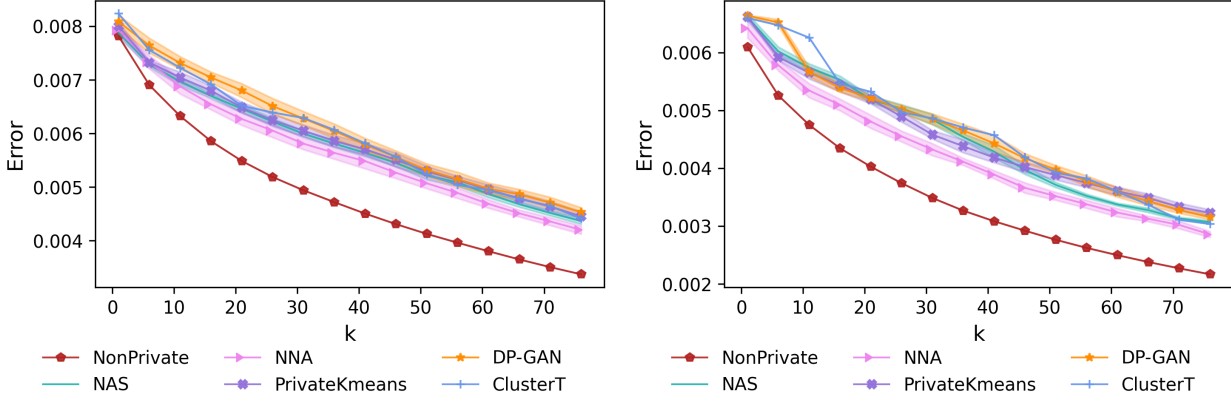

Figure 93: Superconductivity: l → ml

Figure 94: Superconductivity: ml → l

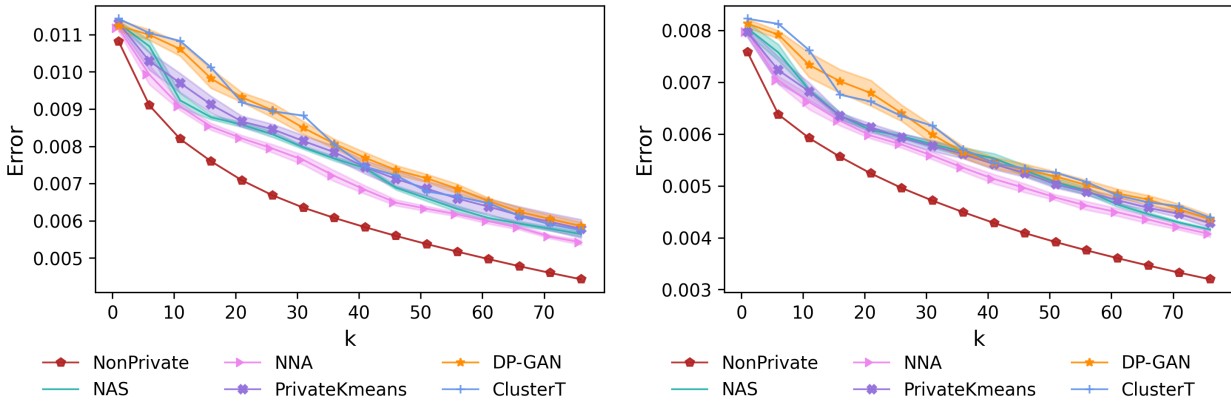

Figure 95: Superconductivity: ml → mh

Figure 96: Superconductivity: mh → ml

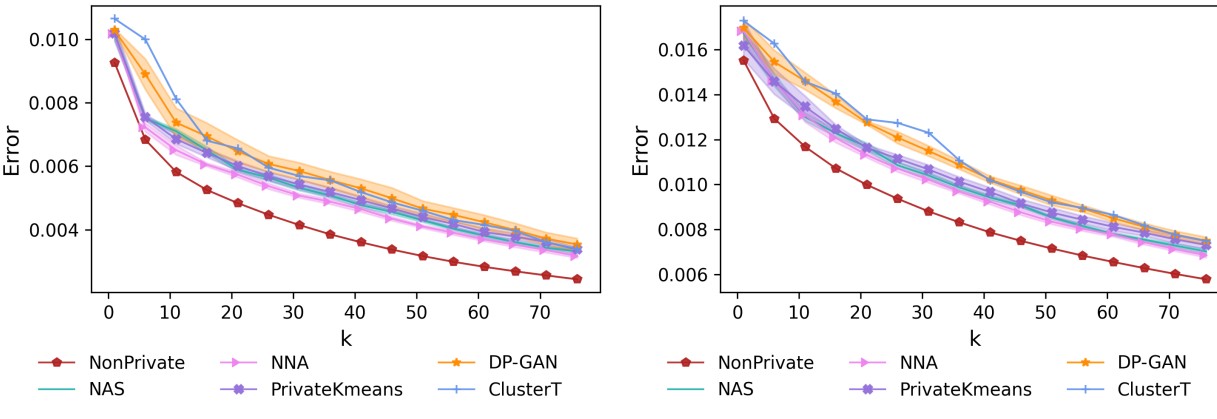

Figure 97: Superconductivity: mh → l                Figure 98: Superconductivity: l → mh

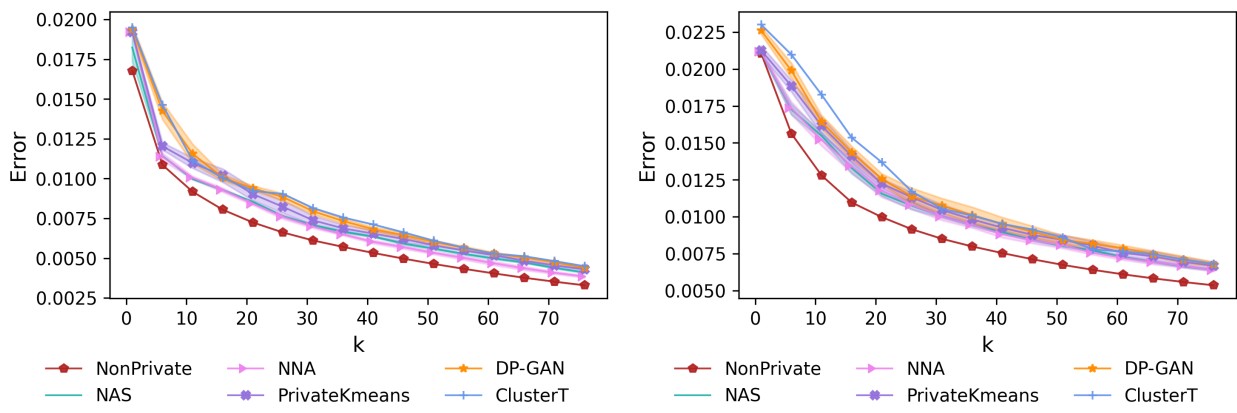

Figure 99: Superconductivity: h → l                Figure 100: Superconductivity: l → h

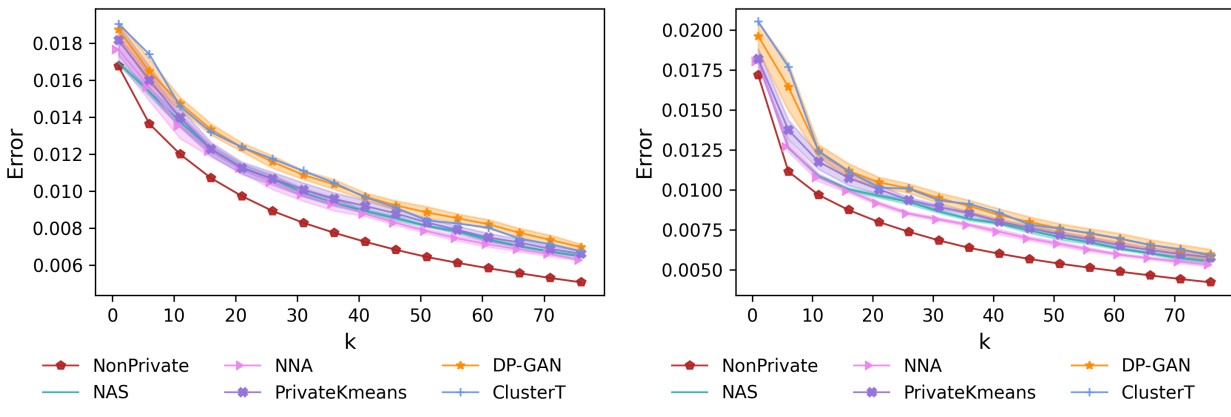

Figure 101: Superconductivity: ml → h                Figure 102: Superconductivity: h → ml

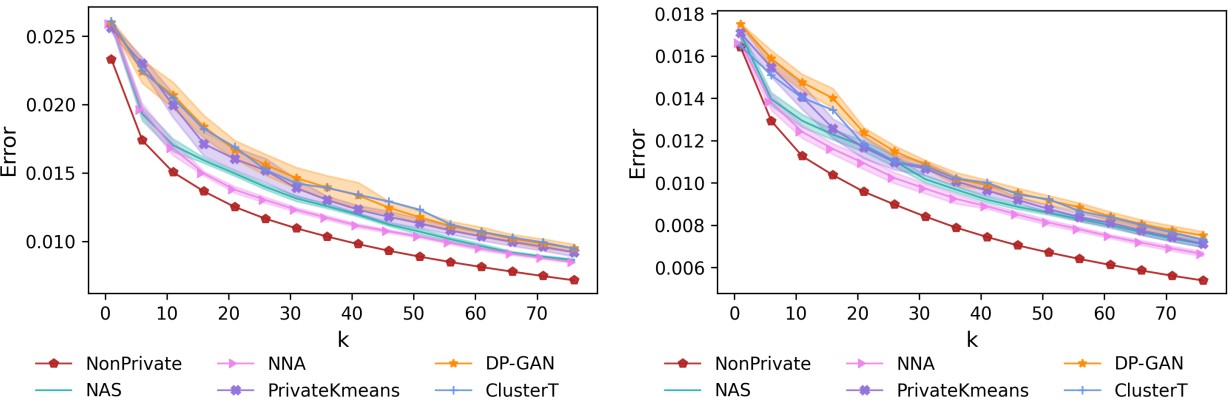

Figure 103: Superconductivity: h → mh

Figure 104: Superconductivity: mh → h

## C.6 The effect of coresets on the baseline algorithms

As discussed in Section 6, using a coreset before clustering $\mathcal{T}$ is helpful for our algorithms, because it reduces the size $n$ of the clustered data, which affects the amount of added noise. In contrast, using coresets for the baseline algorithms is not expected to improve their results, since they gain no benefit from a smaller $\mathcal{T}$. To verify this, we report experiments comparing the use of the baseline algorithms with and without a coreset. It can be seen that using the coreset has a negligible effect on the results of the baseline algorithms. This demonstrates that the advantage of our algorithms cannot be achieved solely by using coresets.

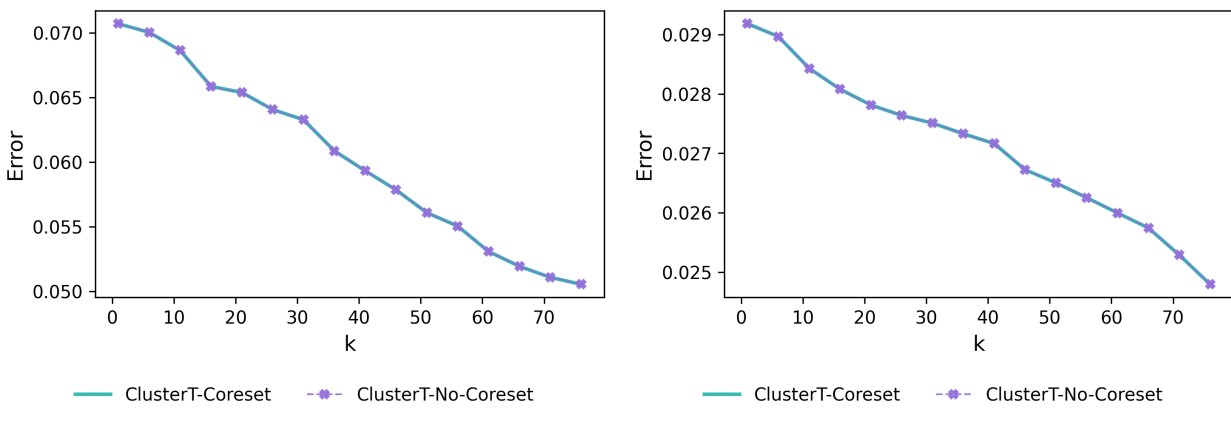

Figure 105: ClusterT, MNIST: 7 → 1

Figure 106: ClusterT, MNIST: 7 → 1ClusterT

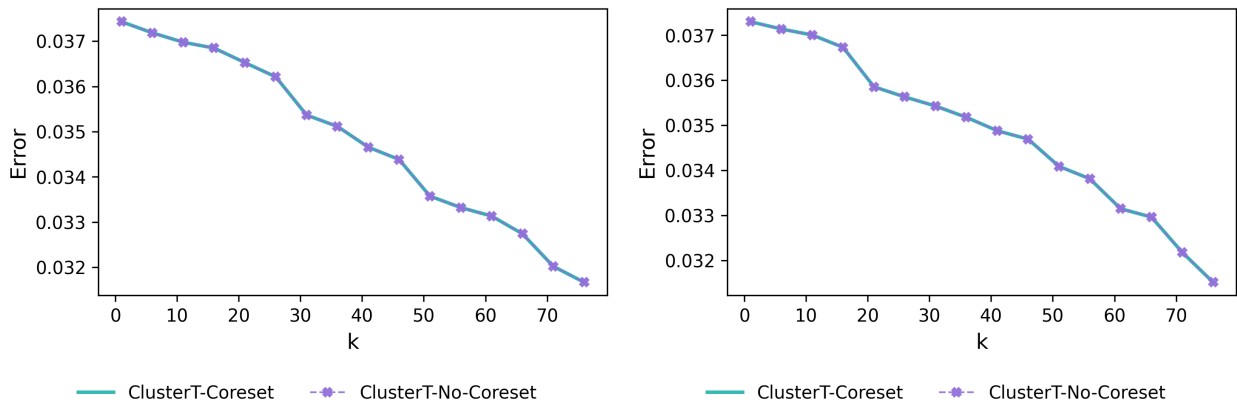

Figure 107: ClusterT, MNIST: 6 → 7

Figure 108: ClusterT, MNIST: 9 → 6

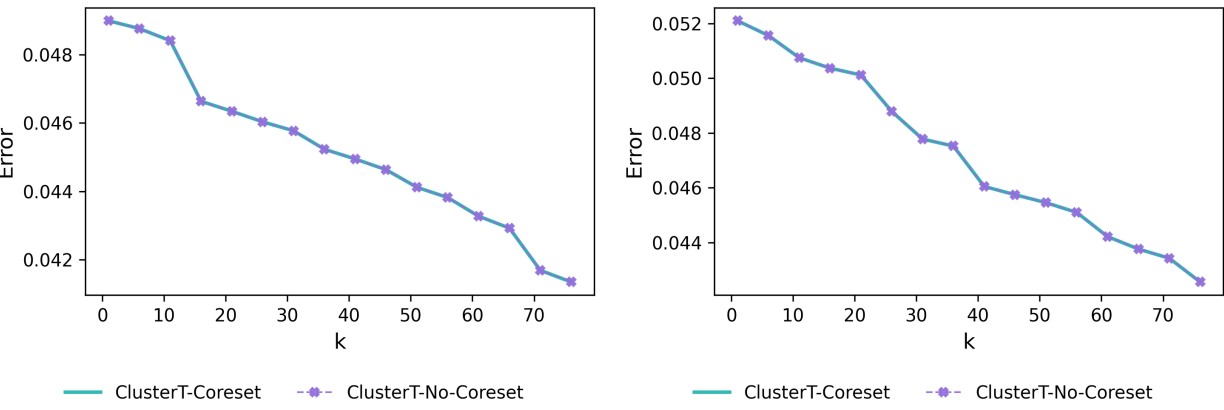

Figure 109: ClusterT, MNIST: 2 → 5

Figure 110: ClusterT, MNIST: 5 → 2

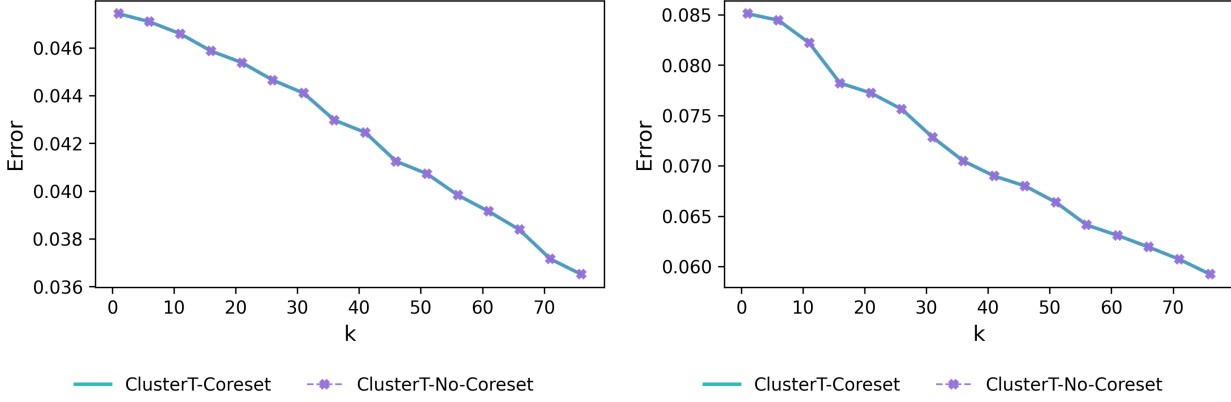

Figure 111: ClusterT, Office: amazon → dslr

Figure 112: ClusterT, Office: dslr → amazon

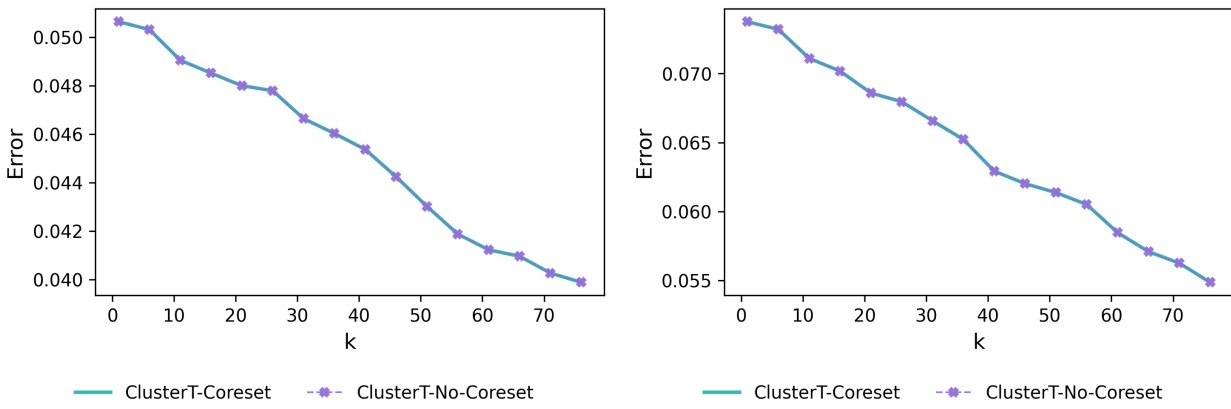

Figure 113: ClusterT, Office: amazon → webcam

Figure 114: ClusterT, Office: webcam → amazon

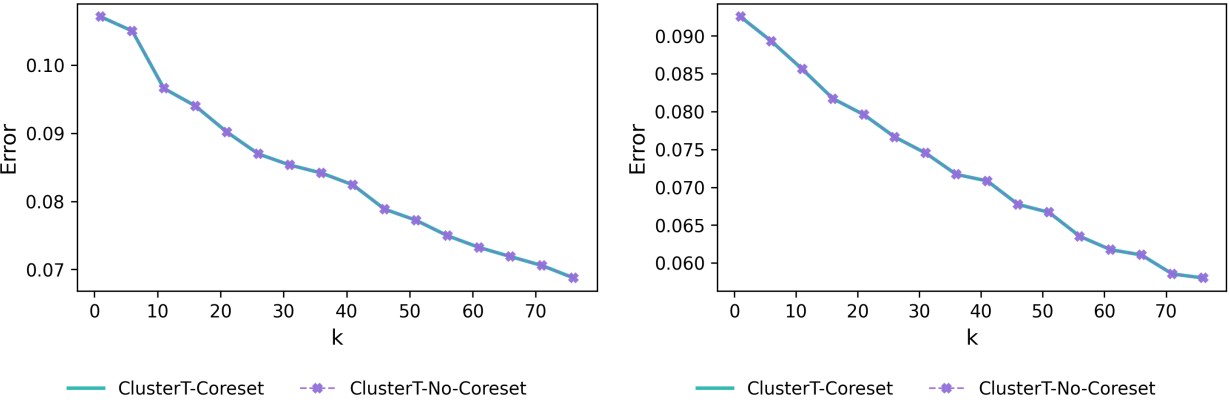

Figure 115: ClusterT, Office: dslr → webcam

Figure 116: ClusterT, Office: webcam → dslr

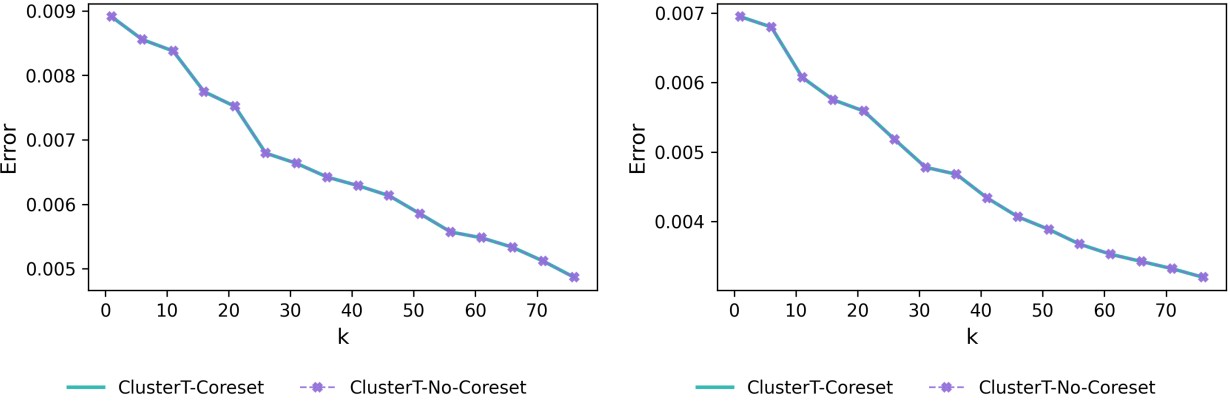

Figure 117: ClusterT, Superconductivity: l → ml

Figure 118: ClusterT, Superconductivity: ml → l

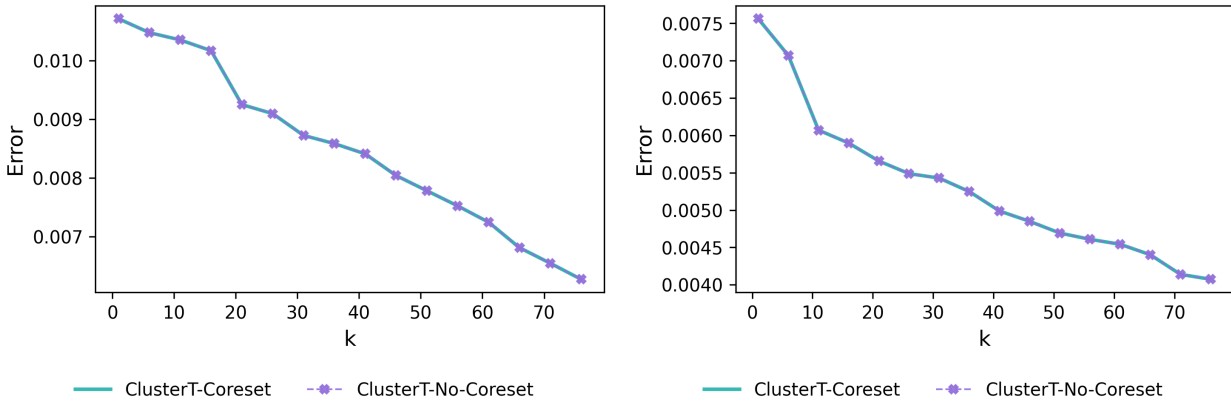

Figure 119: ClusterT, Superconductivity: ml → mh

Figure 120: ClusterT, Superconductivity: mh → ml

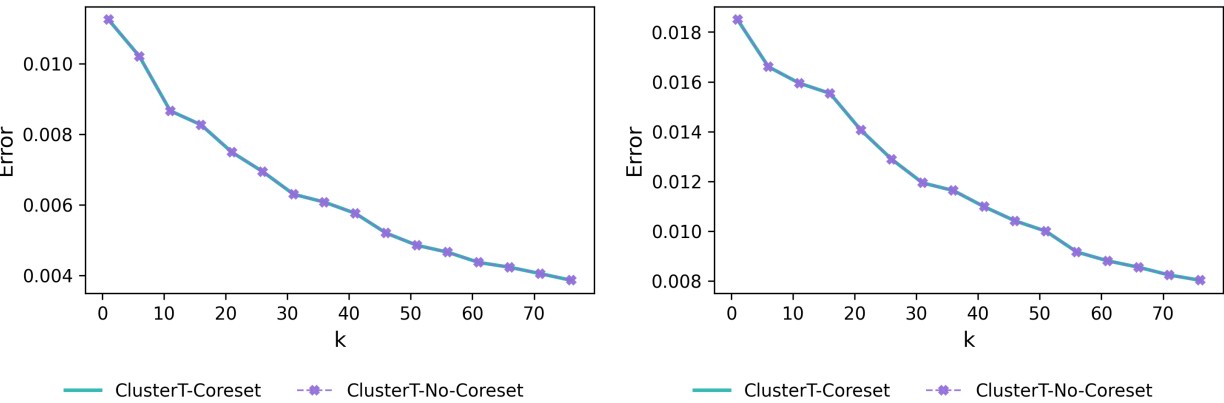

Figure 121: ClusterT, Superconductivity: mh → l

Figure 122: ClusterT, Superconductivity: l → mh

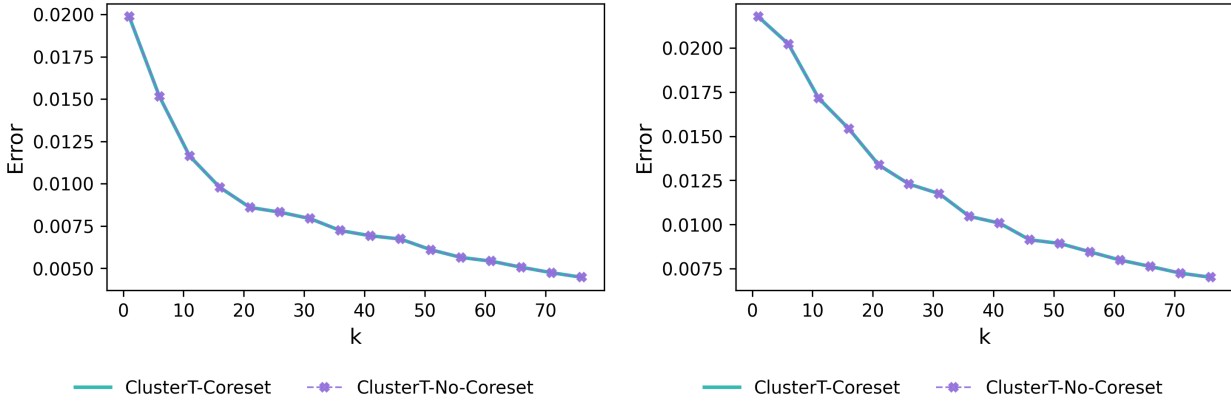

Figure 123: ClusterT, Superconductivity: h → l

Figure 124: ClusterT, Superconductivity: l → h

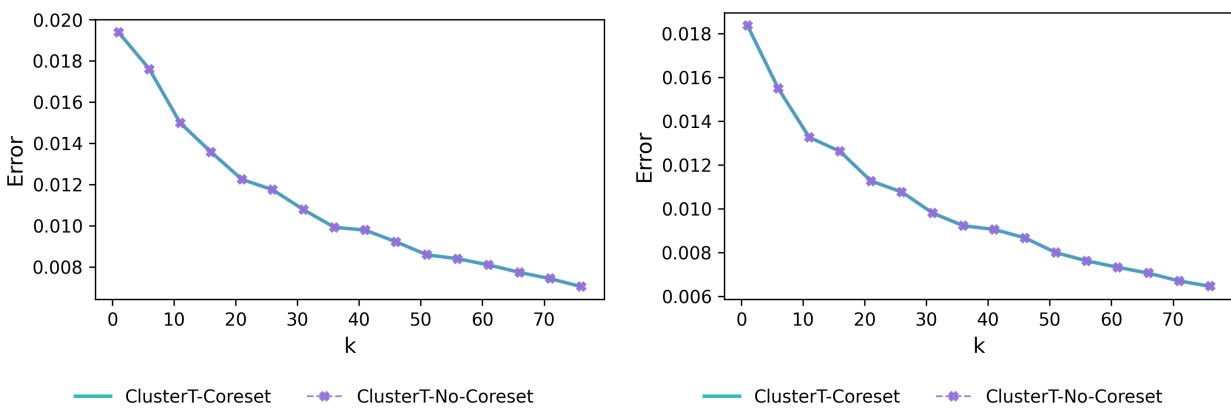

Figure 125: ClusterT, Superconductivity: ml → h    Figure 126: ClusterT, Superconductivity: h → ml

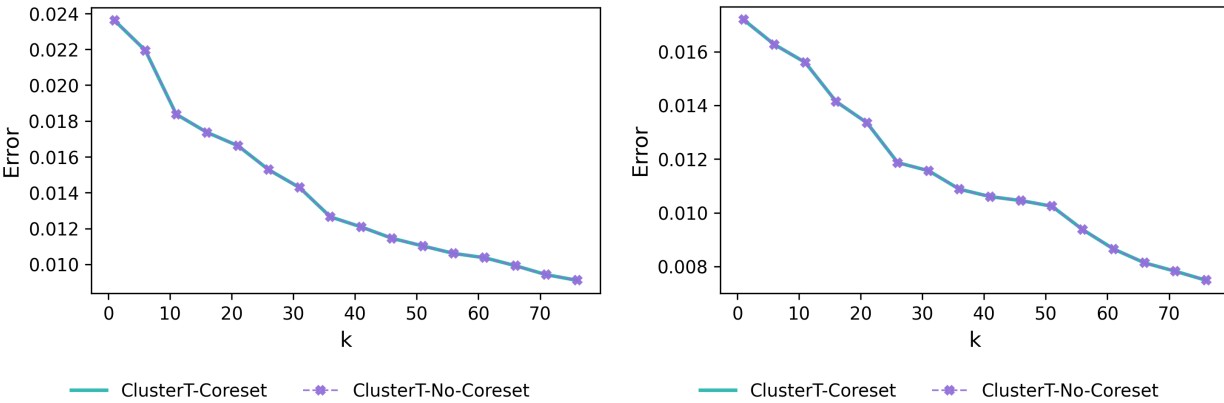

Figure 127: ClusterT, Superconductivity: h → mh    Figure 128: ClusterT, Superconductivity: mh → h

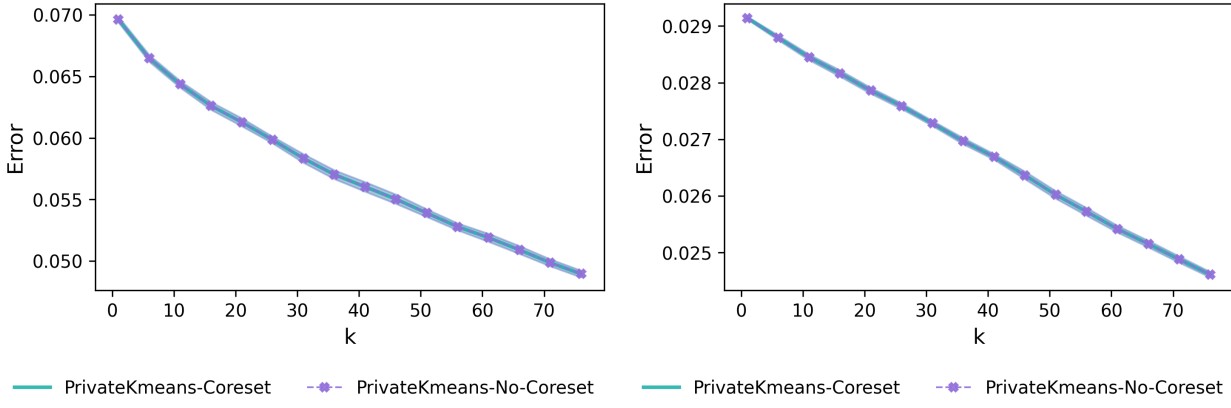

Figure 129: PrivateKmeans, MNIST: 7 → 1    Figure 130: PrivateKmeans, MNIST: 7 → 1

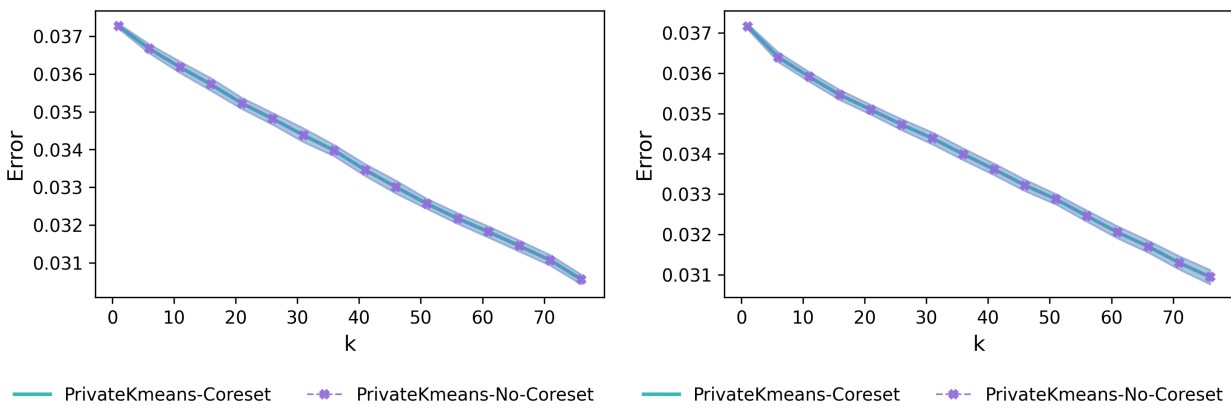

Figure 131: PrivateKmeans, MNIST: 6 → 7

Figure 132: PrivateKmeans, MNIST: 9 → 6

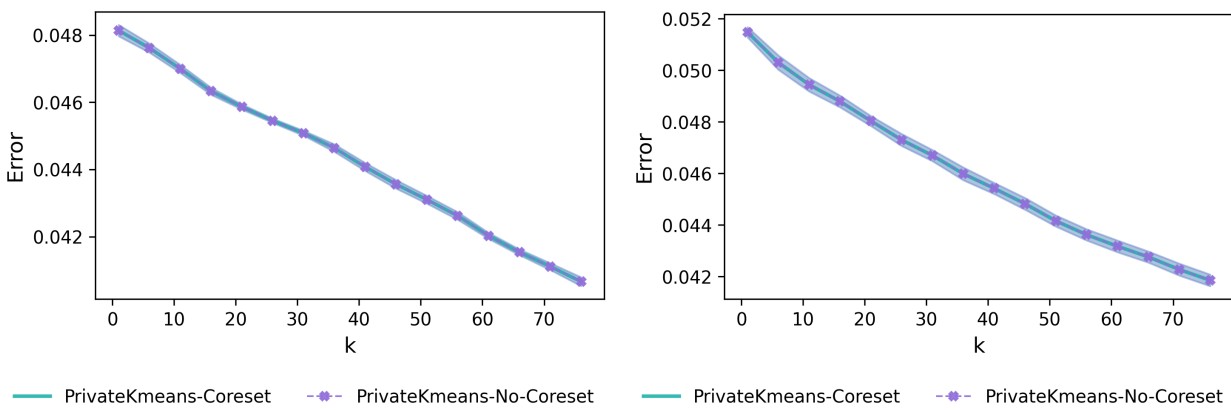

Figure 133: PrivateKmeans, MNIST: 2 → 5

Figure 134: PrivateKmeans, MNIST: 5 → 2

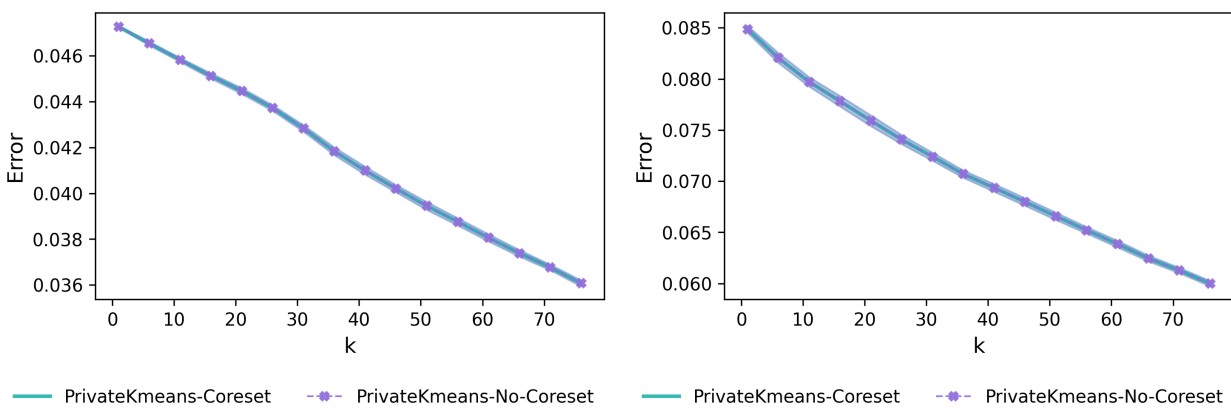

Figure 135: PrivateKmeans, Office: amazon → dslr

Figure 136: PrivateKmeans, Office: dslr → amazon

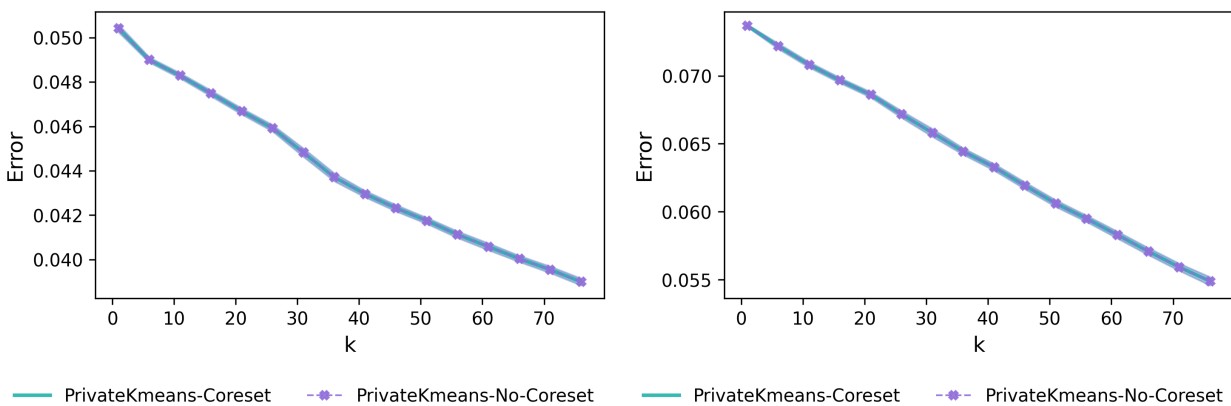

Figure 137: PrivateKmeans, Office: amazon → webcam

Figure 138: PrivateKmeans, Office: webcam → amazon

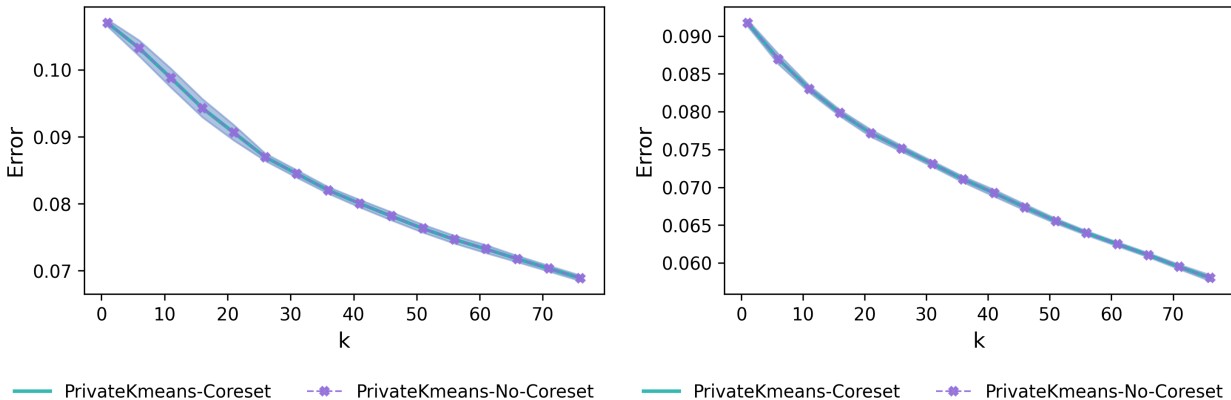

Figure 139: PrivateKmeans, Office: dslr → webcam

Figure 140: PrivateKmeans, Office: webcam → dslr

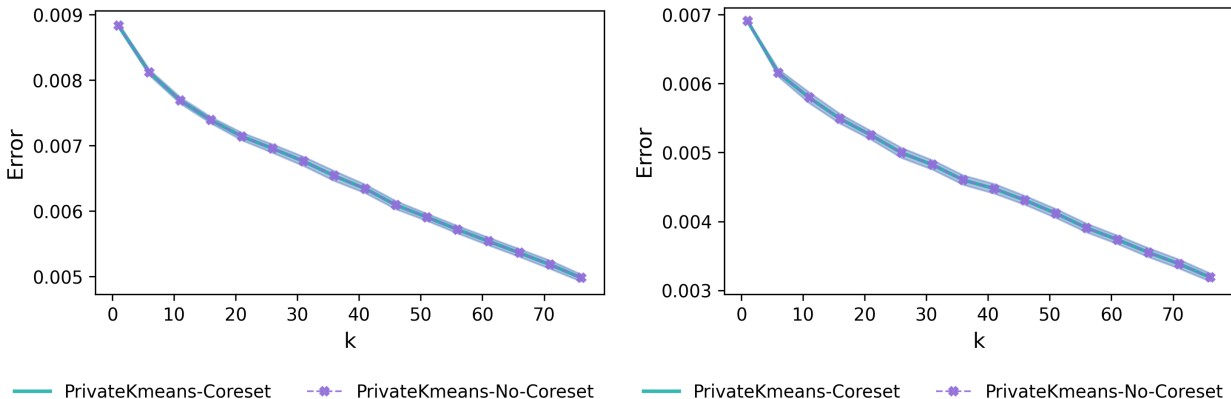

Figure 141: PrivateKmeans, Superconductivity: l → ml

Figure 142: PrivateKmeans, Superconductivity: ml → l

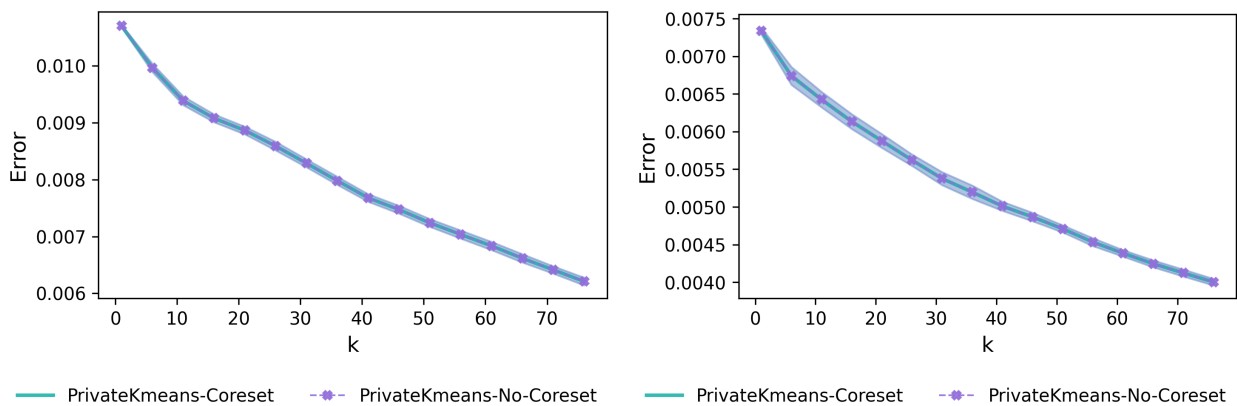

Figure 143: PrivateKmeans, Superconductivity: ml → mh   Figure 144: PrivateKmeans, Superconductivity: mh → ml

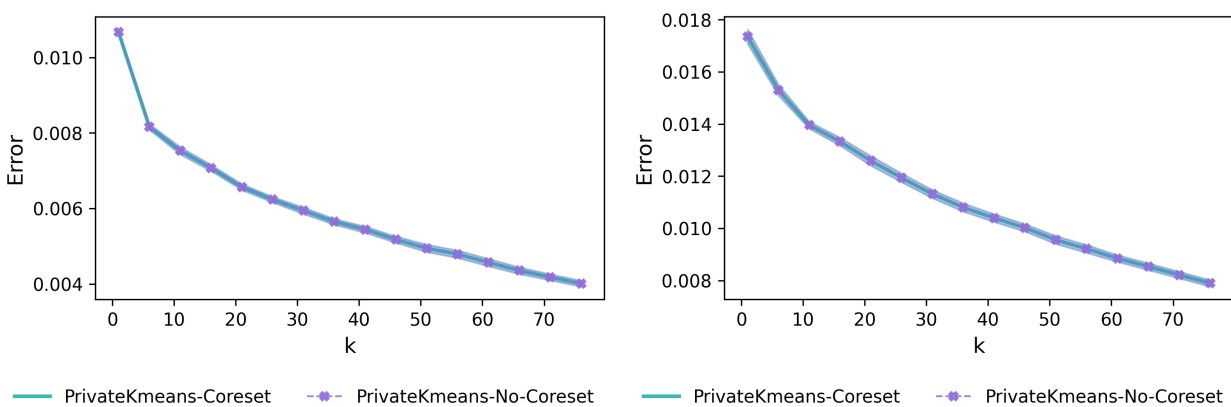

Figure 145: PrivateKmeans, Superconductivity: mh → l   Figure 146: PrivateKmeans, Superconductivity: l → mh

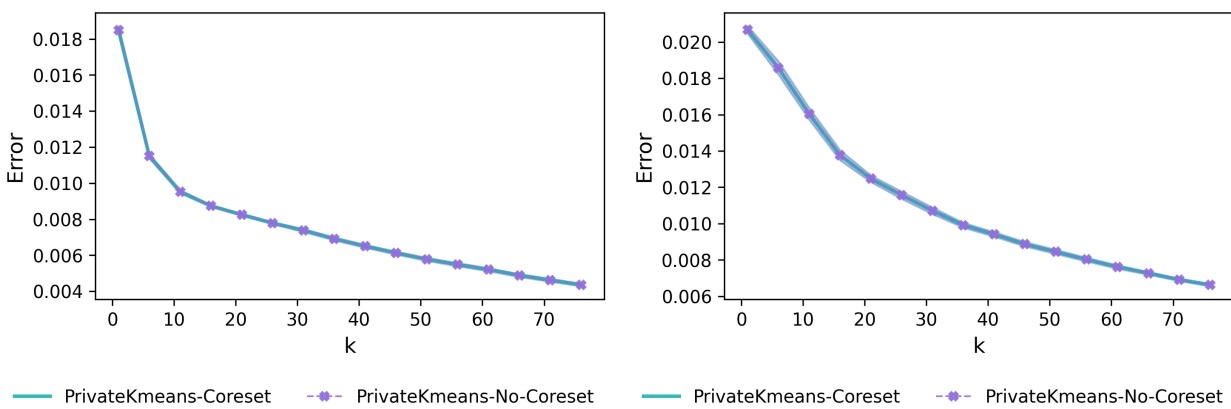

Figure 147: PrivateKmeans, Superconductivity: h → l   Figure 148: PrivateKmeans, Superconductivity: l → h

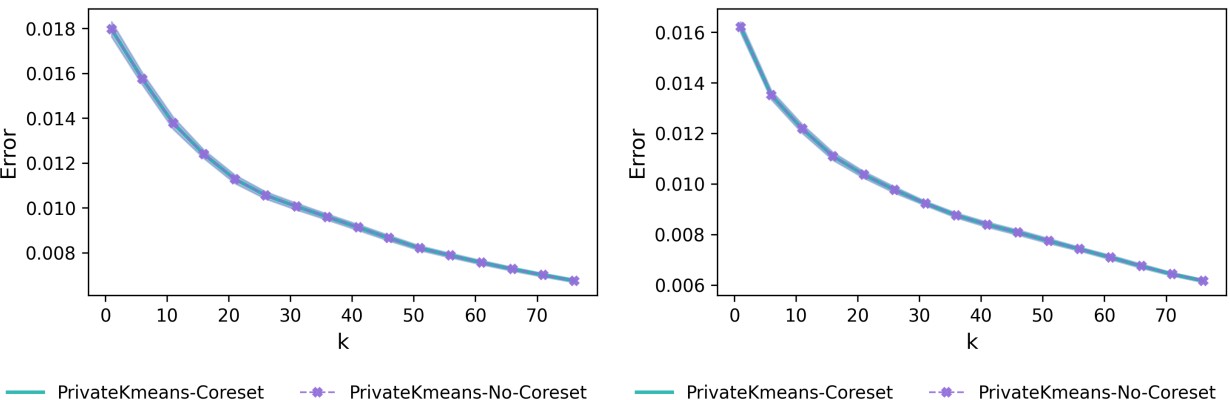

Figure 149: PrivateKmeans, Superconductivity: ml → h  Figure 150: PrivateKmeans, Superconductivity: h → ml

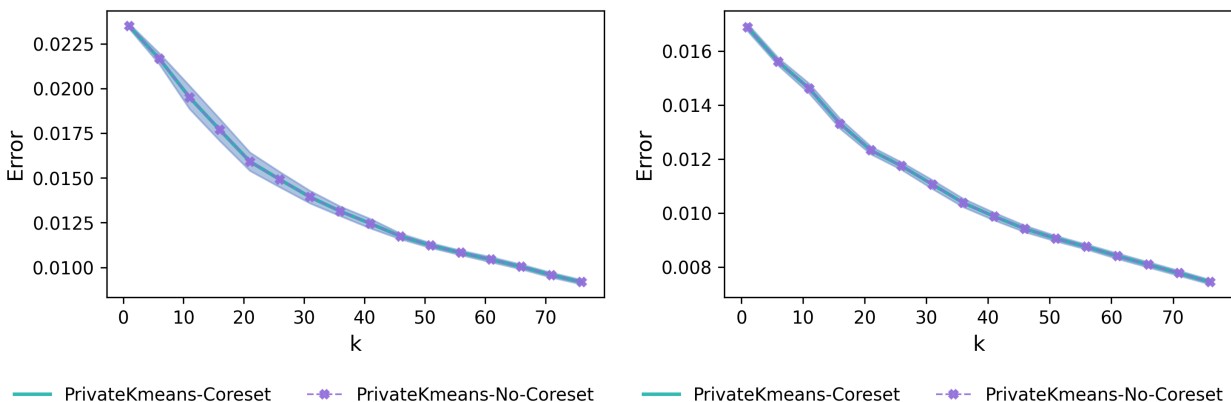

Figure 151: PrivateKmeans, Superconductivity: h → mh  Figure 152: PrivateKmeans, Superconductivity: mh → h

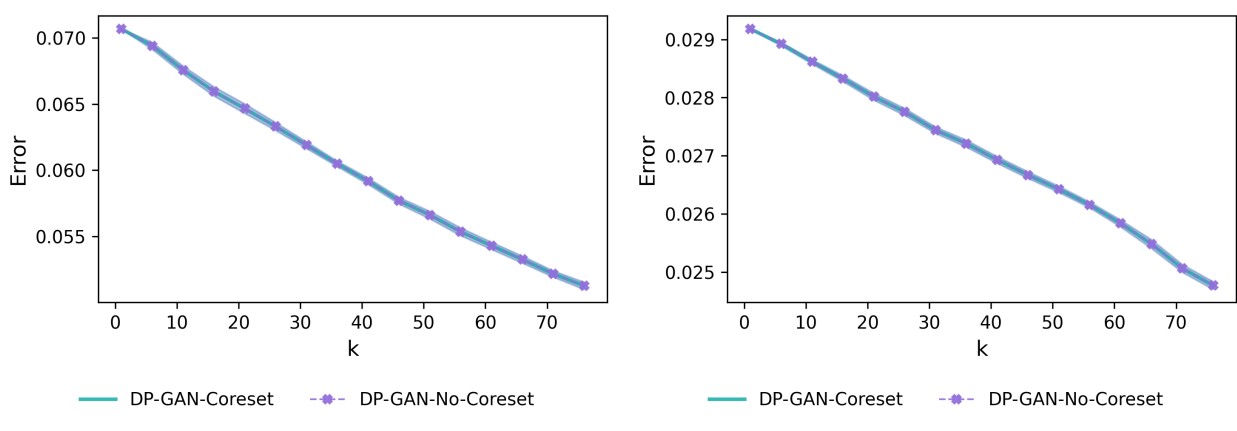

Figure 153: DP-GAN, MNIST: 7 → 1  Figure 154: DP-GAN, MNIST: 7 → 1

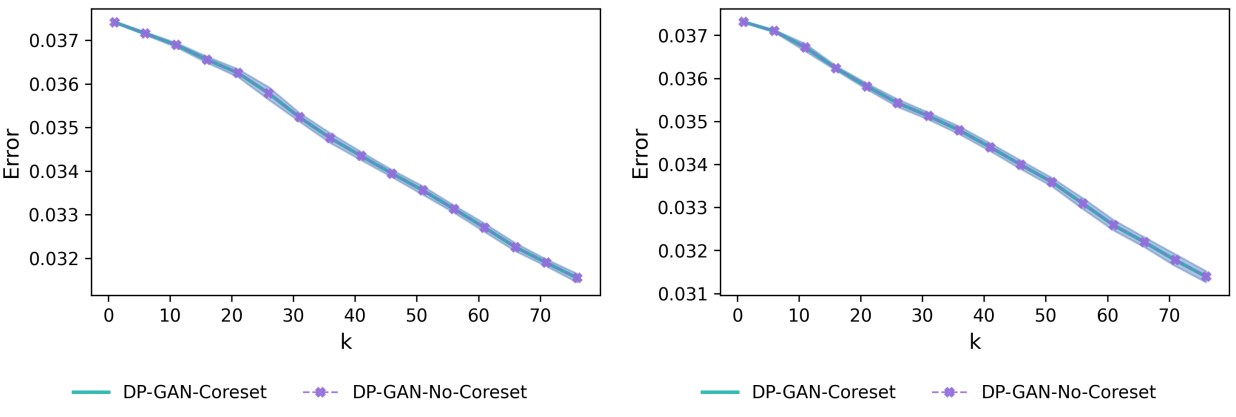

Figure 155: DP-GAN, MNIST: 6 → 7       Figure 156: DP-GAN, MNIST: 9 → 6

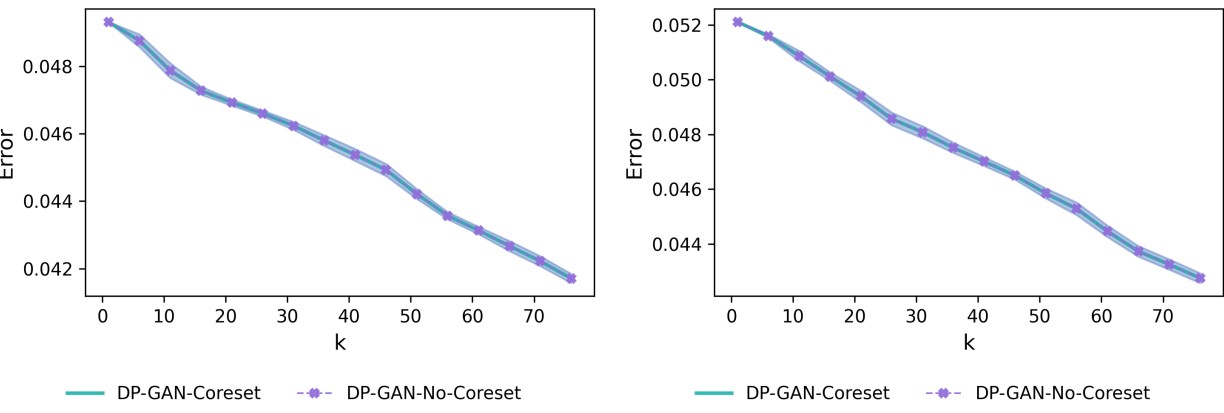

Figure 157: DP-GAN, MNIST: 2 → 5       Figure 158: DP-GAN, MNIST: 5 → 2

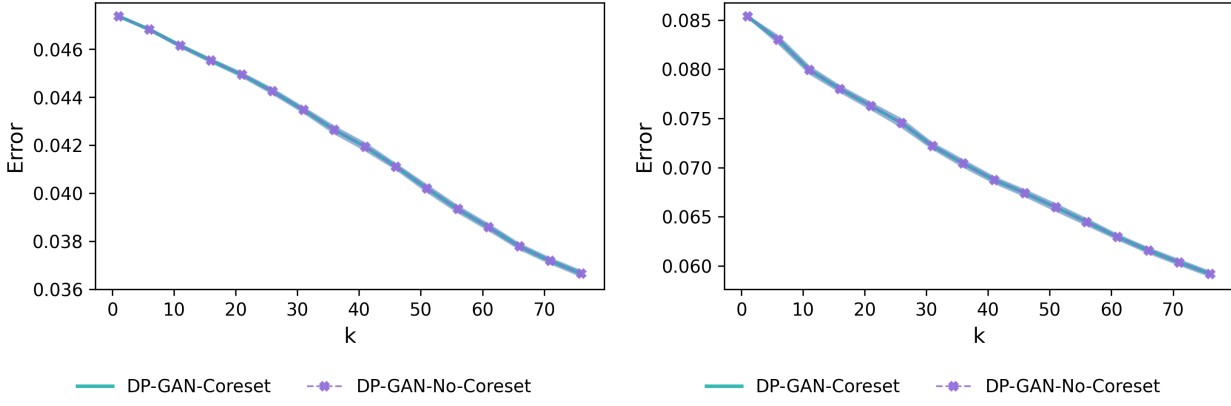

Figure 159: DP-GAN, Office: amazon → dslr       Figure 160: DP-GAN, Office: dslr → amazon

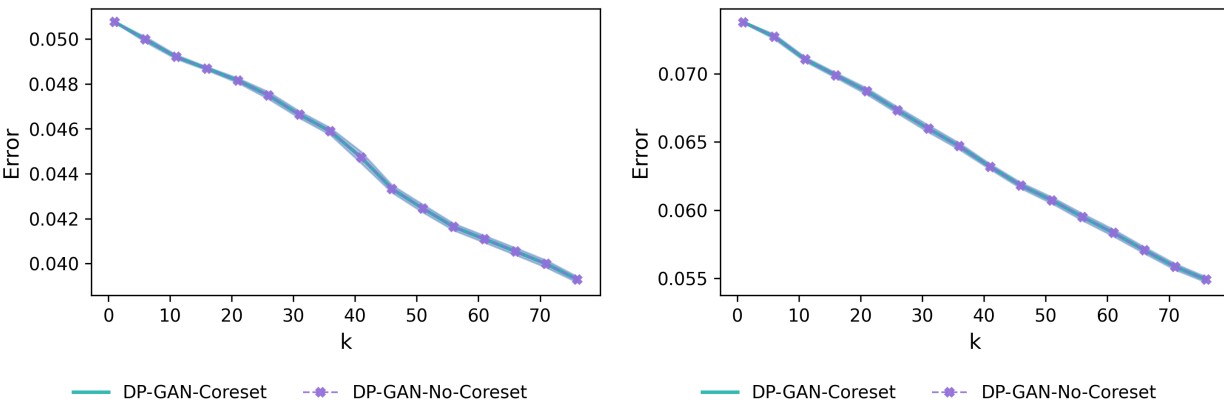

Figure 161: DP-GAN, Office: amazon → webcam

Figure 162: DP-GAN, Office: webcam → amazon

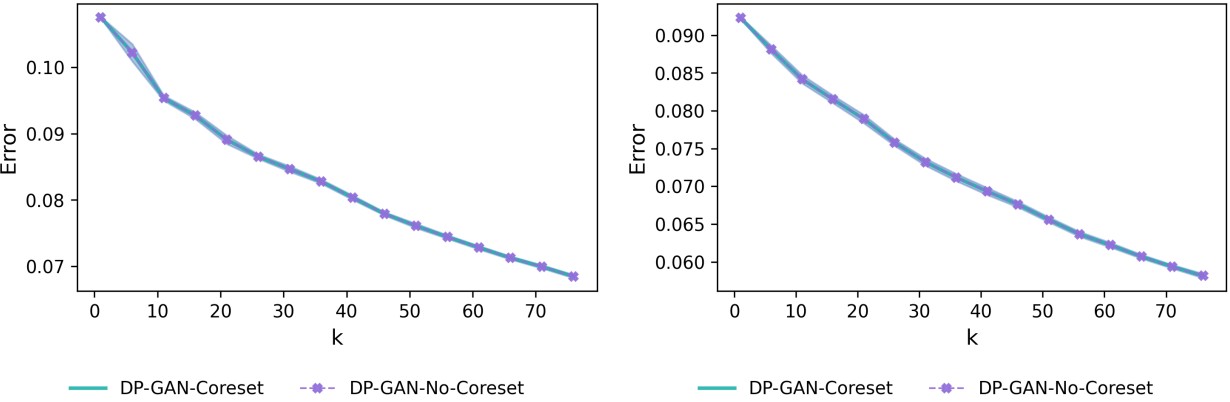

Figure 163: DP-GAN, Office: dslr → webcam

Figure 164: DP-GAN, Office: webcam → dslr

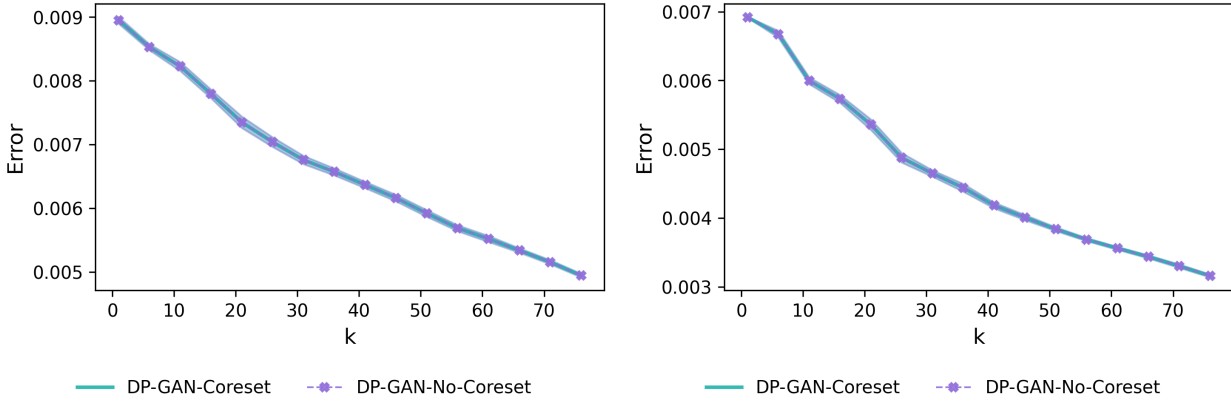

Figure 165: DP-GAN, Superconductivity: l → ml

Figure 166: DP-GAN, Superconductivity: ml → l

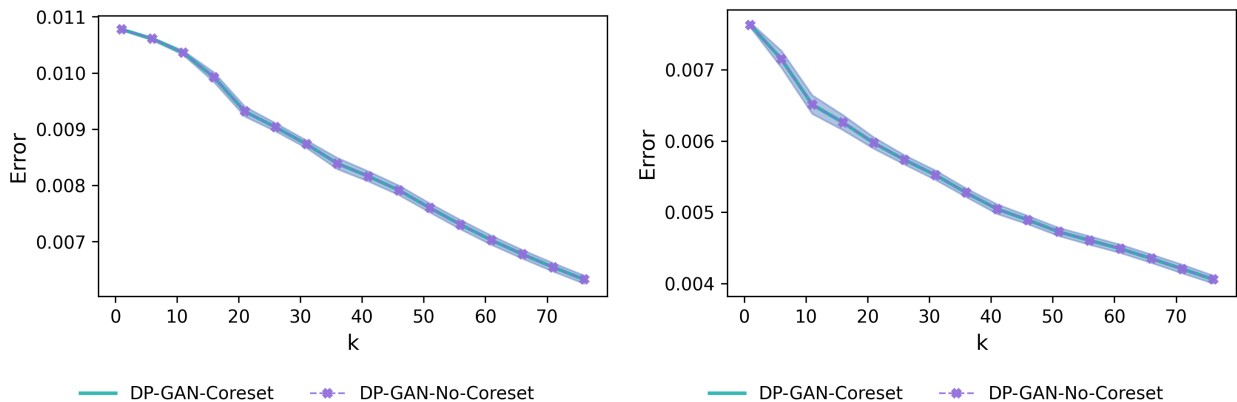

Figure 167: DP-GAN, Superconductivity: ml → mh

Figure 168: DP-GAN, Superconductivity: mh → ml

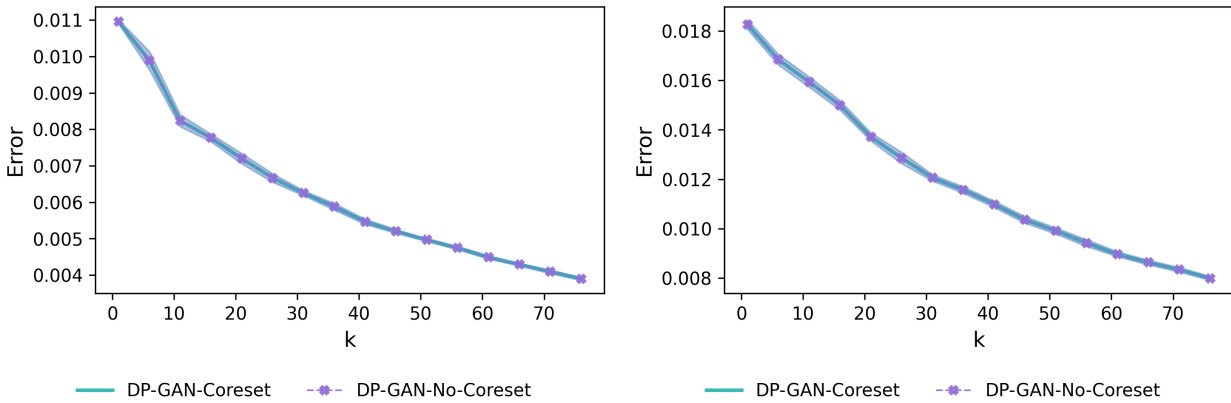

Figure 169: DP-GAN, Superconductivity: mh → l

Figure 170: DP-GAN, Superconductivity: l → mh

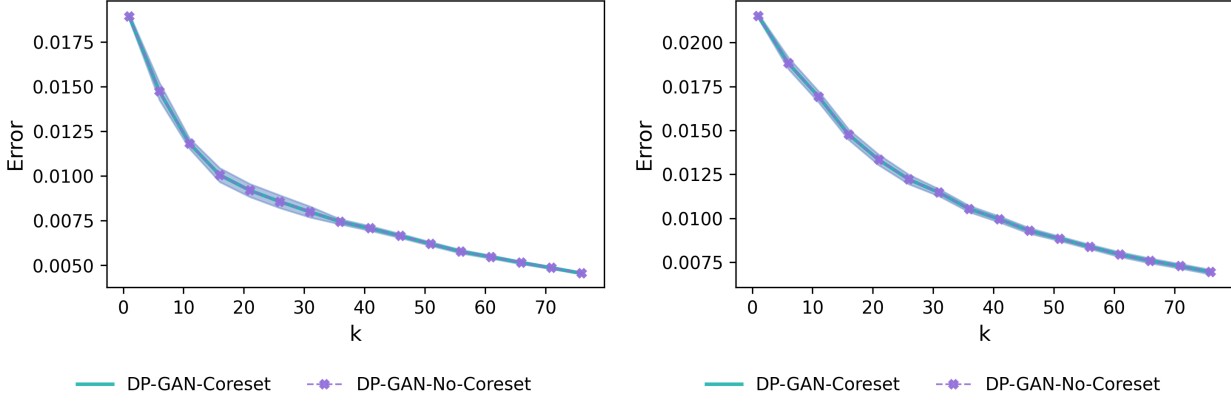

Figure 171: DP-GAN, Superconductivity: h → l

Figure 172: DP-GAN, Superconductivity: l → h

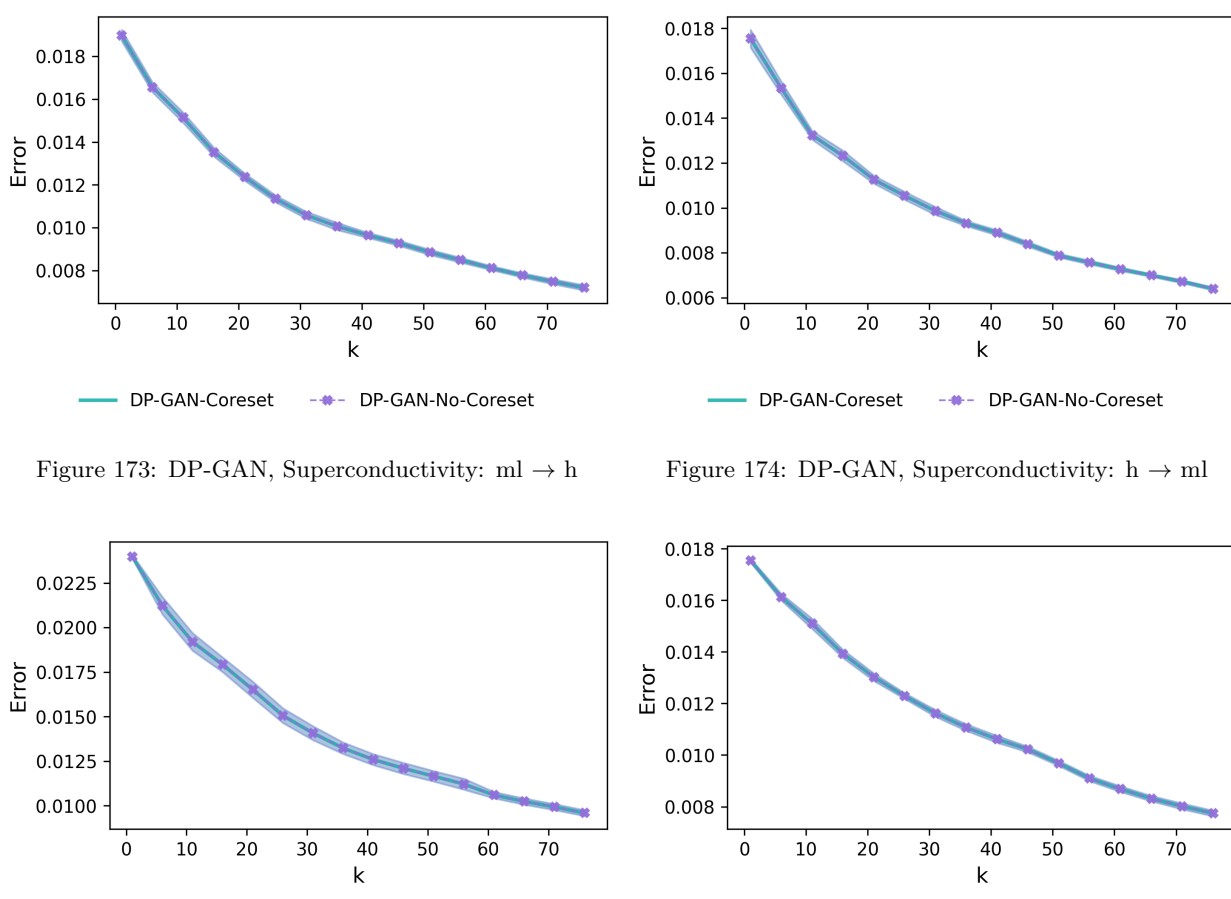

Figure 173: DP-GAN, Superconductivity: ml → h

Figure 174: DP-GAN, Superconductivity: h → ml

Figure 175: DP-GAN, Superconductivity: h → mh

Figure 176: DP-GAN, Superconductivity: mh → h

## C.7 Run time comparison

For each experiment, we report the average running times over 30 runs. All run times were measured when running on one core of an Intel i9-9900K CPU and NVIDIA GEFORCE RTX 2080 Ti GPU. It can be seen that there are no significant differences between the running times of the various algorithms.

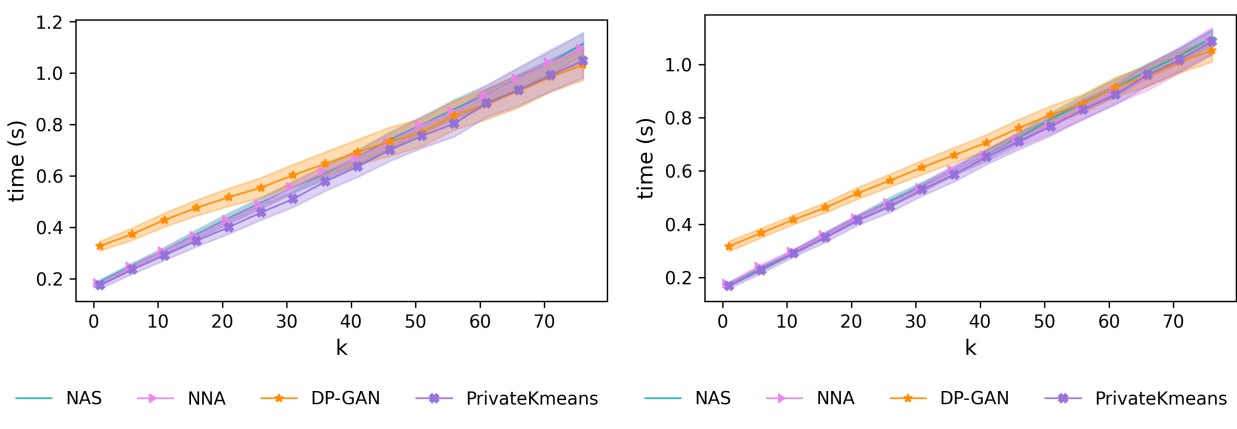

Figure 177: MNIST: 1 → 7

Figure 178: MNIST: 7 → 1

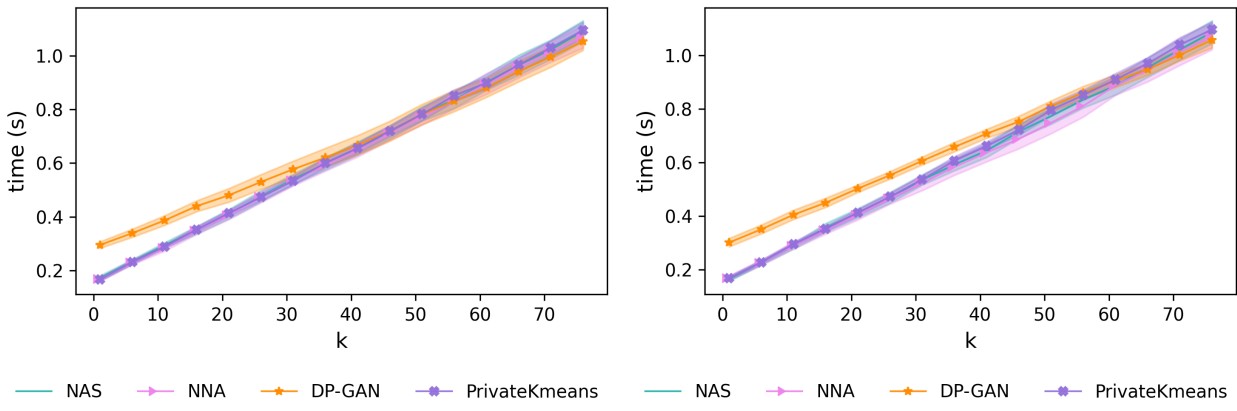

Figure 179: MNIST: 6 → 7

Figure 180: MNIST: 9 → 6

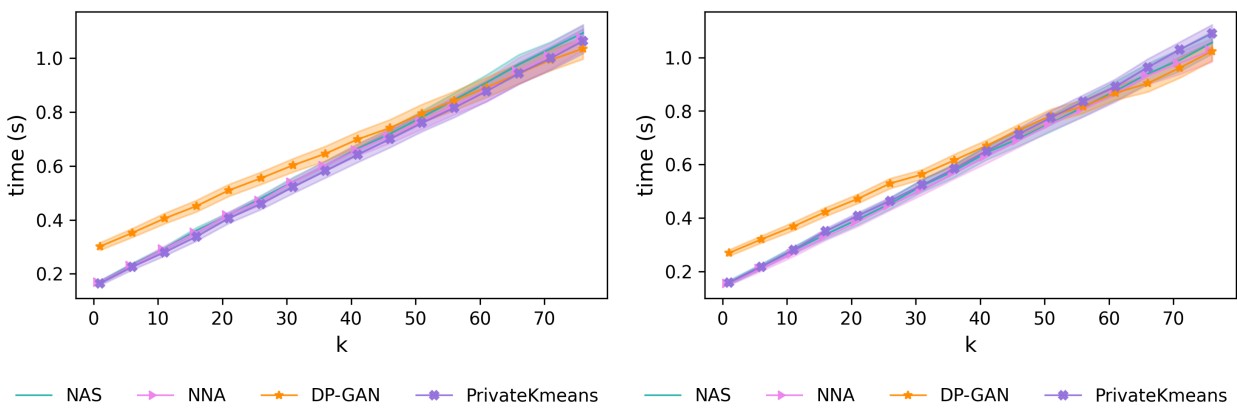

Figure 181: MNIST: 2 → 5

Figure 182: MNIST: 5 → 2

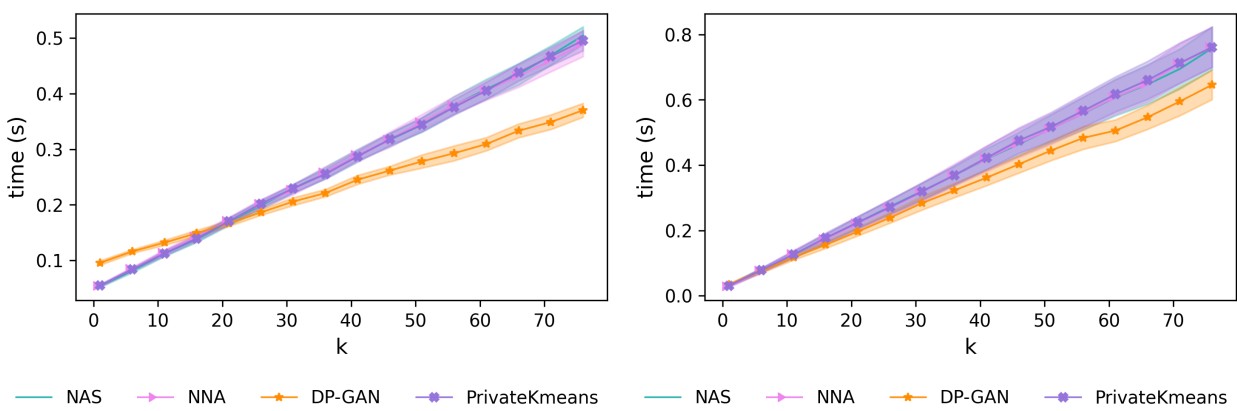

Figure 183: Office: amazon → dslr

Figure 184: Office: dslr → amazon

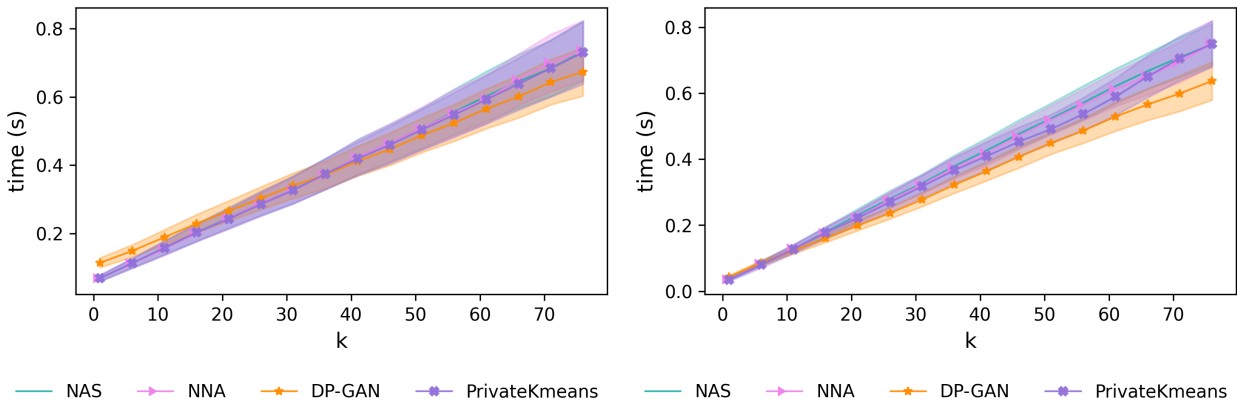

Figure 185: Office: amazon → webcam

Figure 186: Office: webcam → amazon

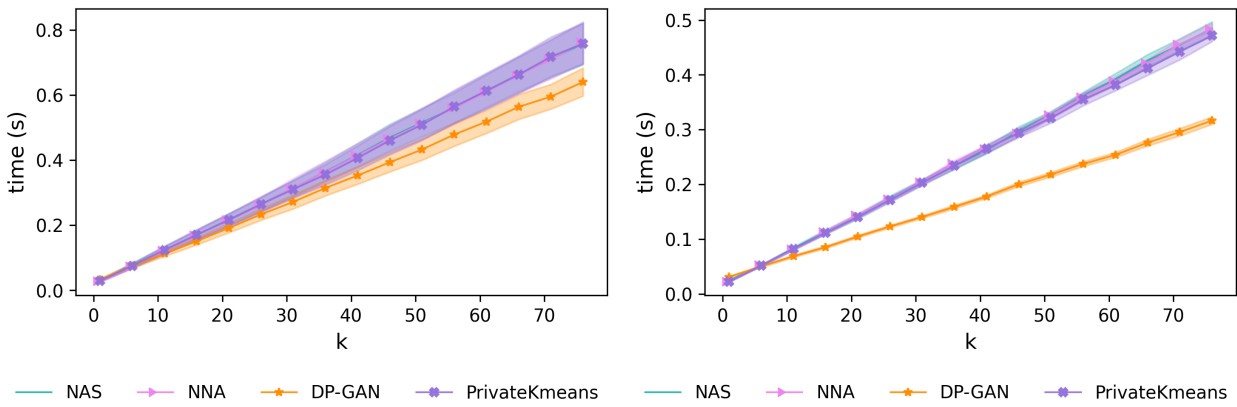

Figure 187: Office: dslr → webcam

Figure 188: Office: webcam → dslr

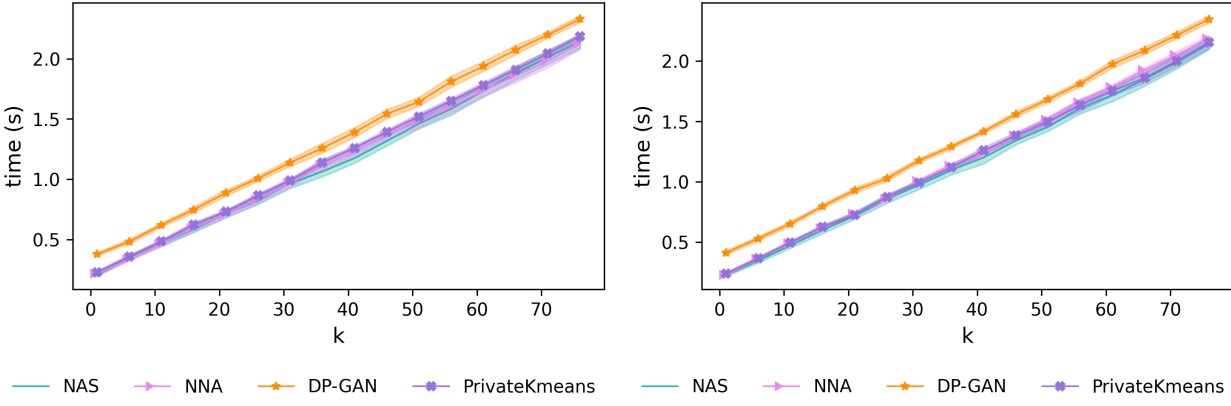

Figure 189: Superconductivity: l → ml

Figure 190: Superconductivity: ml → l

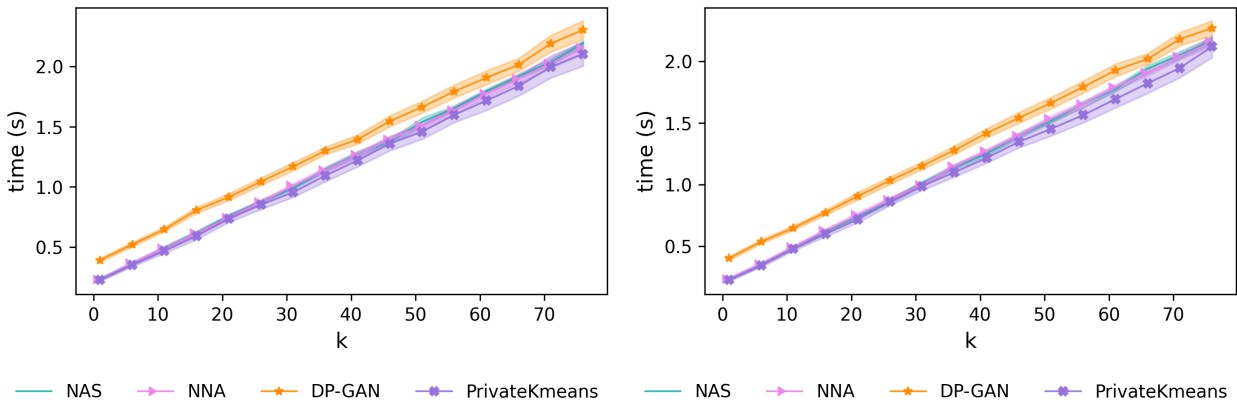

Figure 191: Superconductivity: ml → mh

Figure 192: Superconductivity: mh → ml

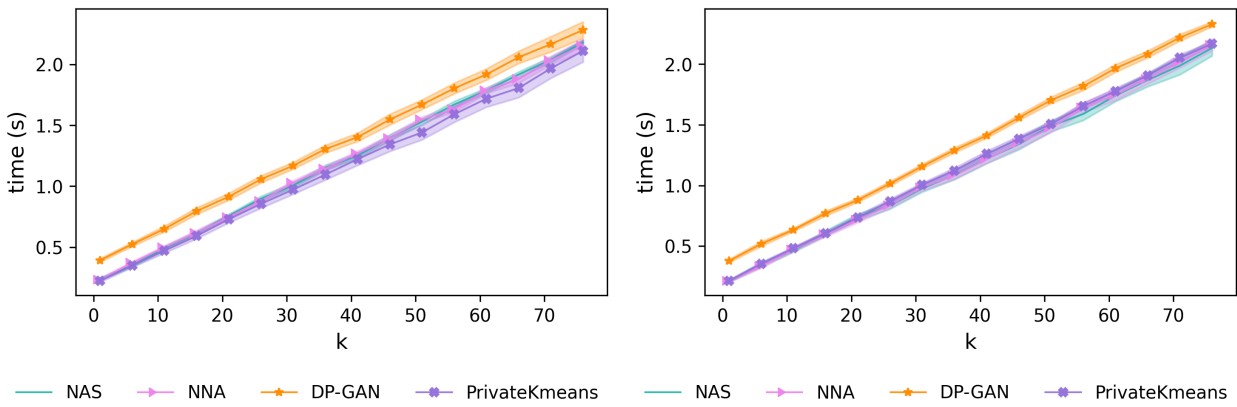

Figure 193: Superconductivity: mh → l

Figure 194: Superconductivity: l → mh

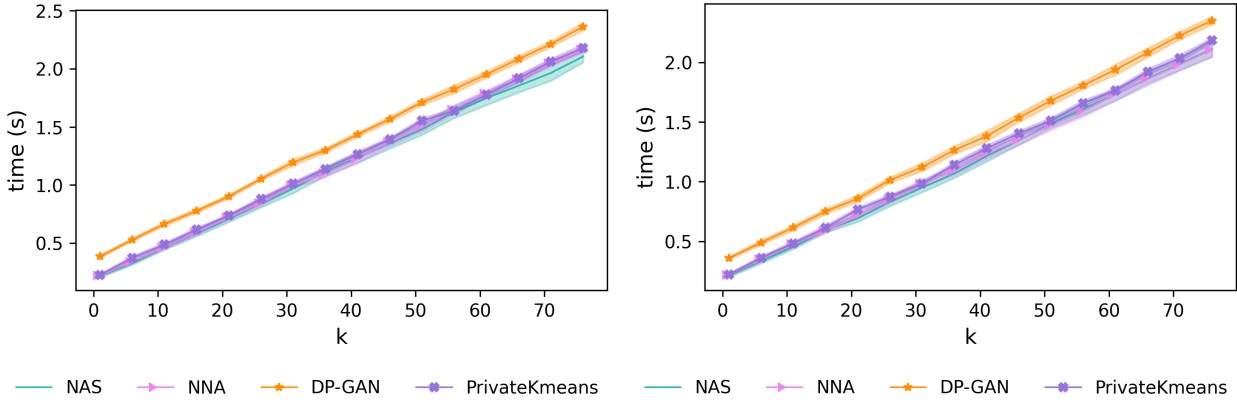

Figure 195: Superconductivity: h → l

Figure 196: Superconductivity: l → h

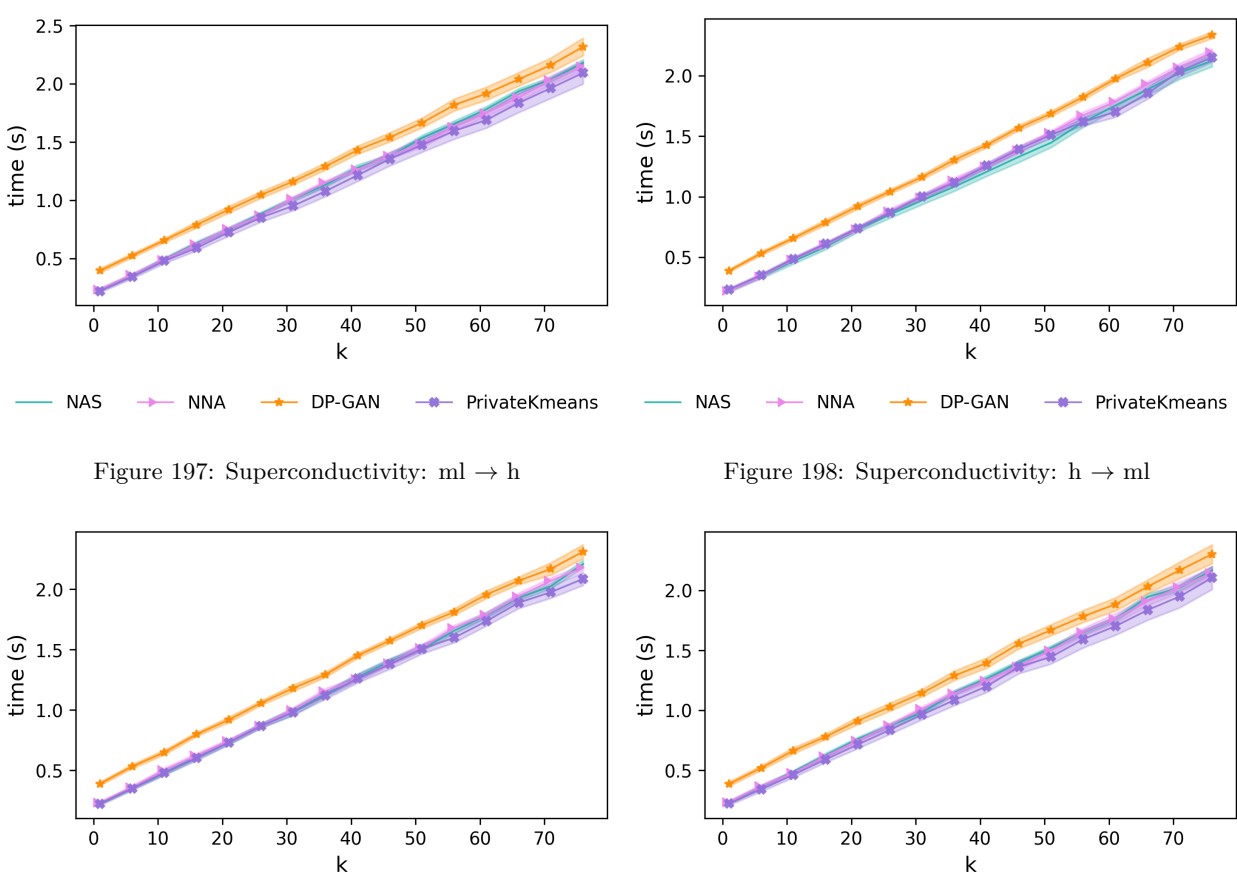

Figure 197: Superconductivity: ml → h

Figure 198: Superconductivity: h → ml

Figure 199: Superconductivity: h → mh

Figure 200: Superconductivity: mh → h

