# OpenReview forum: "Differentially Private Source-Target Clustering"
_TMLR — Accepted by TMLR_

### Review · Reviewer_GfwS · 2024-07-21

**Summary Of Contributions:**

The authors propose the problem of Private STC. In STC, a target dataset T that needs to be clustered by selecting centers, in addition to centers that are already provided in a separate source dataset S. They derive lower bounds for the private STC Cost objective and present a differentially private algorithm under a data-dependent analysis. The experiments show promising results.

**Audience:**

Yes

**Claims And Evidence:**

Yes

**Requested Changes:**

Please respond to the above questions. I would like to list the proposed changes once I the above have been answered.

**Strengths And Weaknesses:**

The authors have done a commendable job in their exposition of the idea, the rigor of the proofs, and the exhaustive experiments. The paper is well written, informative and the proofs (to the extent at which I have read them) seem correct. Nevertheless, I have a few questions and comments,

1. The STC problem and k-mediods in the introduction should be explained better. Also, from the introduction it seems that all points in S are "selected as centers" but that doesn't seem to be the case in section 5
2. I am not sure I understand the Social network use case. Are all promoting individuals in S alone and T only has the audience, and the point is to choose K members of the audience such that the Cost is minimized with out them individuals getting to know which promoter approached them? That seems like an impossible task. The remaining two uses cases seem fine and well motivated.
3. (For my understanding) Let us take the simple case of DP clustering. Two neighboring datasets can have different selection of centers. It can be easily shown that $f(X) - f(X')$ is $2/n$ for two neighboring datasets with the same selection of centers. And is it true that for a different selection of centers  $f(X) - f(X')$  could only get smaller?
4. $[m]$ notation is not defined.
5. Since the privacy guarantee comes from adding Laplace noise, I presume, you could have tried an arbitrary mechanism for M (and not just averaging or NNA). Did you try experimenting with any other methods ? Or if you did, was it the case that averaging gave the best results?
6. It might be useful to have some discussion around NNA for the (lack) of theoretical bounds. I guess the thresholding makes the analysis tricky?
7.  In section 6 starting line do you mean $(\epsilon,\delta)$?
8. What decides m and 100 for DP GAN and Private Kmeans?
9. I find it highly confusing that DP GAN and Private Kmeans are worse than ClusterT for almost all datasets. Does it mean that the datasets being used are not diverse enough? Since for a fixed value of k, one can create S such that ClusterT would fail. Please provide more insights here.
10. Please clarify the experiments for the real datasets. For instance for Office -- what are the source and target? Are all points in S being used as a cluster center or only a subset? And what are the figures in 3? Are they averaged runs over different datasets?

---

> ### Author Response · Authors · 2024-08-01
> **Thank you for your review (1)**
>
> Dear reviewer,
>
> Thank you for your thorough review and comments. Please find below our responses to your comments and questions. We have also uploaded a revision correcting identified issues. Please let us know if any more clarifications are needed.
>
>
> **Comment**: The STC problem and k-mediods in the introduction should be explained better.
>
> **Answer**: We would be glad to clarify any points you believe are not sufficiently clear in the current explanation.
>
>
> **Comment**: Also, from the introduction it seems that all points in S are "selected as centers" but that doesn't seem to be the case in section 5
>
> **Anwer**: Throughout the paper, all points in S are assumed to function as centers. This is reflected in the definition of the Cost function.
>
>
> **Comment**: I am not sure I understand the Social network use case. Are all promoting individuals in S alone and T only has the audience, and the point is to choose K members of the audience such that the Cost is minimized with out them individuals getting to know which promoter approached them? That seems like an impossible task. The remaining two uses cases seem fine and well motivated.
>
> **Anwer**: This scenario assumes that during the recruitment stage, additional promoters from T should be approached without leaking the identify of existing promoters from S. The actual promotion would take place at a later stage and may reveal the identities of all promoters.
>
>
> **Comment**: (For my understanding) Let us take the simple case of DP clustering. Two neighboring datasets can have different selection of centers. It can be easily shown that  𝑓(𝑋)−𝑓(𝑋′) is 2/𝑛 for two neighboring datasets with the same selection of centers. And is it true that for a different selection of centers 𝑓(𝑋)−𝑓(𝑋′) could only get smaller?
>
> **Anwer**: If 𝑓  is the clustering Cost function, It is not always true that 𝑓(𝑋)−𝑓(𝑋′) equals 2/𝑛​ for two neighboring datasets with the same selection of center. For example, if we remove a point x that was very close to a center c such D(x, c) << 2/n  then Cost( X’{x}) -Cost(X’) << 2/n. Therefore, it is not always true that for a different selection of centers, 𝑓(𝑋)−𝑓(𝑋′)  could only get smaller.
>
> It is important to note that in the problem of private STC where the source dataset S is private, removing one point is equivalent to removing a center, which can dramatically change the cost if the solution relied on that center.  We explain this in more detail in Section 4. We would be glad to clarify further if needed.
>
>
> **Comment**: [m] is not defined.
>
> **Anwer**: [m] is the set of integers between 1 and m. We added a formal definition to the paper.
>
>
> **Comment**: It might be useful to have some discussion around NNA for the (lack) of theoretical bounds. I guess the thresholding makes the analysis tricky?
>
> **Anwer**: The purpose of NNA is to provide a practical algorithm for the STC problem, and it is highly motivated by our theoretical algorithm, NAS. The difficulty of deriving a theoretical guarantee for this algorithm stems from the variation that makes it more practical: we use each private point  from S only once to avoid using composition, which means we can’t ensure that we will have enough points to address each point from T. Therefore, we can use $\epsilon$ as a private parameter for M instead of $\epsilon$ / √n (much more accurate). We discuss this in subsection 5.2.
>
>
> **Comment**: In section 6 starting line do you mean  (𝜖,𝛿)?
>
> **Anwer**: NNA and the baseline algorithms are 𝜖-DP (𝛿=0). To have a fair comparison, we implemented the NAS algorithm as 𝜖-DP, which required changing the advanced composition that we used in Theorem 5.3 to basic composition.
>
> **Comment**: What decides m and 100 for DP GAN and Private Kmeans?
>
> **Anwer**: m is the size of the source data set, which is the size of the data set for all algorithms.
> On the last paragraph of page 10, we had a typo where, where we wrote m instead of 100 as the number of samples from the distribution learned by DP-GAN on S. We have fixed this typo in the revision. We similarly used 100 for PrivateKmeans, as specified on the first paragraph of page 11.
>
> **Comment**: I find it highly confusing that DP GAN and Private Kmeans are worse than ClusterT for almost all datasets. Does it mean that the datasets being used are not diverse enough? Since for a fixed value of k, one can create S such that ClusterT would fail. Please provide more insights here.
>
> **Anwer**: The weaknesses of Private K-means and DP-GAN lie in their attempts to map the inter-source dataset S. This can be observed in Figure 2. In Synthetic 2, Private K-means and DP-GAN invest much of the budget in clustering the dense cluster in the top image. However, these clusters are not relevant to T at all. Allocating the privacy budget across the inter-source dataset cloud heavily impacts the centers provided by the algorithms and affects the overall clustering cost.

---

> ### Author Response · Authors · 2024-08-01
> **Thank you for your review (2)**
>
> Dear reviewer,
>
> Please find below responses to the rest of your comments.
>
> **Comment**: Please clarify the experiments for the real datasets. For instance for Office -- what are the source and target?
>
> **Anwer**: The Office dataset includes images of three different types: "amazon", "webcam", and "dslr." We ran experiments for each possible assignment of on of these types to source and target, leading to experiments such as amazon->webcam, dslr->amazon, etc. All of the different combinations of source and target are reported in the appendix. In the main paper (Figure 3), one of the pairs is presented as an example, as specified in the caption of Figure 3. More detailed information on the datasets can be found in Section 6, in the last two paragraphs.
>
> **Comment**: Are all points in S being used as a cluster center or only a subset?
>
> **Anwer**: Indeed, each point from S is selected as a center. This is described in the problem statement and the introduction.
>
> **Comment**: And what are the figures in 3? Are they averaged runs over different datasets?
>
> **Anwer**: Each sub-plot in figure 3 describes results for a different dataset. The name of the dataset appears in the title of the subplot. The caption of Figure 3 provides more information about each experiment.

---

### Review · Reviewer_bz69 · 2024-08-12

**Summary Of Contributions:**

In this paper the authors study  Source-Target Clustering (STC). In this problem, one is given a source set S and target set T, and wants to find k points T_k in T that, together with predetermined centers from S, minimizes the average distance between each point in T and S u T_k. Here, the authors pose a differentially private (DP) version of this problem where the privacy of S needs to be preserved. The authors derive two algorithms for the DP-STC problem and prove different guarantees on the cost of these algorithms. The authors then demonstrate that their algorithms have improved performance over non-private algorithms that ignore S and DP algorithms that do not use the structure of the problem.

**Audience:**

Yes

**Claims And Evidence:**

Yes

**Requested Changes:**

Overall I liked this paper and would be happy to see it accepted in its current state. I only have very minor comments.

- Page 4, “let T be a dataset that includes” → “let T be a dataset that contains”
- It might be nice to have a glossary for the different notations, eg s_x vs s^i_x vs \triangle … it can be hard to keep track
- Lemma 5.6: I don’t understand how to get the inequality \triangle(x, c_x + Z) ≤ \triangle(x, c_x) + ||Z||_2. The triangle inequality gives \triangle(x, c_x + Z) ≤ \triangle(x, c_x) + \triangle(x, Z).
- Lemma 5.6: “Laplace distribution with magnitude …” — the formula has an extra parenthesis.
- Is it possible to modify NNA algorithm to also be (\epsilon,\delta)-DP for \delta>0?
- I thought the Appendix B result on coresets is quite important and could be moved to the main text. Maybe the authors could combine Lemmas 5.6+5.7 if there is not enough space.
- Why are there big jumps for some of the ClusterT lines discrete in top row of Figure 3? (eg in Synthetic 1)

It also would be interesting to generalize synthetic simulation 1 and see what the cost is as a function of the overlap between S and T. In particular what happens when S and T do not overlap at all? Do NNA/NAS  still have low cost? This is a suggested change but not necessary for paper acceptance.

**Strengths And Weaknesses:**

Strengths

- Paper is well-written and easy to follow.
- NNA and NAS algorithms proposed by authors are natural and the theoretical guarantees are sound.
- Real data evaluation is convincing

Weaknesses

- Minor typos/comments below
- Minor improvement to real data analysis (see below)

---

> ### Author Response · Authors · 2024-08-19
> **Thank you for your review**
>
> Dear reviewer,
>
> Thank you for your comments. We have uploaded a revision addressing your suggestions. Please see our responses to your comments below.
>
> **Comment**: Page 4, “let T be a dataset that includes” → “let T be a dataset that contains”
>
> **Answer**: We corrected all instances of this misake in the revision.
>
> **Comment**: It might be nice to have a glossary for the different notations, eg s_x vs s^i_x vs \triangle … it can be hard to keep track
>
> **Answer**: Thank you for the suggestion. We will consider this for the final version, after taking into account all reviews and space considerations.
>
> **Comment**: Lemma 5.6: I don’t understand how to get the inequality \triangle(x, c_x + Z) ≤ \triangle(x, c_x) + ||Z||_2. The triangle inequality gives \triangle(x, c_x + Z) ≤ \triangle(x, c_x) + \triangle(x, Z).
>
> **Answer**: We used the triangle inequality as follows:
> $\Delta(x, c_x +Z) ≤ \Delta(x, c_x) + \Delta(c_x, c_x + Z)$.
> By the definition of \Delta as the Euclidean distance, $\Delta(c_x, c_x + Z) = \|Z\|_2$.
> This gives the resulting inequality. We have added an extra line to the derivation to clarify this.
>
>
> **Comment**: Lemma 5.6: “Laplace distribution with magnitude …” — the formula has an extra parenthesis.
>
> **Answer**: This was corrected in the revision.
>
> **Comment**: Is it possible to modify NNA algorithm to also be ($\epsilon$,$\delta$)-DP for $\delta$>0?
>
> **Answer**: Since NNA is $\epsilon$-DP, it is also by definition ($\epsilon$, $\delta$)-DP. It is possible to further weaken the privacy of NNA to "true" ($\epsilon$, $\delta$)-DP by outputting the original S with probability $\delta$, and running the existing NNA with probability $1-\delta$.
>
> **Comment**: I thought the Appendix B result on coresets is quite important and could be moved to the main text. Maybe the authors could combine Lemmas 5.6+5.7 if there is not enough space.
>
> **Answer**: We agree that the result on coresets in Appendix B is significant. We will consider moving it to the main paper body for the final version, after taking into account all reviews and space considerations.

---

### Review · Reviewer_rV2B · 2024-10-30

**Summary Of Contributions:**

This paper considers a new private variant of the Source-Target Clustering (STC) setting. In STC, there is a target dataset that needs to be clustered by selecting centers, in addition to centers in a source dataset. The goal is to select centers from the target, such that the target clustering cost given the additional source centers is minimized. They consider private STC, in which the source dataset is private. They derive lower bounds for the private STC, and present a differentially private algorithm with asymptotically advantageous results under a data-dependent analysis. By experiments, the reduction in clustering cost is obtained compared to baseline approaches.

**Audience:**

Yes

**Claims And Evidence:**

Yes

**Requested Changes:**

Please see weakness. To check the algorithms and re-design the experiments.

**Strengths And Weaknesses:**

Strengths:
1. They give a lower bound for this algorithm.
2. They give 2 algorithms with theoretical proofs and experiments implementation.

Weakness:
1. For Alg 1, authors use a Laplace mechanism M to get a private dada point $s_x'$, but how can Laplace mechanism M achieve that? This (including the noise parameters) is not shown clearly by the pseudo code of Algorithm 1.
2. For Theorem 5.3, since $f: R^d \to R^d$ and $M: R^d \to R^d$, why the L1-sensitivity is  $\frac{\sqrt{d}}{t}$ instead of $\frac{d}{t}$?
3. For Algorithm 2, Step 3, the implement of Laplace Mechanism M is also not clearly, also the problem of sensitivity.
4. For experiments, the design and implement of the real dataset is not reasonable, such as for MINST, use (1,7) as (source, target). Why use 1 to cluster 7?

---

> ### Author Response · Authors · 2024-11-07
> **Response to review**
>
> Dear reviewer,
>
> Thank you for your comments. We have uploaded a revision addressing the your review as well as the other recent review. Please note that in the revision some theorem numberings have changed. Please see our responses to your comments below.
>
> **For Alg 1, authors use a Laplace mechanism M to get a private dada point , but how can Laplace mechanism M achieve that? This (including the noise parameters) is not shown clearly by the pseudo code of Algorithm 1.**
>
> We present Algorithm 1 in general terms, starting with a general differential privacy (DP) mechanism M. Following the algorithm, we explained how M can be implemented using the Laplace mechanism, by adding Laplace noise to each coordinate of c_x (see page 6) and then continue the proof accordingly. Following the suggestion of another reviewer, we have decided to present the analysis for Alg. 1 using the Gaussian mechanism instead of the Laplace mechanism, so as to provide improved guarantees. The updated analysis is provided in Section 5.1.
>
> **why the L1-sensitivity is $\sqrt{d}/t$ instead of $d/t$?**
>
> Recall that we assume that $\Delta(x, x') \equiv ||x -x'||_2 \leq 1$ for any two data set points. The $L_1$ change in the average when changing a single point is at most $||x-x||_1/t \leq \sqrt{d}||x-x'||_2/t \leq \sqrt{d}/t$.
> This can be contrasted with some DP settings in which the L_infinity norm is bounded and not the L_2 norm, leading to a worse bound of d/t. Our use of the L_2 norm bound is motivated by our clustering application, in which it is standard to use the L_2 norm.
> We have added the derivation details to the proof (now in Theorem 5.10).
>
> For Algorithm 2, Step 3, the implement of Laplace Mechanism M is also not clearly, also the problem of sensitivity.
> The Laplace Mechanism M is implemented in step 5 of Alg 2. The parameters of the implementation are defined in the statement of Theorem 5.11. The Laplace mechanism is defined right before Lemma 5.9.
>
> **For experiments, the design and implement of the real dataset is not reasonable, such as for MINST, use (1,7) as (source, target). Why use 1 to cluster 7?**
>
> The purpose of the (source,target) experiments is to study the behavior of the algorithm when some representative examples from the target data are not well clustered using the source data, which is the main scenario that this work addresses. For instance, if one want to cluster images for digits 1 and 7 together, the algorithm needs to decide which centers from the 7 data set to add, so as to achieve a low clustering cost while preserving differential privacy of the source data (the digit 1 data set). We report experiments on several different pairs of digits in the appendix, as well as other (source,data) pairs from other data sets. The pair (1,7) is presented in the body of the paper as one example of these experiments.

---

### Review · Reviewer_15gL · 2024-10-31

**Summary Of Contributions:**

This paper introduces the differentially private variant of the Source-Target Clustering (STC) problem. In this problem there is a target dataset and a (private) source dataset, and the goal is to output a clustering for the target set such that the cost of clustering for the target set given the additional centers in the source dataset is minimized. Authors show that achieving worst-case guarantees is impossible through strong lower bounds and then propose algorithms that have data-dependent guarantees. The main idea behind their DP algorithms is to output a DP variant of the source set using standard techniques and then running a non-DP STC algorithm on the target set and the DP source set.

**Audience:**

Yes

**Claims And Evidence:**

Yes

**Requested Changes:**

1. If possible, please give an Informal version of your results in Section 1. What exactly is the lower bound? As a reader, it is annoying to scroll to section 4 to see your main results.
2. Please add important refs for DP Clustering e.g.,
Differentially Private Combinatorial Optimization Anupam Gupta, Katrina Ligett, Frank McSherry, Aaron Roth, Kunal Talwar;
Differentially Private Clustering: Tight Approximation Ratios, Badih Ghazi, Ravi Kumar, Pasin Manurangsi.

3. In Section 5.1, can you justify your use of Laplace noise in the approx DP algorithm? Why not directly use the Gaussian mechanism?
4. Please avoid using asymptotic notation for your DP parameters.
5. In Algorithm 2, why are you not using Sparse Vector Technique for Line 6? This could give you some gains in DP composition.

**Strengths And Weaknesses:**

Strengths:
Most of the presentation is clean and well-written. The problem is interesting and well-motivated by the authors in the private setting.

Weaknesses:

The related work on DP Clustering is incomplete, see my comments in Requested Changes.

I am not sure about some of the claims in the paper, see my comments below:

1. In Section 5.1, it is not clear to me as to why you are using Laplacian noise for approximate DP when we typically use Gaussian noise as this provides better utility since we add noise proportional to L2 sensitivity vs L1 sensitivity.
2. In the proof of Theorem 5.3: why is the L1 sensitivity of M $\sqrt(d)/t$. Shouldn’t it be $d/t$? It is also not good practice to write DP guarantees in terms of asymptotics as constants matter for the DP parameters, e.g., your DP claim for each activation of M is written in terms of Big Theta.
3. From my understanding, the only DP step in your Algorithm 1 is computing the average of points. This problem is not new, it can be easily solved using the Gaussian mechanism and one can even relax the dependency on the diameter of the space by using results from this paper:
K. Nissim, U. Stemmer, and S. P. Vadhan. Locating a small cluster privately. In Proceedings of the 35th ACM SIGMOD-SIGACT-SIGAI Symposium on Principles of Database Systems, PODS 2016, San Francisco, CA, USA, June 26 - July 01, 2016, pages 413–427, 2016.
So apart from applying the Noisy averaging to your specific problem, I am not sure what the technical novelty is here.

4. I don’t understand how you are obtaining your pure DP algorithm in Section 5.2 and how it can be (theoretically) better in terms of utility than the algorithm in Section 5.1. Intuitively, for the pure DP mechanism you have to add noise proportional to L1 sensitivity which is more than L2 sensitivity. Can you please clarify this part?

---

> ### Author Response · Authors · 2024-11-07
> **Response to review**
>
> Dear reviewer,
>
> Thank you for your comments. We have uploaded a revision addressing the your review, as well as the other recent review. Please note that in the revision some theorem numberings have changed. Please see our responses to your comments below.
>
> **Responses to numbered comments:**
>
> 1. Regarding the selection of mechanism: Our algorithms are stated in general terms, and any mechanism, including the Gaussian mechanism, can be plugged in, resulting in appropriate bounds. Since our ultimate goal in this paper was to provide an algorithm which works well in practice, we originally preferred to present Section 5 using only the Laplacian Mechanism for ease of presentation, since it allows us to provide pure DP in Alg 2, unlike the Gaussian mechanism. We used the Gaussian mechanism in Section 6 for zCDP. Nonetheless, upon further consideration, we have followed your suggestion to provide the analysis of Alg. 1 for the Gaussian mechanism instead of the Laplace mechanism, so as to provide improved utility guarantees for this algorithm. The presentation and analysis in Section 5.1 were updated accordingly.
>
> 2. The L_1 sensitivity is $\sqrt{d}/t$ because we assume that $\Delta(x, x') \equiv ||x -x'||_2 \leq 1$ for any two data set points. The $L_1$ change in the average when changing a single point is at most $||x-x'||_1/t \leq \sqrt{d}||x-x'||_2/t \leq \sqrt{d}/t$. This can be contrasted with some DP settings in which the L_infinity norm is bounded and not the L_2 norm, leading to a worse bound of d/t. Our use of the L_2 norm bound is motivated by our clustering application, in which it is standard to use the L_2 norm. We have added the derivation details to the proof (now in Theorem 5.10).
>
> Asymptotics: we have rephrased the results to avoid asymptotics and provide explicit constants.
>
> 3. This paper presents a new setting for clustering with differential privacy, which has not studied in previous work. Nissim et al. study a different setting, of clustering when the entire data set is private. Our main contributions, in addition to defining the new STC setting, is that we show how to solve it using differential privacy tools, derive both lower bounds and upper bounds, and provide a practical and useful differentially private algorithm. We further show in experiments that our new algorithms for the STC problem work better in the STC setting than the private k-means algorithm, designed for the private clustering problem of Nissim et al.
>
> 4. We achieve pure DP in Alg 1 by using the Laplace mechanism and weak composition. In Alg 2, we also use the Laplace mechanism. In addition, in Alg 2. we avoid composition altogether, by using every data point at most once for each center.
> Regarding the utility bounds, indeed the pure DP variant of Alg 1 (theorem 5.10) is not better in terms of utility than its approximate DP variant (theorem 5.5). This is discussed in the paper after the proof of theorem 5.10. We present this result to motivate our Alg 2, for which the noise is considerably smaller. Alg 2 is a practical algorithm with significantly better empirical results, as our experiments demonstrate.
>
> **Responses to requested changes:**
>
> 1. The lower bounds show that in the worst case, the error could strongly depend on the number of clusters, where fewer clusters lead to a larger worst-case error. We have added this informal explanation to our summary of contributions in Section 1.
>
> 2. We have added the references to the Related Work (Section 2).
>
> 3. Please see our answer above to comment 1. We have revised the manuscript so that the analysis in Section 5.1 uses the Gaussian mechanism.
>
> 4. We have rephrased the results to avoid asymptotics and provide explicit constants.
>
> 5.  In Alg 2, we do not use composition at all, since the algorithm uses every data point at most once for each center. Therefore, we do not need the Sparse Vector Technique.

---

### Review · Reviewer_67CD · 2024-12-06

**Summary Of Contributions:**

This paper introduces a novel framework for Differentially Private Source-Target Clustering. In this setting, the target dataset is clustered by leveraging centers from an existing source dataset while ensuring the source dataset is protected under the rigorous guarantees of differential privacy .

The primary contributions of this work are as follows:
1. It proposes a novel approach to address the STC problem under differential privacy constraints.
2. It establishes a theoretical lower bound for the problem

**Audience:**

Yes

**Broader Impact Concerns:**

There are no broader impact concerns.

**Claims And Evidence:**

Yes

**Requested Changes:**

I agree to pass the paper as it is.

**Strengths And Weaknesses:**

Strengths
It is well defined and neatly structured on an interesting topic. It is also important to provide theoretical guarantees for the proposed algorithm.

Weaknesses
The part where the Laplace mechanism was applied was my main question, but it was resolved by the answers to the other reviewers' comments.

---

### Decision · Action_Editor_gYLV · 2025-01-17

**Recommendation:** Accept as is

**Comment:**

This paper introduces a novel differentially private variant of the Source-Target Clustering (STC) problem, where the goal is to cluster a target dataset while preserving the privacy of a source dataset. The authors prove strong lower bounds showing the impossibility of achieving worst-case guarantees and propose two data-dependent algorithms (NNA and NAS) with theoretical proofs and real-data experiments demonstrating their effectiveness. These algorithms use a differentially private representation of the source dataset combined with standard STC techniques to achieve reduced clustering costs compared to non-private and baseline methods. The paper is well-motivated, clearly written, and provides sound theoretical and empirical results, making it a valuable contribution to the study of privacy-preserving clustering.

**Audience:**

The paper is relevant to differential privacy researchers.

**Claims And Evidence:**

The claims made in the paper were found to be accurate and evidence was provided. All the reviewers who raised questions were eventually satisfied by the edits and the explanations